

# Explicit and parametrised representation of under ice shelf seas in a z* coordinate ocean model

Pierre Mathiot[1,2]; Adrian Jenkins[1]; Christopher Harris[2]; Gurvan Madec[3]

[1] British Antarctic Survey, NERC, Cambridge, UK
[2] Met Office, Exeter, UK
[3] Sorbonne Universités (University Pierre et Marie Curie Paris 6)-CNRS-IRD-MNHN, LOCEAN Laboratory, Paris, France

Correspondence to: Pierre Mathiot (pierre.mathiot@metoffice.gov.uk)

**Abstract.** Ice shelf/ocean interactions are a major source of fresh water on the Antarctic continental shelf and have a strong impact on ocean properties, ocean circulation and sea ice. However, climate models based on the ocean/sea ice model NEMO currently do not include these interactions in any detail. The capability of explicitly simulating the circulation beneath ice shelves is introduced in the non-linear free surface model NEMO. Its implementation into the NEMO framework and its assessment in an idealised and realistic circum-Antarctic configuration is described in this study.

Compared with the current prescription of ice shelf melting (i.e. at the surface) inclusion of open sub-ice-shelf leads to a decrease sea ice thickness along the coast, a weakening of the ocean stratification on the shelf, a decrease in salinity of HSSW on the Ross and Weddell Sea shelves and an increase in the strength of the gyres that circulate within the over-deepened basins on the West Antarctic continental shelf. Mimicking the under ice shelf seas overturning circulation by introducing the meltwater over the depth range of the ice shelf base, rather than at the surface is also tested. It yields similar improvements in the simulated ocean properties and circulation over the Antarctic continental shelf than the explicit ice shelf cavity representation. With the ice shelf cavities opened, the widely-used "3 equations" ice shelf melting formulation enables an interactive computation of melting that has been assessed. Comparison with observational estimates of ice shelf melting indicates realistic results for most ice shelves. However, melting rates for Amery, Getz and George VI ice shelves are considerably overestimated.

## 1 Introduction

Ice shelf melting, which accounts for 55% of the ice mass loss from Antarctica, is one of the main sources of fresh water input to the Antarctic coastal ocean. The net basal meltwater flux released to the Southern Ocean is estimated to be 1,500±237 Gt.y$^{-1}$ (or 48±8 mSv), compared with 1,265±141Gt.y$^{-1}$ (or 39±4 mSv) from iceberg calving (Rignot, et al., 2013). The total Antarctic mass discharge is thus similar to the 76 mSv due to surface atmospheric forcing (P-E) south of 63°S (Silva et al., 2006). The ice shelf melting contribution to the Southern Ocean freshwater forcing is very specific since the freshwater is injected into the ocean at depth whereas precipitation is input at the surface and calving injects water at a range





of depths, but primarily in the top ~100m. Therefore, the effect of ice shelf melting on coastal ocean stratification and circulation is very different from that of iceberg melt and precipitation.

The net ice shelf discharge (melting and calving) does not directly contribute to eustatic sea level change, because ice shelves are already floating, but does make a small steric contribution, because of the associated freshening (Jenkins and Holland, 2007). However, the strong mechanical coupling between ice sheet and ice shelf controls the ice flux across the grounding line from the ice sheet. Modifications to the ice shelf geometry associated with changes in ice thickness or extent lead to changes in buttressing at the grounding line. A reduction in buttressing can trigger a speed up of the discharge from the ice sheet, a process that has been implicated in widespread mass loss from the Antarctic Ice Sheet (Scambos et al., 2004, Rignot et al., 2004, Favier et al., 2014). Therefore, understanding of ice shelf/ocean interaction is a key factor in advancing our understanding of the ice sheet contribution to sea level rise.

Basal melting of ice shelves is driven by the properties of the water masses that spread over the continental shelves, enter the ocean cavities and reach the grounding line where they initiate melting. The associated input of buoyancy triggers an overturning circulation with inflow at depth and outflow along the ice shelf base that carries meltwater towards the surface. Jacobs et al. (1992) identified three modes of overturning, depending on the inflowing water mass that could be either High Salinity Shelf Water (HSSW, Mode 1), modified forms of Circumpolar Deep Water (CDW, Mode 2), or less saline water masses that could collectively be referred to as Antarctic Surface Water (AASW, Mode 3). Mode 1 melt rates are low, because HSSW has a temperature close to that of the surface freezing point and can melt ice at depth only because of the lowering of its freezing point with increasing pressure, Mode 2 melting can be high, if almost unmodified CDW has access to the sub-ice-shelf cavities, and Mode 3 melting is intermediate and variable, depending on whether only the near-freezing core of ASSW, often designated Winter Water (WW), or also the seasonally warmer upper layers can access the cavities. When the inflow has a temperature at or close to the surface freezing point (HSSW or WW), melting at depth is accompanied by partial refreezing at higher levels, as the falling pressure results in a rising freezing point temperature. The process is usually referred to as an ice pump (Foldvik, et al., 1985) and produces an out flowing water mass, designated Ice Shelf Water (ISW), that has a temperature below the surface freezing point. At the edge of the broad continental shelves of the southern Weddell and Ross seas and along the Adelie Land coast, ISW mixes with CDW and HSSW to form Antarctic Bottom Water (Foldvik, et al., 1985, Williams et al., 2008) that contributes to the global overturning circulation. A modelling study (Hellmer, 2004) further suggests that 20 cm of the total sea ice thickness in the Ross and Weddell seas results from the cooling and freshening of shelf water by ice shelf melting.

To improve the representation of the Antarctic coastal ocean and global sea level rise in the coupled Ocean/Sea-ice model NEMO, ice-shelf/ocean interactions need to be properly included. In previous NEMO simulations, ice-shelf melt was uniformly distributed around the coast of Antarctica and input at the surface. Furthermore, as the ocean/sea-ice model is also used as a component within Earth System Models, global conservation is an important issue. To tackle this issue the z* vertical coordinate has been included within the NEMO framework (Madec, 2012), and the ice shelf module as well as the ice shelf parametrisation are developed using z* coordinates and considering ice shelf melting as a mass flux.

This study is based on that of Losch (2008) (hereafter L08), describing the development of an ice shelf module within MITgcm. We follow a similar strategy to introduce ice shelf-ocean interactions into the NEMO framework (Madec, 2012). The work is a first step towards adding an ice sheet component and its interaction within NEMO, and including these interactions within climate models such as IPSL (Dufresne, et al., 2013), the Hadley Centre models (Hewitt, et al., 2011 and

Hewitt, et al., 2016), EC earth (Hazeleger, et al., 2010), CNRM (Voldoire, et al., 2013) and CMCC (Scoccimarro, et al., 2011).

Ice shelves range in size from the giant Ross ice shelf (500,000 km$^2$) to the tiny Ferrigno ice shelf (117 km$^2$). This means that current global ocean model configurations are not able to resolve explicitly all the ice shelf cavities. So whatever the resolution, some cavities remain unresolved. For this reason, a simple way to include unresolved ice shelf melting in the

ocean model that mimics the circulation driven by ice shelf melting at depth is also presented here.

The paper is structured as follows: firstly, the NEMO model, as well as the ice shelf module, are described, then idealized experiments are presented to validate the ice shelf module and ice shelf parametrisation, followed by its application to a realistic circum-Antarctic configuration at 0.25° resolution. The sensitivity of the ocean and sea ice properties to the inclusion of the ice shelf cavity, the effect of the ice shelf cavity parametrisation under prescribed ice shelf melting and the

resulting meltwater flux are then discussed. Finally, in a summary section, the major results as well as the remaining issues are highlighted, and we conclude with details of code availability.

## 2 Model description

### 2.1 Ocean model

NEMO (Nucleus for European Modelling of the Ocean) is a primitive equation ocean model, and this study uses version 3.6

of the code. The variables are distributed on an Arakawa C-grid (Arakawa, 1966), and we make use of the non-linear filtered free surface option. A complete description of the schemes and options available in NEMO is available in the documentation (Madec, 2012). A full description of the configurations used in this study is presented in Sect. 3.1 for the idealised configuration and in Sect. 4.1 for the realistic configuration.

### 2.2 Ice shelf/ocean interaction description

### 2.2.1 Ocean dynamics

The z* vertical coordinate in NEMO can be used with sea ice model (Madec, 2012). However, modelling the ocean circulation within an ice shelf cavity in z* coordinates requires some modification of the existing code. Beneath sea ice, the number of ocean levels is kept constant, and the levels are squeezed between the bottom surface of the ice and the seabed. The resulting pressure gradient error term is small because the ratio of sea ice thickness to total water column thickness is

small and almost spatially constant. Within an ice shelf cavity, a z* coordinate used as a surface following coordinate will




face the same limitation as terrain following coordinates, especially along the ice shelf front. The pressure gradient error will be large, particularly at the vertical ice front, and the tiny vertical cell thickness where the water column is thin will limit the stable time-step that is achievable.

To avoid these issues, we follow the idea of Grosfeld et al. (1997) for an s-coordinate model. All cells between the surface
(z=0) and the ice shelf base are masked at the model initialisation stage. By masking the ice shelf cells, the z* iso-surfaces are close to horizontal and the associated slopes are small, even at the ice front. Outside the ice shelf cavity, the definition of the cell thickness and the computation of the pressure gradient are not changed compared with the original code that follows Adcroft & Campin (2004). Within the cavities, the ice shelf thickness and the associated masked cells are constant over time, so the z* iso-surfaces are defined as:

$$Z_w(1) = \eta \tag{1}$$

$$if\ k < k_{isf},\ Z_w(k) = \sum_{kz=1}^{k_{isf}-1} dz_{0,T}(kz) \tag{2}$$

$$if\ k \geq k_{isf},\ Z_w(k) = \sum_{kz=1}^{k_{isf}-1} dz_{0,T}(kz) + \sum_{kz=k_{isf}}^{k-1} dz_{t,T}(kz) \tag{3}$$

$$dz_{t,T}(kz) = dz_{0,T}(kz)\left(1 + \frac{\eta}{H}\right) \tag{4}$$

with $Z_w$ the depth of the w interface (interface between 2 cells along the z axis in the Arakawa C-grid), $dz_{t,T}$ the vertical level
thickness at time t, $dz_{0,T}$ the vertical level thickness at time 0, k the model level, η the sea surface height, H the total water column thickness (sum of all the wet cells) and $k_{isf}$ the first wet level.

The pressure p at a depth z is computed in a standard way (Beckmann et al., 1999; L08). We assume the ice shelf to be in hydrostatic equilibrium in water at the reference density $\rho_{isf}$, taken to be the density of water at a temperature of −1.9°C and a salinity of 34.4. The total pressure at any depth is computed from the sum of the ice shelf load and the pressure due to the
water column above that depth. The pressure gradient is formulated as suggested by Campin et al. (2004) for z* coordinate models:

$$p(z) = \int_{z_{isf}}^{0} \rho_{isf} g\, dz + \int_{z}^{z_{isf}} \rho g\, dz \tag{5}$$

$$\nabla_z p(z) = \nabla_{z*} p(z) + \rho g \nabla_{z*} z \tag{6}$$

where p(z) is the pressure at depth z, ρ is the water density at depth z, and $z_{isf}$ is the ice shelf draft.

In this study, we assume the ice shelf to be in an equilibrium state (i.e. the ice shelf draft is temporally constant), so that all the ice melted by the ocean is assumed to be replaced by the seaward advection of new ice. The pressure of the ice shelf on the ocean therefore stays constant, but the ocean volume increases due to ice shelf melting. Dealing with an evolving ice shelf thickness is beyond the scope of this paper.

Representation of the bottom topography and flow along the sea floor is challenging in z coordinate models. The partial cell
scheme allows a more accurate representation of bottom topography through the use of partially wet cells (Adcroft et al., 1997). Solutions obtained with this scheme compare favourably with those obtained with sigma coordinates models (Adcroft et al., 1997) and also with more realistic solutions (Barnier et al., 2006). The same limitation is expected for the





representation of the flow along an ice shelf base. Following L08, we apply the partial cell scheme developed for the bottom topography to the top cells beneath the ice shelf base. For stability reasons, the minimum thickness of the bottom and top cells is set to the smaller of 25m or 20% of a full cell.

Where the water column is thinner than two cells, vertical circulation cannot be represented. In order to simulate the overturning circulation generated by ice shelf melting in such regions, we modify the bathymetry or the ice shelf draft sufficiently to open a new cell in the water column. In places where the cavity is thin and the slopes of the bathymetry and ice shelf draft are steep, it is sometimes necessary to create more than one new cell in order to open a minimum of two cells at the velocity points (at the centre of the cell faces on the Arakawa C-grid). Rather than making such extensive modifications to the topography, we regard the combination of vertical and horizontal resolution as too coarse to represent the sub-ice cavity geometry in these places, and instead we ground the ice shelf.

For regional configurations with open boundaries, the net inflow (the combination of inflow at the open boundary, runoff, ice shelf melting and precipitation) and net outflow (the combination of outflow at the open boundary, ice shelf freezing and evaporation) need to be balanced to avoid unrealistic sea surface elevation trends. This is achieved through a correction to the barotropic velocities at the open boundary.

### 2.2.2 Thermodynamics

Two formulations of the ice shelf melt rate are available: a simple one used in the idealised cases, for consistency with earlier studies, and a more sophisticated one used in the realistic configuration.

For the idealized study, the heat flux and the freshwater flux (negative for melting) resulting from ice shelf melting/freezing are parameterized following Grosfeld et al. (1997). This formulation is based on a balance between the vertical diffusive heat flux across the ocean top boundary layer (tbl) and the latent heat due to melting/freezing:

$$Q_h = \rho c_p \gamma (T_w - T_f) \tag{7}$$

$$q = \frac{-Q_h}{L_f} \tag{8}$$

where $Q_h$ (W.m$^{-2}$) is the heat flux, q (kg.s$^{-1}$.m$^{-2}$) the freshwater flux, $L_f$ the specific latent heat, $T_w$ the water temperature in the boundary layer, $T_f$ the freezing point computed from Millero (1978) using the pressure at the ice shelf base and the salinity of the water in the boundary layer, and $\gamma$ the thermal exchange coefficient. Hereafter, Eq. (7) and (8) are referred to as the ISOMIP formulation.

For realistic studies, the heat and freshwater fluxes are parameterized following Jenkins et al. (2001). This formulation is based on 3 equations: a balance between the vertical diffusive heat flux across the boundary layer and the latent heat due to melting/freezing of ice plus the vertical diffusive heat flux into the ice shelf (Eq. 10); a balance between the vertical diffusive salt flux across the boundary layer and the salt source or sink represented by the melting/freezing (Eq. 11); and a linear equation for the freezing temperature of sea water (Eq. 12, Jenkins, 1991):

$$c_p \rho \gamma_T (T_w - T_b) = -L_f q - \rho_i c_{p,i} \kappa \frac{T_s - T_b}{h_{isf}} \tag{10}$$





$$\rho\gamma_S(S_w - S_b) = (S_i - S_b)q \tag{11}$$

$$T_b = 0.0901 - 0.0575S_b - 7.61 \times 10^{-4}z_{isf} \tag{12}$$

$T_b$ is the temperature at the interface, $S_b$ the salinity at the interface, $\gamma_T$ and $\gamma_S$ the exchange coefficients for temperature and

salt, respectively, $S_i$ the salinity of the ice (assumed to be 0), $h_{isf}$ the ice shelf thickness, $\rho_i$ the density of the ice shelf, $c_{p,i}$ the specific heat capacity of the ice, $\kappa$ the thermal diffusivity of the ice and $T_s$ the atmospheric surface temperature (at the ice/air interface, assumed to be -20°C). The linear system formed by Eq. 10, Eq. 11 and the linearised equation for the freezing temperature of sea water (Eq. 12) can be solved for $S_b$ or $T_b$. Afterward, the fresh water flux (q) and the heat flux ($Q_h$) can be computed. $\gamma_T$ and $\gamma_S$ are velocity dependent (Jenkins et al., 2010) and can be written as:

$$\gamma_T = \sqrt{C_d}u_w\Gamma_T \tag{13}$$

$$\gamma_S = \sqrt{C_d}u_w\Gamma_S \tag{14}$$

with $u_w$ the ocean velocity in the tbl, $C_d$ the drag coefficient and $\Gamma_{T/S}$ a constant. The choices of the thermal Stanton number ($\sqrt{C_d}\Gamma_T = 0.0011$) and the diffusion Stanton number ($\sqrt{C_d}\Gamma_S = 3.1 \times 10^{-5}$) are based on the recommendation of (Jenkins et al., 2010). The drag coefficient is chosen to be $1.0 \times 10^{-3}$ . This value lies within the range used in the literature. However,

there are no direct measurements of the drag coefficient beneath an ice shelf. Dansereau et al. (2014) highlight that the range of values used for the top drag coefficient is large (from $1.0 \times 10^{-3}$ to $9.7 \times 10^{-3}$). Furthermore, uncertainties in the Stanton numbers are also large, as the study (Jenkins, et al., 2010) used to determine their values are based on data from a single borehole. Parameter values used in Eq. 7-12 are defined in Table 1. Hereafter, Eq. (10) to (12) are referred to as the "3 equation" ice shelf melting formulation.

Following L08, in the idealized experiments, the ice shelf forcing is applied as an effective heat flux and a virtual salt flux (no ocean volume change). For realistic configurations, the velocity divergence at the ice shelf base is adjusted in order to apply the ice shelf melting as a volume flux of freshwater at the freezing point temperature.

L08 shows that z coordinate models with partial cells generate a noisy melt rate pattern due to the variation of the top cell thickness. The melt rate is proportional to the difference between the in-situ basal temperature and in-situ temperature in the

first wet cell. Because the thickest cells cool down more slowly than the smallest cells, for a given initial basal temperature, the melt rate in the thickest cells is larger than in the smallest cells. Following L08, the noise due to the spatially varying size of the top cells is suppressed by computing $T_w$ and $S_w$ in Eq. 7,10 and 11 as the mean value over a constant thickness, assumed to represent the thickness of the tbl (i.e. properties are averaged over the first cell and that part of the second wet cell required to make up the constant boundary layer thickness). The top ocean velocity $u_w$ is defined as the velocity

magnitude derived from the mean zonal/meridional velocity at U/V points within the tbl averaged at T points. The heat and fresh water fluxes are distributed over the same constant thickness. If the first wet cell is thicker than the specified top boundary layer thickness, the tbl thickness is set to the full cell thickness. A complete description of this parametrisation is available in L08. Using z* instead of pure z coordinates does not alter the noise seen in the melt rate. Therefore, the





parametrisation proposed by L08 is applied in each simulation used in this study. The tbl thickness is set to a default value of 30 m, but different values are used for the simulations with varying vertical resolution, as presented in Table 2.

## 2.3 Simplified representation of ice shelf melting

Global ocean model configurations are typically unable to resolve all the ice shelves around Antarctica, which range in size from the vast Filchner-Ronne and Ross ice shelves to the much smaller ice shelves of the Bellingshausen and Amundsen seas. Despite their limited extent, the smaller ice shelves nevertheless make a significant contribution to the total meltwater flux from the ice sheet. We therefore need a way to mimic the impact of unresolved cavities on the ocean.

Beckmann and Goosse (2003, hereafter BG03) suggest a simple parametrisation for the melting beneath an ice shelf and prescribe the input of melt water at the ocean level corresponding to the base of the ice shelf (Figure 1c). One of the main issues with this parametrisation is that for the same ice shelf melting, the effect on the ocean dynamics will be the same whatever the grounding line depth is.

The idea tested in this paper is to spread the fresh water due to ice shelf melting evenly between the grounding line depth and the depth of the calving front. In this case, the model creates its own plumes along the vertical wall (Figure 1d, no cavity in this case) and thus an overturning between the grounding line depth and the equilibrium depth.

In this part of the study we focus on how to inject the observed ice shelf meltwater flux into the ocean model. Therefore, the ice shelf melting is prescribed and the heat flux is derived from the fwf using Eq. 8. The computation of the melt rate from the off-shore ocean properties and ice shelf geometry could be included using the Beckmann and Goosse (2003) parametrisation or some adaptation of the Jenkins (2011) plume model, but testing these interactive melt parametrisations is beyond the scope of the study.

## 3 Academic case

In order to compare the sub-ice shelf cavity capability in NEMO with that in other models, the idealized configuration used in this study is the one described in the Ice Shelf-Ocean Model Intercomparison Project (ISOMIP). ISOMIP is an open, international effort to identify systematic errors in sub-ice-shelf cavity ocean models and the reference configuration is based on a very simple setup, briefly described below.

## 3.1 ISOMIP setup

The ISOMIP setup follows the recommendations of the intercomparison project (Hunter, 2006). The geometry is based on a closed domain with a flat seabed fixed at 900 m. The grid extends over 15° in longitude, from 0 to 15°E with a resolution of 0.3°, and 10° in latitude, from 80°S to 70°S with a resolution of 0.1°. The spatial resolution ranges from 6 km at the southern boundary to 11 km at the northern boundary. The whole domain is covered with an ice shelf, and includes no open ocean





region. The ice shelf draft is uniform in the east-west direction, is set at 200 m between the northern boundary and 76°S and deepens linearly south of 76°S down to 700 m at the southern boundary.

The vertical resolution is uniform and fixed at 30 m, allowing a direct comparison with the results of L08. The density is computed using the polyEOS80-bsq function. It takes the same polynomial form as the polyTEOS10 function (Roquet et al.,

2015), but the coefficients have been optimized to accurately fit EOS-80 (Roquet, personnal communication). The melt formulation is the "ISOMIP" one. All the results presented are taken from day 10 000 at which time the system is in quasi-steady state.

## 3.2 Model comparison

The ISOMIP experiment has been carried out with many models using different vertical coordinates during the last 10 years,

including ROMS (http://www.ccpo.odu.edu/~msd/ISOMIP/), OzPOM (http://staff.acecrc.org.au/~johunter/isomip/isomip.html), MITgcm (Losch, 2008) and POP (Asays-Davis, 2013). All these models agree on a common circulation and melt pattern. The melting and freezing along the base of the ice shelf drives an overturning circulation of about 0.1 Sv. Associated with the meridional overturning circulation, all the models generate a cyclonic gyre with a western boundary current beneath the sloping ice shelf of about 0.3 Sv. This horizontal circulation

drives water that is warmer than the freezing point into the south-eastern part of the cavity. The inflow of warm water causes melting at the ice shelf base that is concentrated along the eastern and southern boundaries. On the western side of the ice shelf cavity, the boundary current advects colder water towards the ice front. Shoaling of the ice shelf base causes super-cooling of the water in contact with the ice and thus drives freezing. A detailed discussion of this circulation can be found in Grosfeld et al. (1997). The maximum melting/freezing rates are model dependent. The range is 0.7 - 1.8 m/y for the

maximum freezing rate and 0.7- 2.4 m/y for the maximum melting rate.

The NEMO response to the ISOMIP setup is shown in Figure 2. It is similar to the one previously simulated with a z coordinate model (L08). The strength of the overturning circulation is 0.11 Sv. The transport of the western boundary current generated by the cyclonic gyre beneath the sloping ice shelf is 0.32 Sv. The pattern of melting and freezing is similar to that in L08. The melting occurs as expected in the south-eastern corner with a maximum of 2.7 m/y and the freezing takes place

beneath the western boundary current with a maximum of 1.9 m/y. The low noise is the result of the Losch parametrisation (Figure 2). In simulations without this parametrisation (not shown) the noise in the melt pattern is as shown in L08.

## 3.3 Sensitivity of ocean circulation to the vertical resolution

Depending on the scientific question to be addressed, the ocean models commonly used have very different vertical resolutions, ranging from 1m to 100m. The representation of the top boundary layer is strongly affected by the choice of

vertical resolution. To evaluate the impact of this choice on the ocean circulation beneath the ice shelf, 9 simulations with vertical resolution ranging from 5m to 150m have been carried out (Table 2).



The choice of vertical resolution strongly affects the ice shelf melting, with finer resolution giving weaker melting. Under melting conditions, a thin, fresh and cold tbl appears in the top metres of the ocean next to the ice shelf base. With finer vertical resolution a thinner and colder tbl can be resolved, resulting in weaker melting (Fig. 3a). Our sensitivity experiments show a maximum melt rate 4 times higher in the 150M simulation (4.3 m.y$^{-1}$) and 3 times higher in the 60M simulation (3.1

m.y$^{-1}$) than in the 5M simulation (0.9 m.y$^{-1}$). In analogous experiments, L08 found a similar sensitivity, with maximum melting 3 times larger at 45m resolution than at 10m resolution. With very coarse resolution (100M/150M), the model is unable to represent a top boundary layer at all and the total melting saturates. Total melting is 20% smaller in the 5M simulation than in both the 100M and 150M simulations, which have the same total melt (Fig. 3a). With variable vertical resolution, such as is typically used in global configurations of NEMO, the coarsest resolution in the cavity seems to

determine the total melt. This could be an issue for modelling ice shelf melting with the standard configuration used for climate applications, which has 75 vertical levels and a resolution varying from 1m at the surface to 200 m at 6000 m depth, with a maximum of about 40 m beneath the major ice shelves.

The vertical resolution also has a major impact on the noise pattern (Fig. 4). As the noise in the melt pattern is closely linked with variations in the thickness of the first wet cell, the finer the vertical resolution, the weaker the noise.

In contrast, the barotropic stream function and the overturning circulation in the cavity are not altered by any choice of vertical resolution between 5 and 150m (Figure 3b). One of the reasons could be that with the bulk formulation of melting used in the ISOMIP simulations, there is no direct link between the ocean current velocity at the ice-shelf/ocean interface and the melt rate, because the thermal exchange coefficient is defined to be a constant.

## 4 Ice shelf cavity parametrisation

While the ice shelf module as described so far works well in idealised cases, for a wider range of applications (including ice shelves of varying extent at all likely horizontal resolutions) we also need the capability of representing the impact of circulation and melting within unresolved cavities. In this section, we investigate the ability of our ice shelf cavity parametrisation to mimic the circulation and water mass properties produced by the full cavity simulation, and compare the results with those produced by the parametrisation of BG03. Both parametrisations are evaluated in an idealised

configuration derived from the ISOMIP setup.

The geometry is the same as ISOMIP except in the top 200 m, where the flat ice shelf is replaced by open water (Figure 5a). The simulations are initialised with the warm linear profile suggested by Asay-Davis et al. (2016), typical of conditions on the continental shelves of the Amundsen and Bellingshausen seas, and radiative open boundary conditions are applied at the northern boundary (Treguier et al., 2001). The viscosity is increased from 600 m$^2$.s$^{-2}$ to 3000 m$^2$.s$^{-2}$ to damp the noise

generated along the ice shelf front.

Three experiments are run for 30 years: one with the ice shelf cavity open (A_ISF), but with a steady pattern of basal melt/freeze imposed; another with the open ocean circulation driven by the cavity parametrisation of BG03 (A_BG03); and a



third with the cavity parametrised as outlined in Sect. 2.3 (A_PAR). In all these experiments the same heat and freshwater fluxes are applied, derived from the basal melt/freeze pattern obtained in the last month of a dedicated 30-year run with explicit ice shelf melting calculated using the "ISOMIP" formulation.

A_ISF drives a deep inflow toward the ice shelf, and corresponding outflow in the top 400 m toward the open ocean, of 0.9 Sv at the northern boundary (Fig. 5a). In a stratified ocean, this circulation has a crucial effect on the total amount of heat advected toward the ice shelf, on the properties of the water drawn into the overturning circulation and on the overall stratification in the basin. In A_BG03 the overturning is too weak (0.2 Sv compared with 0.9 Sv in A_ISF) and too shallow (-200m compared with -400 m in A_ISF). Consequently, the water masses drawn into the overturning come from a different depth and have different T/S properties, and the resulting stratification is too strong, with colder surface waters and warmer deep waters (Fig. 5c). In A_PAR, because the freshwater flux is distributed over the same depth range as in A_ISF (between 200m and 700m), the vertical extent of the overturning and the water masses drawn in are the same in both A_PAR and A_ISF. The result is a circulation on the shelf that is similar in depth and magnitude and a stratification that is similar in strength to those simulated in A_ISF (Fig. 5b).

With far-field conditions typical of the cold, salty continental shelves of the Ross and Weddell seas, where the water column is well mixed by brine rejection from growing sea ice in winter and less heat is available at depth, the differences in the stratification resulting from the two parametrisations and the simulation with the open ice shelf cavity should be smaller.

## 5 Real ocean application

In the ISOMIP test cases, the ocean circulation in the cavity compares well with that simulated by other models. Furthermore, the suggested parametrisation of ice shelf melting mimics well the circulation and water properties generated by the presence of an open ice shelf cavity. Nevertheless, the bathymetry and ice shelf draft are smooth in these idealised cases and the heat transfer coefficient is constant, so the behaviour described above may differ in a realistic configuration. In the next section we assess both the explicit ocean cavity representation and the cavity parametrisation in a realistic circumpolar configuration.

### 5.1 Antarctic configuration setup

ePERIANT025 is a circum-Antarctic configuration based on the PERIANT025 configuration (Dufour et al., 2012) covering the ocean from 86.5°S to 30°S, using a ¼° isotropic Mercator grid. A feature of the Mercator grid is that the mesh spacing reduces with decreasing distance from the South Pole, so that the farthest south grid boxes strongly constrain the model time step. To maintain a model time step equal to that used in current global ¼° configurations, the Mercator grid is replaced south of 67°S with 2 quasi-isotropic bipolar grids, one for the Bellingshausen, Amundsen and Ross sea sector and one for the Weddell sea sector (Fig. 6). Each sector is built following the semi-analytical method used to create the tripolar ORCA grid north of 22°N (Madec and Imbard, 1996). The effective resolution is 13.8 km at 60°S, increasing to 3.8 km at 86.5°S, where





a pure Mercator grid would have a resolution of 2.2 km. The model uses 75 vertical levels with thicknesses varying from 1m at the surface to 200 m at 6000 m depth, giving a vertical resolution ranging from 10 to 150 m beneath the ice shelves.

The bathymetry used for the model domain north of the Antarctic continental shelf is that described by Megann et al., (2014). Over the Antarctic continental shelves the IBCSO data set (Arndt et al., 2013) is used, while under the ice shelves,

bathymetry and ice draft are taken from BEDMAP 2 (Fretwell et al., 2013). The resulting model bathymetry is shown in Fig. 6. Note that we impose a minimum of 2 vertical grid cells within the ocean cavities so that an overturning cell can develop. Where necessary, either the bathymetry or the ice shelf draft, depending on the local configuration, is modified to fit the criterion. If more than 1 cell has to be modified to fit the water column criterion, the entire water column is masked. Using this procedure, Totten and Moscow University ice shelves and the deepest part of Amery Ice Shelf are almost fully masked.

The model uses z* coordinates (Adcroft and Campin, 2004), and bottom and ice shelf topography are represented with partial steps (Barnier et al., 2006 and L08). Other choices (the momentum advection, tracer advection, diffusion, viscosity, vertical mixing, double diffusion, bottom friction, bottom boundary layer and tidal mixing parametrisations) are as used in Megann et al., (2014). For the sea ice we use the Louvain-la-Neuve sea-ice model LIM2 (Fichefet and Morales, 1997) with ice rheology based on an elasto-visco-plastic law as described in Bouillon et al. (2013).

The geothermal heat flux is assumed to be constant and set to 86 mW/m$^2$ (Emile-Geay and Madec, 2010), while tidal mixing data come from FES 2012 (Carrère et al., 2012). Sea surface salinity restoring is applied north of 55°S, river runoff comes from Dai and Trenberth (2002), and iceberg melting based on Rignot et al. (2013) is evenly distributed at the surface along the Antarctica coast. Ice shelf melt is applied either into the open cavities, at depth following our parameterisation, or as surface runoff. The total ice shelf melt in each individual cavity is either interactively computed using the "3 equation"

formulation or prescribed following the Rignot et al. (2013) estimates.

Radiative boundary conditions are applied at the northern open boundary (Treguier et al., 2001) using velocity, temperature and salinity data from a global NEMO ORCA025 simulation (Barnier et al., 2012) forced by the DFS5.2 atmospheric forcing developed by the DRAKKAR project. To minimize inconsistency, the model is also driven by the same DFS5.2 atmospheric forcing. The methodology applied to build the DFS forcing series is described in Brodeau et al. (2010), and the details of the

DFS5.2 are given in a report by Dussin et al. (2016). Initial conditions come from the World Ocean Atlas 2013 (Locarnini et al., 2013 and Zweng et al., 2013). The model is run for 10 years starting in 1976, and the first order response is investigated using output from the last year of the simulation.

## 5.2 Experiment description

In order to evaluate both the explicit ice shelf module (Sect. 2.2) and the improved parametrisation (Sect. 2.3) in this realistic

case, 4 simulations are run:

- R_noISF: a simulation without ice shelf cavities. Both the ice shelf freshwater flux and the latent heat flux associated with melting of the ice are prescribed at the surface (Fig. 1a).





- • R_ISF: a simulation with explicit ice shelf cavities (Fig. 1b), but where both the melt rate of the ice shelves and the latent heat flux at the ice shelf/ocean interface are specified.

- • R_PAR: a simulation without ice shelf cavities (Fig. 1d). Both freshwater and latent heat fluxes from the ice shelves are uniformly distributed along the calving front from its base down to the grounding line depth, or the seabed if it is shallower.

- • R_MLT: a simulation with explicit ice shelf cavities and interactive melt rates computed by the "3 equation" formulation (Fig. 1b).

For R_ISF, R_noISF and R_PAR the same total inputs of fresh water and latent heat are prescribed for each ice shelf and the fluxes are constant over time; only the location of the input changes. The melting pattern used in R_ISF is provided by the simulation R_MLT, while the magnitude is scaled so that the total for each ice shelf matches that from Rignot et al. (2013). The associated latent heat flux is derived from the melt rate using Eq. 8.

Initially, results from R_noISF and R_ISF are used to evaluate the sensitivity of the ocean and sea ice properties to the presence of ice shelf cavities in a control setup with prescribed melting. Next, results from R_PAR are compared with those from R_noISF and R_ISF in order to evaluate and validate the ice shelf parameterisation in a realistic case. Finally, results from R_MLT are used to evaluate the "3 equation" ice shelf melting formulation in NEMO.

### 5.3 Sensitivity of ocean properties to the ice shelf cavities

In both R_noISF and R_ISF, large-scale open ocean features are well represented. Simulated ACC transport (135 Sv) and Weddell gyre transport (56 Sv) are similar and compare well with the observations of 137 Sv for the ACC transport (Cunningham et al., 2003) and 56 Sv for the Weddell gyre transport (Klatt et al., 2005). Temperature and salinity properties north of the continental shelves are also similar in both simulations and compare reasonably with WOA2013 (Fig. 7-8). In contrast, the presence of ice shelf cavities in R_ISF substantially affects the ocean properties and dynamics in the coastal Antarctic seas (Fig. 7, 8 and 10).

Over the Bellingshausen and Amundsen seas, the input of fresh water at the surface in R_noISF leads to strong stratification in the upper 250 m, weak stratification below (Fig. 9), a weak and shallow vertical circulation (maximum overturning is 0.01 Sv at 70 m depth) and a weak barotropic circulation over the continental shelf (Fig. 10). In R_ISF, the input of buoyancy at the ice shelf base activates the buoyancy-forced overturning, driving upwelling along the ice shelf/ocean interface. The overturning circulation entrains 0.23 Sv of a mix of ambient water (CDW) and meltwater along the ice shelf base. This upwelling generates a barotropic circulation that follows the f/h contours over the Amundsen and Bellingshausen sea continental shelf (Fig. 10a,c) as explained in Grosfeld et al. (1997). The resulting mixture of CDW and meltwater stabilises at an equilibrium depth between 400 m and 60 m (not shown). The upwelling of CDW into the surface mixed layer weakens the thermohaline stratification and warms and salinifies the surface layer. These changes in ocean dynamics on the shelf lead




to a more realistic continental shelf temperature and salinity distribution (Figure 7-8) and stratification (Fig. 9) in R_ISF compared with R_noISF.

In Pine Island Bay and elsewhere on the Amundsen and Bellingshausen Sea shelves, the bottom water properties in the over-deepened basins are determined by the properties in the open ocean at the sill depth (Walker et al., 2007) close to the shelf

break. So the bottom temperature bias present in R_ISF could be strongly affected by the model bias in the ACC, the possible sources of which are beyond the scope of this paper. In R_noISF, as the overturning is not activated, there is no process to flush the bottom water trapped in the over-deepened basins, so the waters there are not affected by external forcing, and the bottom properties still match the initial conditions after 10y of the run (Fig. 9).

Over the Ross and Weddell Sea continental shelves, the cold, salty HSSW in R_noISF matches the observations and spreads

northward across the shelf break toward the open ocean. In R_ISF, the HSSW produced is too fresh (-0.2 PSU, Fig. 8). Weak winds in the atmospheric forcing (Dinniman et al., 2015) associated with a fresher coastal current (Nakayama et al., 2014), the opening of a new pathway for HSSW circulation beneath the ice shelves (Budillon et al., 2003; Nicholls et al., 2009), mixing of HSSW with light surface waters all year long, and a deficiency of the sea-ice model in representing coastal polynyas could all help to explain the absence of HSSW in R_ISF.

**5.4 Sensitivity of sea ice properties to the ice shelf cavities**

Winter sea ice extent compares well with the 18.3 million km$^2$ estimated from satellite observations (Comiso, 2000) in both R_ISF (18.2 million km$^2$) and R_noISF (18.4 million km$^2$). The position of the ice edge is not changed significantly by the presence of ice shelf cavities, being too far south in Amundsen Sea and too far north in Weddell Sea and around East Antarctica in both simulations (Fig. 11).

Over the warm continental shelves of the Amundsen and Bellingshausen seas, sea ice is thicker in the R_noISF than in the R_ISF simulation (+1 metre). In R_noISF, because the fresh water and the latent heat sink from the melting of land ice are prescribed at the surface, the consequent freshening and cooling of the surface waters considerably enhances the formation of sea ice. This leads to very thick sea ice in R_noISF, greater than 3 m locally (Fig. 11c). In R_ISF, the overturning circulation driven by melting at the ice shelf ocean interface entrains warm CDW and mixes it into the surface layer. This

upward heat flux decreases the sea ice formation and has a direct effect on sea ice thickness (Fig. 11a). Sea ice is thus thinner in R_ISF than in R_noISF, although it still exceeds 1 m along the coast (Fig. 11a).

Over the cold continental shelves of the Ross and Weddell seas and around the coast of East Antarctica, sea ice thickness differences between R_ISF and R_noISF are much smaller, typically about 20 cm (Fig. 11). The ocean is well mixed and the shelf water temperature is close to the freezing point (Fig. 7). So the amount of heat entrained into the buoyant overturning

along the ice shelf base is smaller, as the impact on sea ice.

Comparison with spring sea ice thickness estimates derived from sea-ice freeboard and snow thickness measurements (Fig. 11d, Kurtz & Markus, 2012) shows that sea ice thickness in R_ISF is closer to observation by about 1 m over the warm




shelves of West Antarctica. Over the cold shelves, the modelled sea-ice thicknesses are similar in both simulations (less than 20 cm differences) and comparable with the observations, which are subject to +/- 40 cm uncertainties.

## 5.5 Assessment of the simplified ice shelf representation

The implementation of the ice shelf cavities in a realistic configuration showed a great improvement in the circulation on the
Antarctic continental shelves, especially in the Amundsen and Bellingshausen seas. However, many climate models lack the horizontal and vertical resolution needed to represent all these cavities. Our parametrisation described in Sect 2.3 has been developed to address this issue. The evaluation of our parametrisation in a simple idealised case showed very encouraging results. Here, by comparing R_ISF and R_noISF with R_PAR, we evaluate the parametrisation for all ice shelves of the Southern Ocean.

Over the warm shelves of West Antarctica, R_PAR reproduces well the R_ISF shelf properties and circulation (Fig. 12a-b and Fig. 10). Critically, the prescription of the ice shelf meltwater flux at depth drives an overturning circulation and spins up the associated gyres within the over-deepened basins. The magnitudes of the gyres are similar in both the R_ISF and the R_PAR simulations (Fig. 10b,c). Shelf water properties generated by R_ISF are better reproduced by R_PAR than by R_noISF over all the West and East Antarctic shelves (Fig. 12a-d). Over the Amundsen shelf, R_PAR also decreases the
stratification and improves the mean temperature and salinity profiles compared with R_noISF (Fig. 9).

Over the Ross and Weddell sea shelves, HSSW produced in R_PAR is saltier than in R_ISF (+0.1 PSU). The salinity gradient between the salty western side and the fresher eastern side of the shelves is larger than in R_ISF (Fig. 12c) and larger than in the observations (Fig. 8). This is due to the lack in R_PAR of a HSSW circulation pathway beneath the giant Ross (Budillon et al., 2003) and Filchner-Ronne (Nicholls et al., 2009) ice shelves that in reality carries HSSW formed in the
west over to the central or eastern shelf. Instead of this sub-ice shelf circulation that is captured in R_ISF (Fig. 10), R_PAR drives individual gyre circulations within each of the over-deepened basins, similar in structure to, but stronger than, those in R_noISF.

Sea ice extent and thickness in R_PAR match well the R_ISF sea ice characteristics (Fig. 11). Thickness is smaller by more than 1m in West Antarctica compared with the R_noISF simulation. Around East Antarctica, and over the Ross and Weddell
sea shelves, despite the deficiency in representing the giant ice shelves, sea ice thickness in R_PAR is similar to that in R_ISF.

These comparisons between R_ISF/R_PAR and R_noISF suggest that not only the presence and the amount of melt water are important, but also the depth at which it is input to the model. The parametrisation directly addresses this latter feature of the sub-ice-shelf ocean circulation and so is able to represent the ocean dynamics associated with the overturning circulation
within the cavity. However, the parametrisation is not fully adapted to mimic the large-scale horizontal gyre circulation that is spun up under the giant ice shelves. Nevertheless, current coarse resolution ocean models have a nominal resolution of $1° × \cos(latitude)$ which is sufficient to explicitly represent the two giant ice shelves (L08, Hellmer et al., 2004, Hellmer et al., 2012).



## 5.6 Ice shelf melting

In the previous section we showed that specifying a realistic melting pattern at the ice shelf/ocean interface gives convincing results with major improvements in the properties and circulation of the ocean beyond the ice shelves, especially in the Amundsen and Bellingshausen seas. However, prescribing the freshwater flux represents a strong constraint on the range of
applications, since the specified fluxes will only be valid for the present oceanic state. To compute melt rates for other oceanic states interactively, and eventually to couple the ocean model to an evolving ice sheet model, requires the "3 equation" formulation for ice shelf melting, which we evaluate next.

The total ice shelf melting simulated in R_MLT (1865 Gt.y$^{-1}$) is slightly above the range of the observational estimate of Rignot et al. (2013) (Table 3). In R_MLT, as in the observations, we can separate the ice shelves into two different regimes
based on the temperature of the water masses on the continental shelves (Fig. 7d) and the average melt rate: the cold water (Fig. 13b-d) and the warm water (Fig. 13a) ice shelves.

### 5.6.1 Cold water ice shelves

For the Ross, Weddell and East Antarctic continental shelves, the agreement between computed and observed ice shelf melt rates varies. The total melt in R_MLT for these ice shelves (818 Gt.y$^{-1}$) lies within the range of the observations (475-867
Gt.y$^{-1}$) (Table 3). These ice shelves all experience low melt rates (Fig. 13b-d) due to the presence of cold water on the shelves (Fig. 8).

For Filchner-Ronne Ice Shelf (FRIS) the total melt in R_MLT is in agreement with the observation based estimates (Table 3), while the spatial pattern of melting and freezing is also similar to other simulations without tidal forcing (Makinson, et al., 2012). FRIS experiences strong melt close to the grounding line, along the ice front and along the paths of the main
inflows. Large freezing rates occur along the paths of the main outflows that follow the eastern coasts of the Antarctic Peninsula, Berkner Island and Henry Ice Rise. The latter generates a particularly large area of intense freezing in the central part of the ice shelf, north of the ice rises, in agreement with the observation based distributions of Joughin and Padman (2003) and Moholdt et al. (2014).

For Ross Ice Shelf, R_MLT generates a total melt of 111 Gt.y$^{-1}$, with high melt rates concentrated along the ice front, and
lower freezing rates in the central part of the ice shelf (Fig. 13). The total melt is within the range of previous model based estimates (51 - 260 Gt.y$^{-1}$) and the melting/freezing pattern is in good agreement with earlier modelling studies (Timmermann et al., 2012, Holland et al., 2003 and Dinniman et al., 2007). However, the total melt simulated in R_MLT is 30 Gt.y$^{-1}$ above the observational range, because melt rates along the ice front and on the western side of the ice shelf are larger than those inferred from observation (Rignot, et al.,2013; Moholdt, et al.,2014).

Total melt of Amery Ice Shelf is overestimated by at least a factor of 5 (Table 3), because the waters on the continental shelf in front of the cavity are warmer than observed by more than 1.2°C (Fig. 14). As a consequence, the freezing within the



cavity, evaluated from remote sensing and in situ data (Wen, et al.,2010) and simulated by Galton-Fenzi et al. (2012), is absent in R_MLT.

### 5.6.2 Warm water ice shelves

The ice shelves along the West Antarctic coastline between the Ross and Weddell seas experience a large integrated melt rate in R_MLT (1142 Gt.y$^{-1}$) (Fig. 12a), due to the presence of CDW on the continental shelf. This total melt is about twice the recent observation based estimate (541, Gt.y$^{-1}$) (Table 3).

The melt rates in R_MLT are realistic for Abbot Ice Shelf (52 Gt.y$^{-1}$) (Table 3), but slightly underestimated for Thwaites (74 Gt.y$^{-1}$) and Pine Island Glacier (PIG, 87Gt.y$^{-1}$) compared with observation (Table 3). By comparison with previous modelling studies, R_MLT results for Abbot and PIG ice shelves are in the range of earlier work (Timmermann, et al., 2012; Nakayama, et al., 2014; Shodlock et al., 2016) while for Thwaites the results are above those obtained previously.

Most of the warm ice shelf melting overestimate in R_MLT comes from Getz (337 Gt.y$^{-1}$) and George VI (298 Gt.y$^{-1}$) ice shelves (+178Gt.y$^{-1}$ and +181Gt.y$^{-1}$ respectively, table 3). R_MLT estimates are also well above earlier estimates obtained with FESOM by Timmermann et al. (2012) and Nakayama, et al. (2014), respectively, 164 and 127 Gt.y$^{-1}$ for Getz Ice shelf, and 86 and 88 Gt.y$^{-1}$ for George VI Ice Shelf. However, Schodlok et al, (2016) obtained similar melt rates using MITgcm (respectively 303.9 and 373.1 Gt.y$^{-1}$).

These large inter-model differences could have two causes. First, the ability of off-shelf CDW to cross the shelf break and spread across the continental shelf is a key control on the water mass structure within the ice shelf cavities. In R_MLT (Fig. 14) and MITgcm (Shodlok et al., 2016), CDW flow onto the shelf is well established. However, in the FESOM simulations of Nakayama et al. (2014), the shelf water is colder than the observations by 0.5°C to 3°C, depending of the horizontal resolution used. Analysis of why CDW can cross the continental shelf break in some models and not in others is beyond of the scope of this paper.

Secondly, NEMO and MITgcm both use z coordinates, while FESOM is a sigma-coordinate model. In sigma coordinate model the vertical resolution within the cavity is higher due to the concentration of level beneath the ice shelf. In R_MLT, the number of wet levels in the cavities varies from ~10 levels near the ice fronts to 2 levels at the grounding line, while in FESOM there are 21 levels everywhere. This allows better resolution near the grounding line and in the top boundary layer. Shodlok et al. (2016) and the sensitivity experiments performed in Sect. 3.3 show that some ice shelves (West, Moscow University, Totten, George VI, Larsen C and FRIS for example) are highly sensitive to the vertical resolution, which affects the ocean properties on the continental shelf, the representation of the top boundary layer beneath the ice shelf, and the ability to resolve details of the cavity geometry.



### 5.6.3 Limitations

In addition to the inter-model differences described above, ice shelf/ocean models in general are still subject to several limitations. Most of them are specific to our model setup as the large uncertainties in geometry and forcing data, and critical gaps in our knowledge of dynamics at the ice/ocean interface.

For many ice shelves, there are only indirect observations of ice draft, based on satellite surface elevation data, while the sub-ice bathymetry is often based on nothing more than extrapolation from surrounding regions of grounded ice and open ocean. More data are needed for effective modelling (Fretwell, et al., 2013), because cavity geometry has a major impact on the simulated melting by controlling the water mass structure and circulation within the cavity (Rydt, et al., 2014).

Tides have a strong impact on high frequency variability in melting as well as the magnitude and spatial pattern of the
temporal mean melt rate (Makinson et al., 2012), but they are not taken into account in the present study.

Subglacial runoff can enhance melting at the ice/ocean interface, especially near the grounding line (Jenkins, 2011). However, the location, magnitude and variability of subglacial outflows from beneath the Antarctic Ice Sheet are poorly known (Dierssen et al, 2002, Fricker et al., 2007).

The drag coefficient, as well as the friction law directly, affect the top velocity and hence the turbulent exchange coefficients
(Eq. 13 and 14). The appropriate drag coefficient for the base of an ice shelf of unknown roughness is highly speculative, and the range of values discussed in the literature is wide, ranging from $1.5 \times 10^{-3}$ (Holland and Jenkins, 1999) to $9.7 \times 10^{-3}$ (Jenkins, et al., 2010), while the basal melting simulated in models is very sensitive to the value chosen (Dansereau et al. 2014; Gwyther, et al., 2015, Jourdain et al., 2016). Furthermore, the friction law commonly used to compute the drag is overly simplistic. The same drag coefficient and friction law are used to compute the stress whatever the dynamic regime
appropriate for the grid point location beneath the ice shelf (i.e. whether it lies within the boundary layer or the free stream flow beyond).

In addition, Dutrieux et al. (2013) suggest that melting can be concentrated around kilometre-scale heterogeneities in ice thickness, such as keels and channels, especially near the grounding line. Furthermore, Stanton et al. (2013), from density measurements in the top 30m of the ocean beneath Pine Island Glacier, suggest that the top boundary layer can be less than 5
m thick. This means either very high horizontal and vertical resolution or a better melt formulation, or both, are needed to improve the representation of processes near the grounding line and the ice shelf base.

### 6 Conclusions

An ice shelf capability has been implemented and evaluated in the NEMO model framework following Losch et al. (2008). The work represents the first step toward a couple ice sheet/ocean model. The working hypothesis used here is that the ice
shelf is in equilibrium, with the mass removed by melting being replenished by the flow of the ice shelf, so the shape of the sub-ice-shelf cavity remains constant over time.



In an idealised case (ISOMIP setup) the simulated ocean circulation and ice shelf melting are similar to those described by Losch et al. (2008) using the MITgcm model. Ice shelf melting appears to be very sensitive to vertical resolution. At high vertical resolution, models are better able to simulate the cold, fresh, top boundary layer that occurs under melting conditions and that tends to decrease the thermal forcing and thus the simulated melt rate. At coarse resolution, the cold, top boundary layer is absent, leading to much larger melt rates.

To apply this work to a realistic case, a southward-extended global ORCA grid (eORCA) has been set up using two quasi-isotropic bipolar grids south of 67°S. The impact of including the ice shelf cavities has been evaluated in a circum-Antarctic version of the eORCA grid, by comparison with a control simulation without ice shelf cavities. The fresh water and heat flux resulting from ice shelf melting is specified at the ice shelf/ocean interface for the simulation with cavities and at the ocean surface for the control run.

For warm water shelves, prescribing the ice shelf melting at the surface (R_noISF) leads to a stratification that is too strong compared with the observations. With ice shelf cavities included (R_ISF), melting into the cavity drives a buoyant overturning circulation and entrains warm and salty CDW into the upwelling branch that subsequently mixes into the cold, fresh surface layers outside of the cavity. The entrainment of CDW thus weakens the thermocline by warming and increasing the salinity of the upper ocean layers, resulting in a decrease of the ocean stratification. The activation of the overturning circulation also creates a barotropic circulation that follows f/h contours on the continental shelf.

For cold water shelves, High Salinity Shelf Water (HSSW) is slightly lighter than observed in R_noISF, but when ice shelf cavities are present, the model is unable to maintain HSSW on the shelf at all. Compared with the simulation without ice shelf cavities, two extra processes consume the HSSW. The vertical overturning circulation driven by melting acts to mix the HSSW with the upper layers all year long, and the presence of new pathways beneath Ross and Filchner-Ronne ice shelves increases the export of HSSW from its formation location on the western continental shelf. The loss of HSSW with the ice shelf cavity opened is not balanced by increased dense water formation at the surface. This could be a result of deficiencies in any or all of the atmospheric forcing, the sea-ice model used in this study, or the representation of coastal polynyas.

The effects on sea ice are very dependent on the amount of ocean heat available at depth. Over warm water shelves, the CDW entrained into the cavity overturning circulation warms the surface layer all year long and thus restricts the sea ice formation. This warming of the surface layer leads to thinning of the sea ice by more than 1m in coastal regions of the Bellingshausen and Amundsen seas (2m locally). Over cold water shelves, including the sub-ice-shelf cavities has a smaller effect on sea ice thickness (less than 20 cm).

Hence, the inclusion of the ice shelf capability in NEMO has a major impact on ocean and sea ice properties. However, the ice shelves vary greatly in area, from O(100 km$^2$) to O(100 000 km$^2$), so depending on the application, more or fewer ice shelves will remain unresolved. In our ¼° configuration the unresolved ice shelves contribute 25% of the total ice shelf meltwater flux from Antarctica, and at coarser resolutions the majority of the fresh water source could be missing.

To mimic the circulation driven by these unresolved ice shelves, the ice shelf melting is uniformly distributed over the depth and width of the unresolved cavity opening, from the mean ice front draft down to the seabed, or the grounding line depth if





it is shallower. This simple representation of the ice shelf melting drives a buoyant overturning circulation along the coast similar to that that would be present within the ice shelf cavity. Idealised and realistic circum-Antarctic experiments show that this parametrisation mimics the effect of the overturning circulation within small ice shelf cavities and its impact on water mass properties and circulation on the continental shelf. However, for large ice shelves, such as Ross and Filchner-

Ronne, the parameterisation is unable to mimic the effect of the large-scale horizontal ocean circulation beneath the ice shelf. Thus, the redistribution of melt water and High Salinity Shelf Water between the different troughs on the continental shelf via their connections under the ice shelf is missing.

The specification of ice shelf melting, either over the area of the ice shelf base for resolved cavities or over the area of the cavity opening for unresolved cavities, leads to major improvements in the water mass properties, ocean circulation and sea

ice state on the Antarctic continental shelf. However, a model that interactively computes ice shelf melting is crucial for simulating the ocean and ice sheet response to perturbations as well as for developing coupled ice-sheet-ocean models. With the parameterised version of the ice shelf presented here, we only explain how to prescribe the melt rate in an ocean model without ice shelf cavities in a physically-sensible way. We do not describe a way to compute the melt rate itself. To tackle this issue, this work needs to be combined with a parameterisation of ice shelf melting (for example: Beckmann and Goosse,

2003; Jenkins et al., 2011).

With the ice shelf cavities opened, the widely-used "3 equation" ice shelf melting formulation enables an interactive computation of melting that has been assessed in the circum-Antarctic configuration. Comparison with observational estimates of ice shelf melting reported by Rignot et al. (2013) indicates realistic results for most ice shelves. However, melting rates for Amery, Getz and George VI ice shelves are considerably overestimated and some key ice shelves, such as

Totten and Moscow University, are missing because of inadequate horizontal and vertical resolution. Possible causes of the overestimated melt rates include poor representation of shelf water properties, inaccurate or poorly resolved cavity shape, unknown ice shelf ocean drag coefficient and poor representation of boundary layer processes.

Despite some deficiencies in the simulation of ice shelf melting and the parametrisation of ocean processes in unresolved ice shelf cavities, this work is a step forward toward a better representation of ice-shelf-ocean interaction in the NEMO

framework for all model resolutions. In practice, for horizontal resolutions finer than 2°, some of the ice shelf cavities can be resolved (Ross ice shelf for example), while at almost any useable resolution some cavities will have to be parametrized. The most suitable choice of which can be explicitly resolved and which must be parameterised will depend on the combination of horizontal and vertical resolution used.

To apply this work to a coupled ice sheet/ocean model, we will need some further developments. First, a better knowledge of

sub-ice-shelf cavity geometries and key processes that contribute to melting (drag, tides, etc ...) could lead to improvements in the ice shelf representation. Secondly, parameterisations need to be developed to represent the non-hydrostatic processes and land ice/ocean interactions along vertical ice faces such as the calving fronts of ice shelves and tidewater glaciers. Finally, a conservative wetting and drying scheme needs to be developed to allow the grounding line (and calving front) to move back and forth.




**Code availability**

The model code for NEMO 3.6 is available from the NEMO website (www.nemo-ocean.eu). On registering, individuals can access the FORTRAN code using the open source subversion software (http://subversion.apache.org/). The branch used for both configurations used in this study is the 2015 development branch named dev_r5151_UKMO_ISF at revision 5200. The

ice shelf module is now included in the public NEMO distribution.

**Data availability**

The ISOMIP configuration is distributed in NEMO version 3.6 as an unsupported configuration. No file is required to run ISOMIP configuration. For the Circum-Antarctic configuration, the input files (cpp keys, namelist, bathymetry, ice shelf draft, iceberg runoff, initial condition, river runoff, tidal mixing, and weights for the surface forcings) could be requested

from the authors. The surface forcing and the open boundary were provided by the DRAKKAR consortium (http://www.drakkar-ocean.eu).

**Acknowledgements**

The authors acknowledge financial support from the National Environmental Research Council and the UK Met Office. Computational resources were provided by the supercomputing facilities of the British Antarctic Survey and the ARCHER

UK National Supercomputing Service. The DFS5.2 forcing fields and the open boundary conditions from the ORCA025-GRD100 simulation were provided by the DRAKKAR coordination (CNRS GDRI No 810).

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



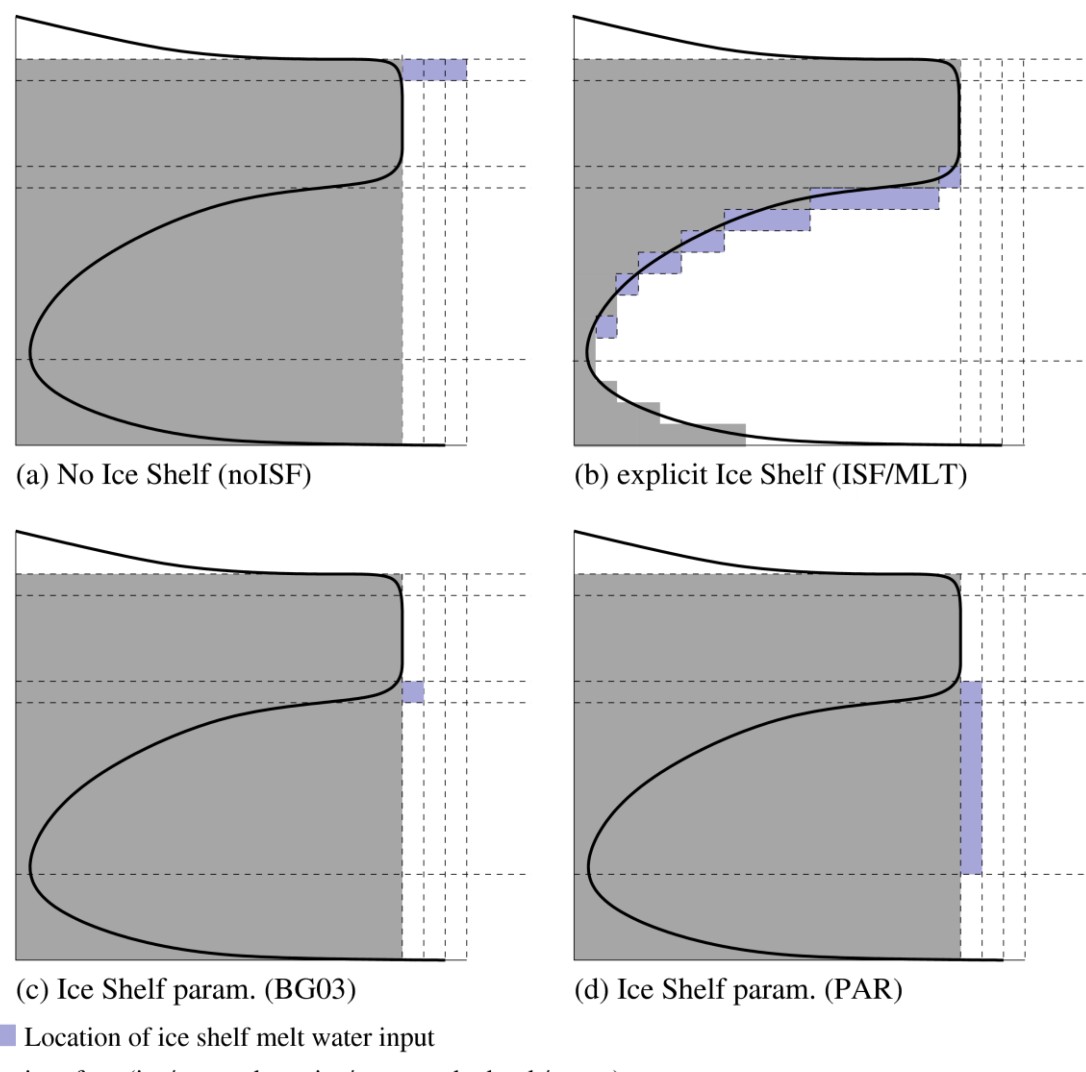

Figure 1: Freshwater and associated latent heat introduced a) at the surface (R_ noISF), b) beneath the ice shelf (A_ISF, R_ISF and R_MLT), c) at the ice shelf base level (A_BG03) and d) over the depth range of the ice shelf base (A_PAR and R_PAR).



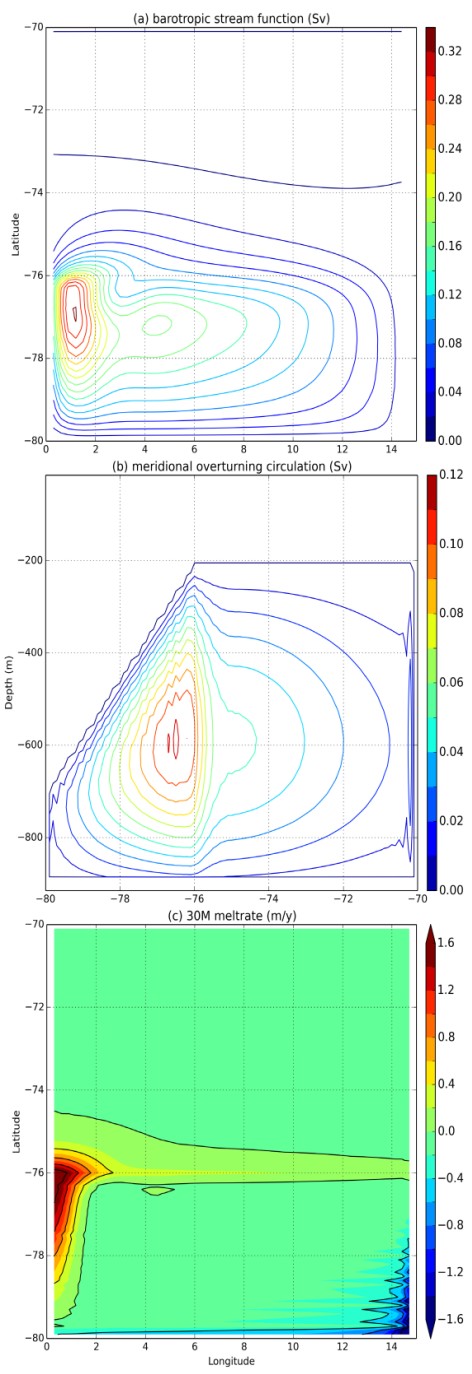

**Figure 2: Near steady state (after 10000 days) solution of the ISOMIP experiment. a) Melt rate in m/y (positive for melting and negative for freezing) with a contour interval of 0.4 m/y. b) Meridional overturning circulation (moc) in Sv with a contour interval of 0.01. c) Horizontal stream function (Psi) in Sv with a contour interval of 0.02 Sv**





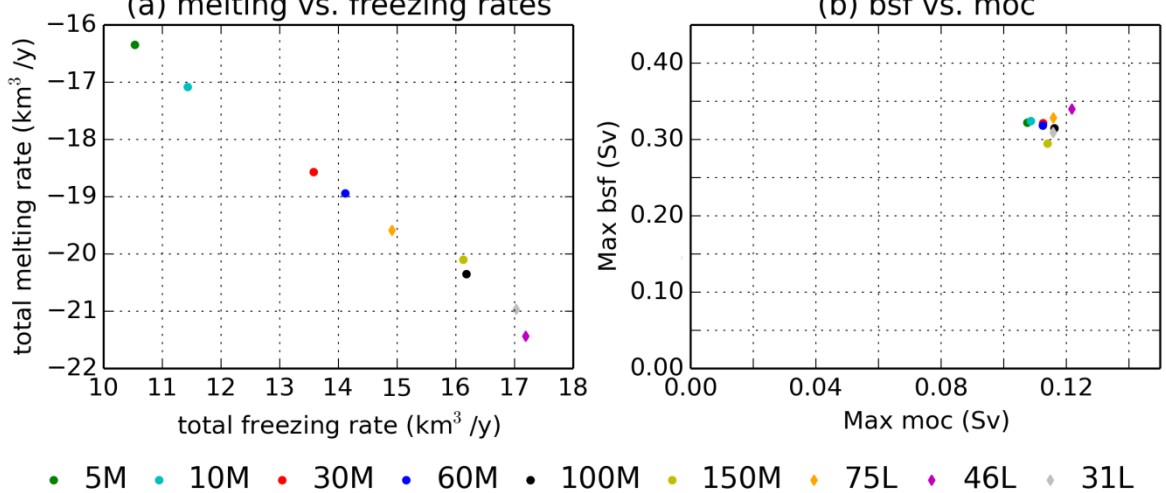

**Figure 3: a) Total melting rate versus total freezing rate, and b) meridional overturning circulation versus barotropic stream function (bsf) for all the ISOMIP sensitivity experiments (5M, 10M, 30M, 60M, 100M, 150M, 31L, 46L and 75L).**

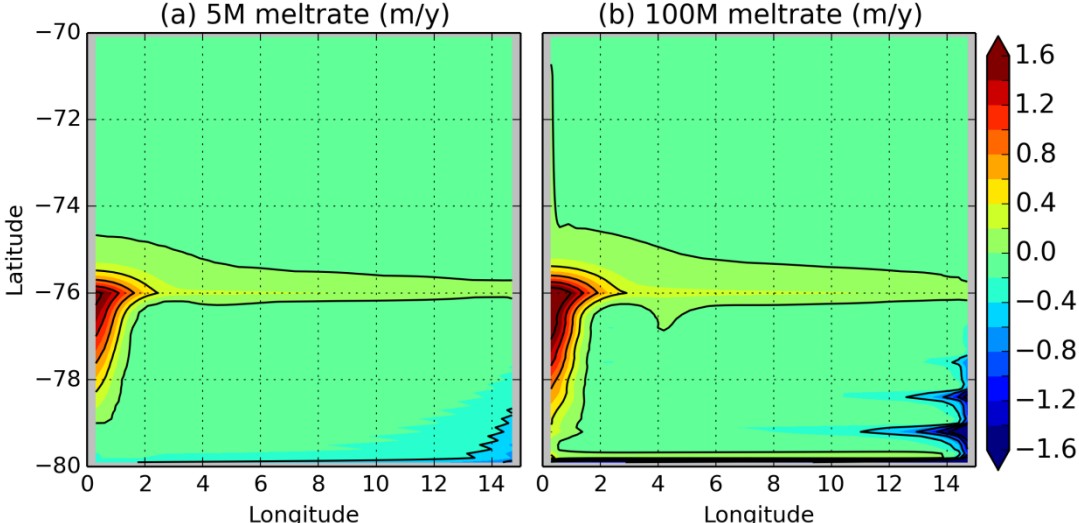

5    **Figure 4: Melt rate in a) the 5M simulation, and b) the 100M simulation in m/y (positive for melting and negative for freezing) with a contour interval of 0.4 m/y.**





**Figure 5: (a) Meridional overturning stream function (MOC) in the A_ISF experiment. Colour background represents the zonal mean temperature (°C) after 10 years of the run. b) Overturning stream function in the A_PAR experiment. Colour background represents the zonal mean temperature difference (°C) with respect to A_ISF experiment (A_PAR-A_ISF). c) as b) but for A_BG03.**

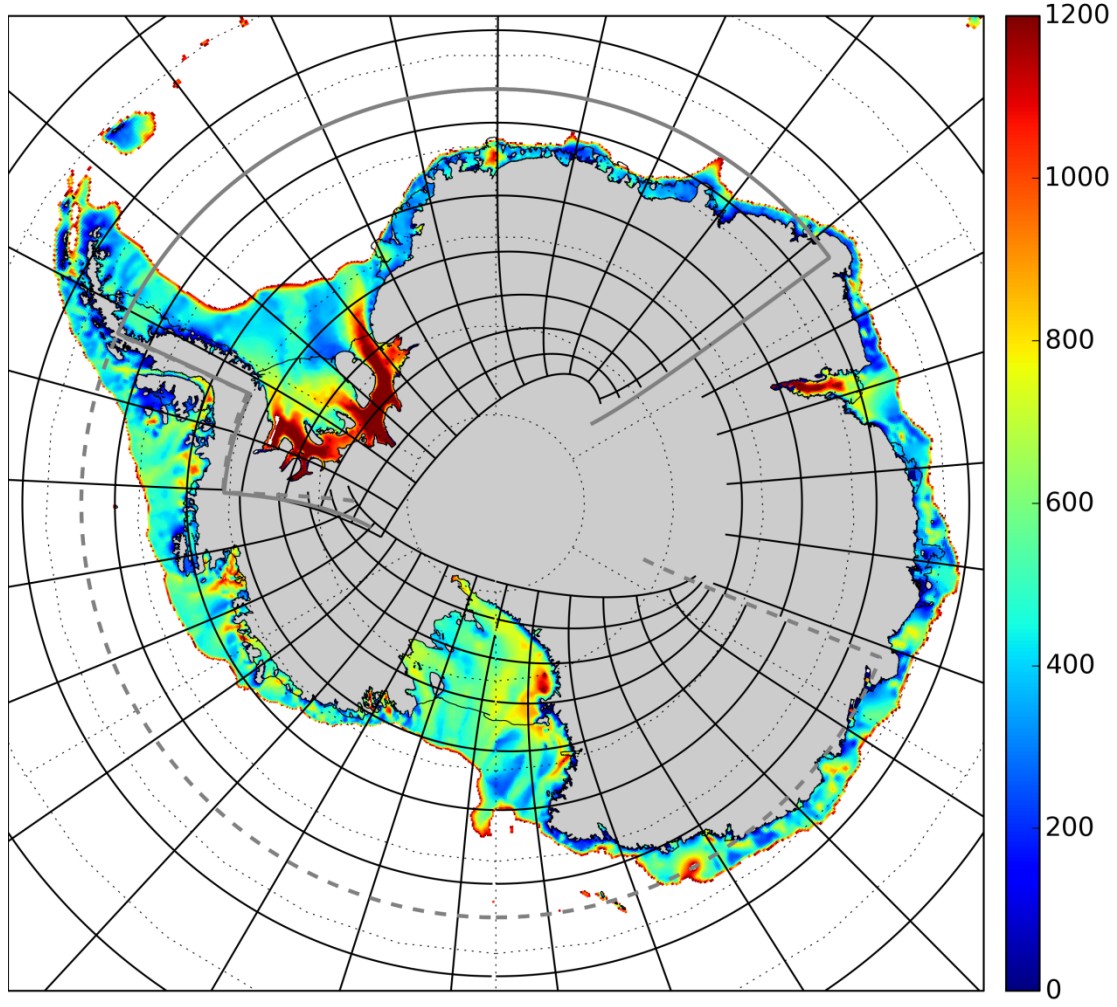

**Figure 6: Bathymetry (m) over the Antarctic continental shelf and beneath the ice shelves. Black lines are the cell edges (plotted every 25 cells). The thick grey line is the limit of the Weddell sector of the grid and the thick dashed grey line is the limit of the Ross, Amudsen and Bellinghausen sector.**





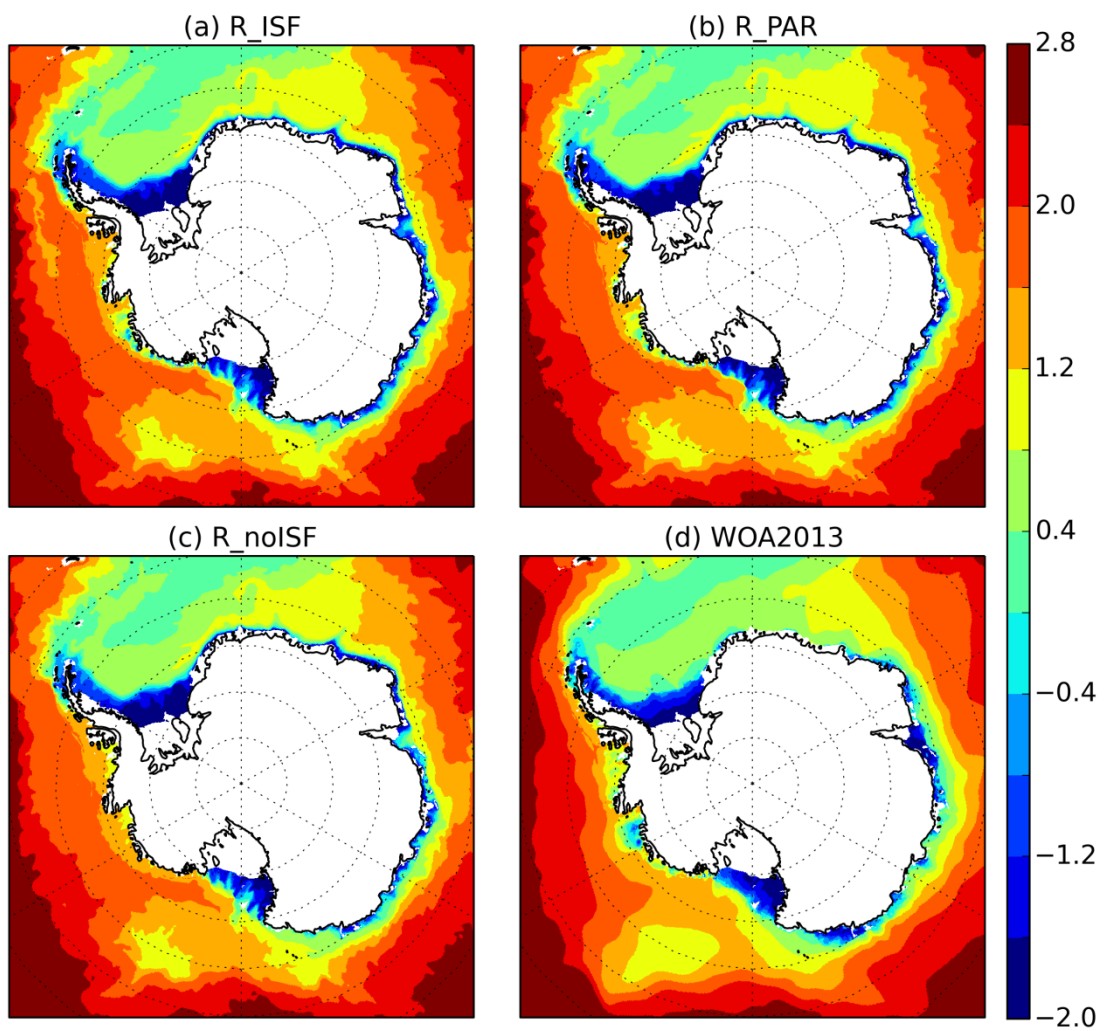

**Figure 7: Temperature (°C) averaged between 300 and 1000m from a) R_ISF, b) R_PAR, c) R_ noISF, d) World Ocean Atlas 2013.**





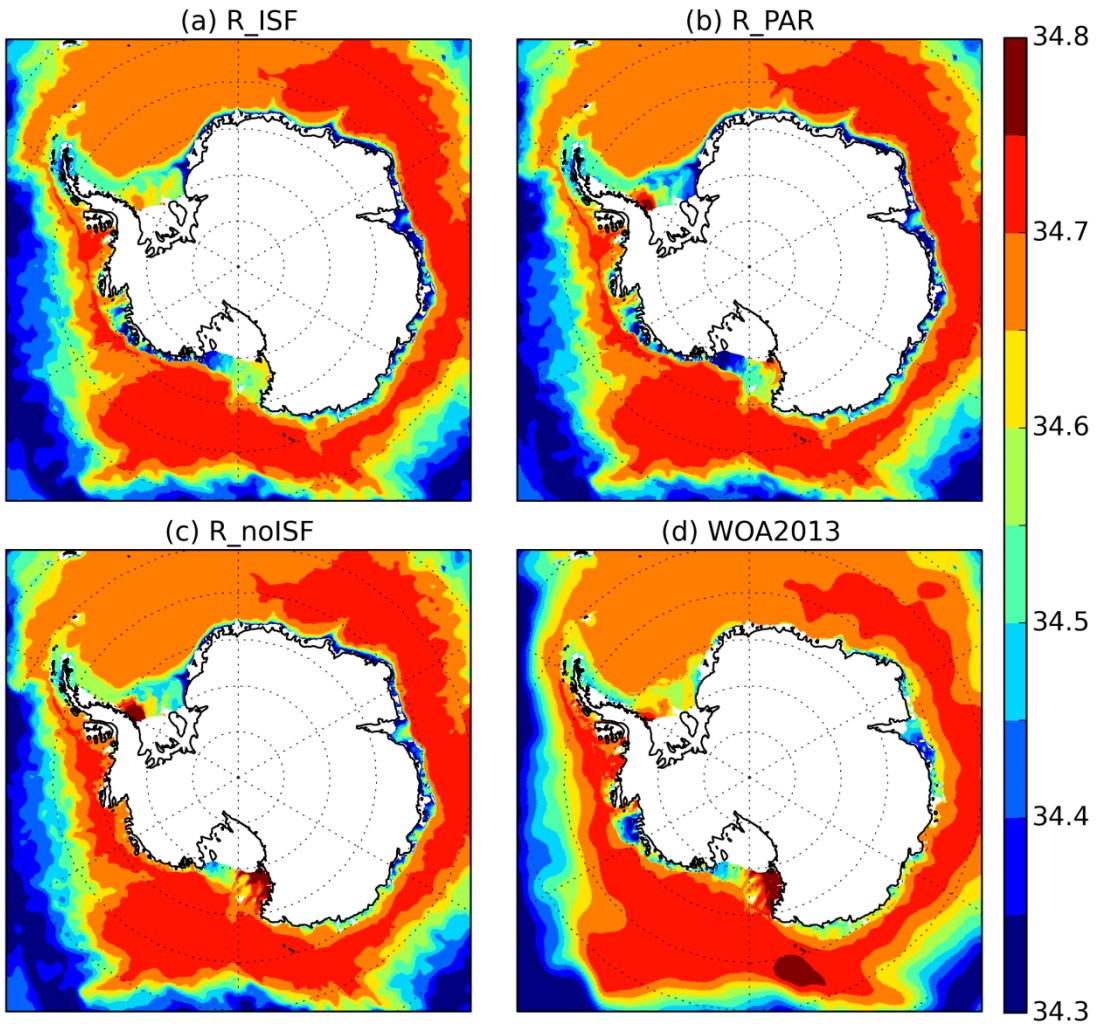

**Figure 8: Salinity (PSU) averaged between 300 and 1000m from a) R_ISF, b) R_PAR, c) R_ noISF, d) World Ocean Atlas 2013 (WOA2013).**





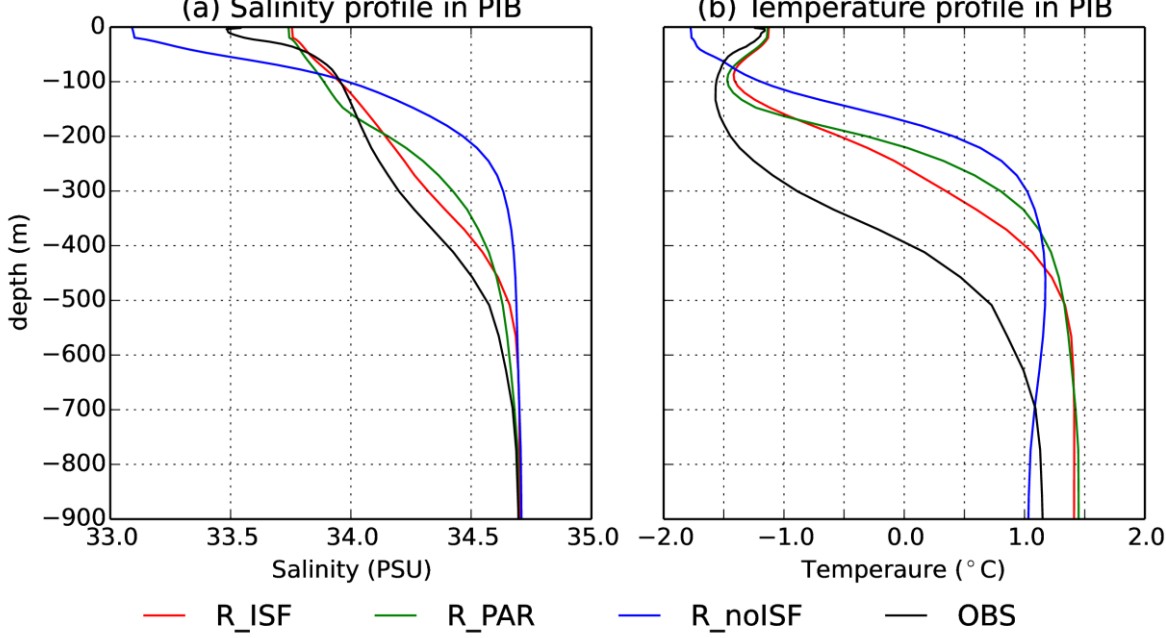

Figure 9: Profiles in Pine Island Bay in R_noISF (blue), R_ISF (red) and R_PAR (green) of a) salinity and b) temperature. Climatology (Dutrieux et al, 2014) is in black.





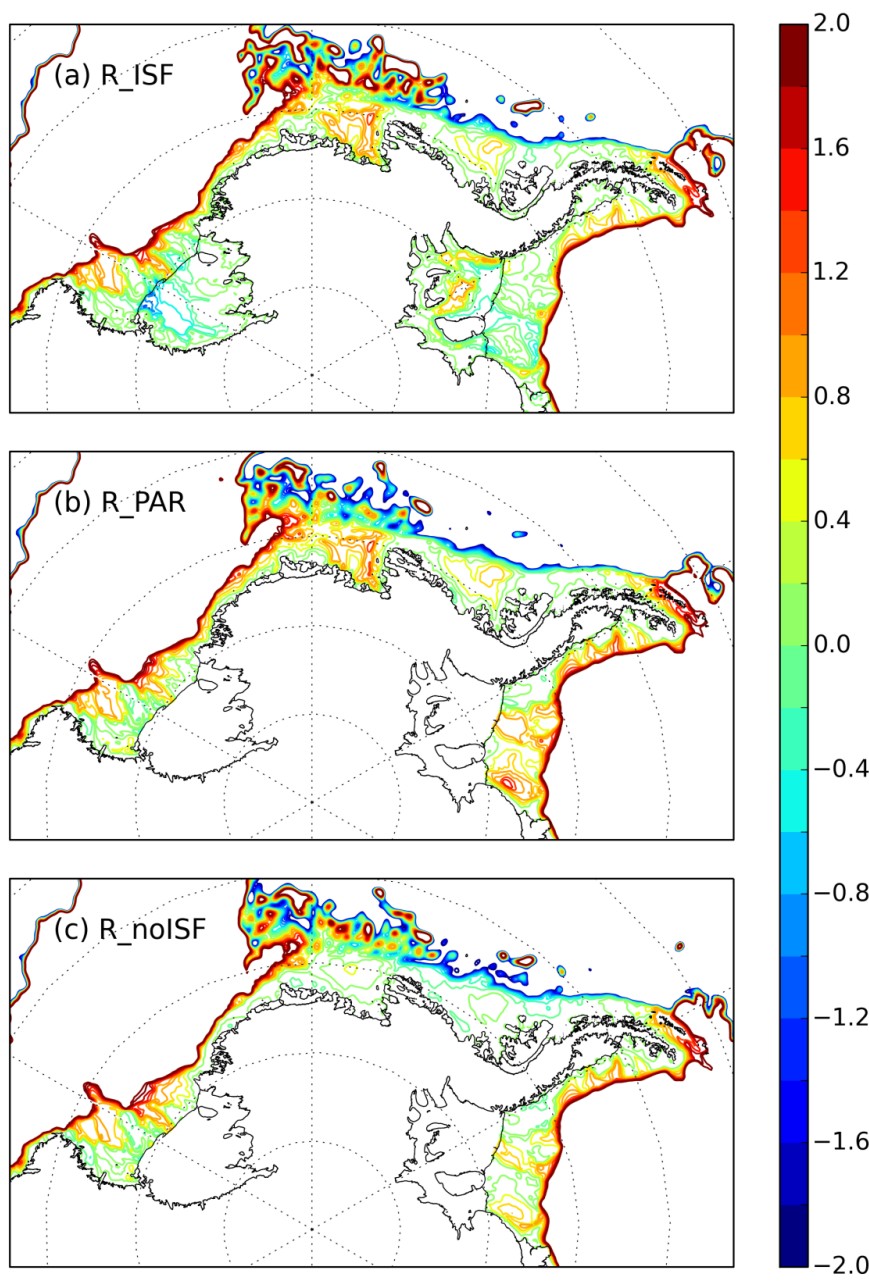

**Figure 10: Barotropic stream function (Sv) on the Ross, Amundsen, Bellingshausen and Weddell continental shelves in a) R_ISF, b) R_PAR and c) R_ noISF. Stream function isolines out of the +/- 2Sv range are not plotted.**





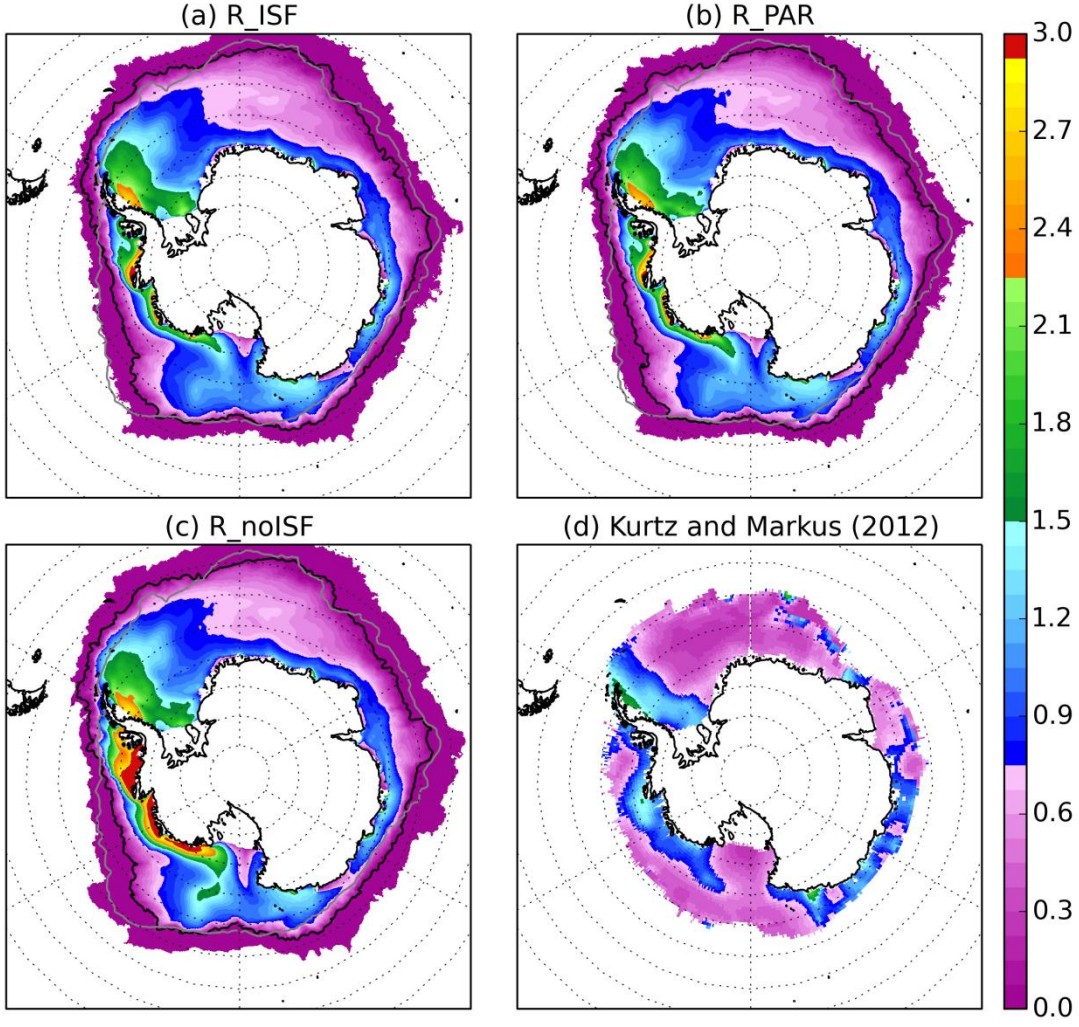

**Figure 11: Mean sea ice thickness (m) from September to November (SON) in colour. Lines represent the sea ice extent (threshold set at 15% ice concentration) in the observations of Comiso (2000) (grey) and the corresponding simulation (black). a) R_ISF, b) R_PAR, c) R_ noISF, d) Kurtz and Markus (2012) data. The observational uncertainty is +/-40 cm.**





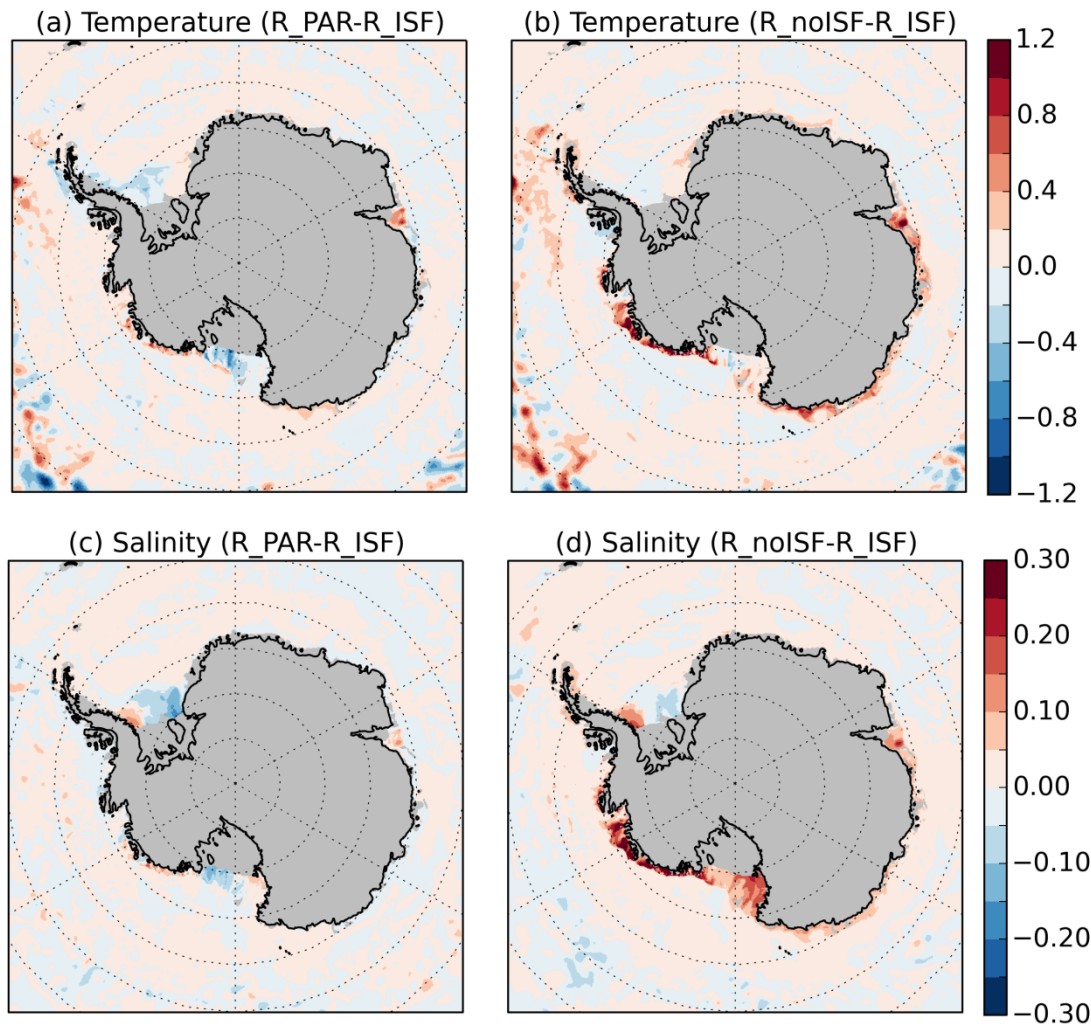

**Figure 12: Map of temperature in °C (a,b) and salinity in PSU (c,d) differences between R_PAR and R_ISF (a,c) and R_noISF and R_ISF (b,d) averaged between 300m and 1000m.**


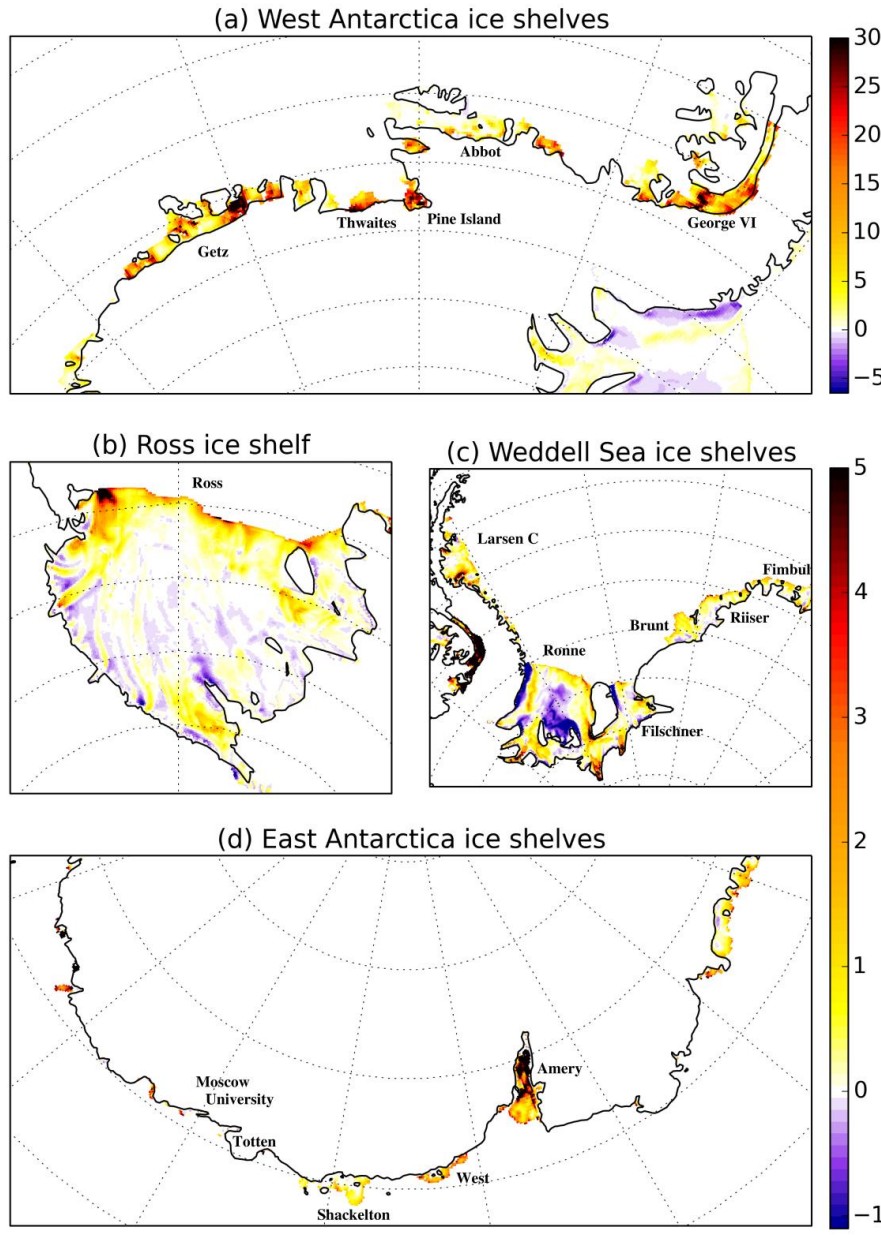

**Figure 13: Ice shelf melting (m/y) in the R_MLT simulation for a) the West Antarctic ice shelves, b) Ross Ice Shelf, c) Filchner-Ronne Ice Shelf and d) the East Antarctic ice shelves. Note that panels (a) and (b,c and d) have different colorbars.**





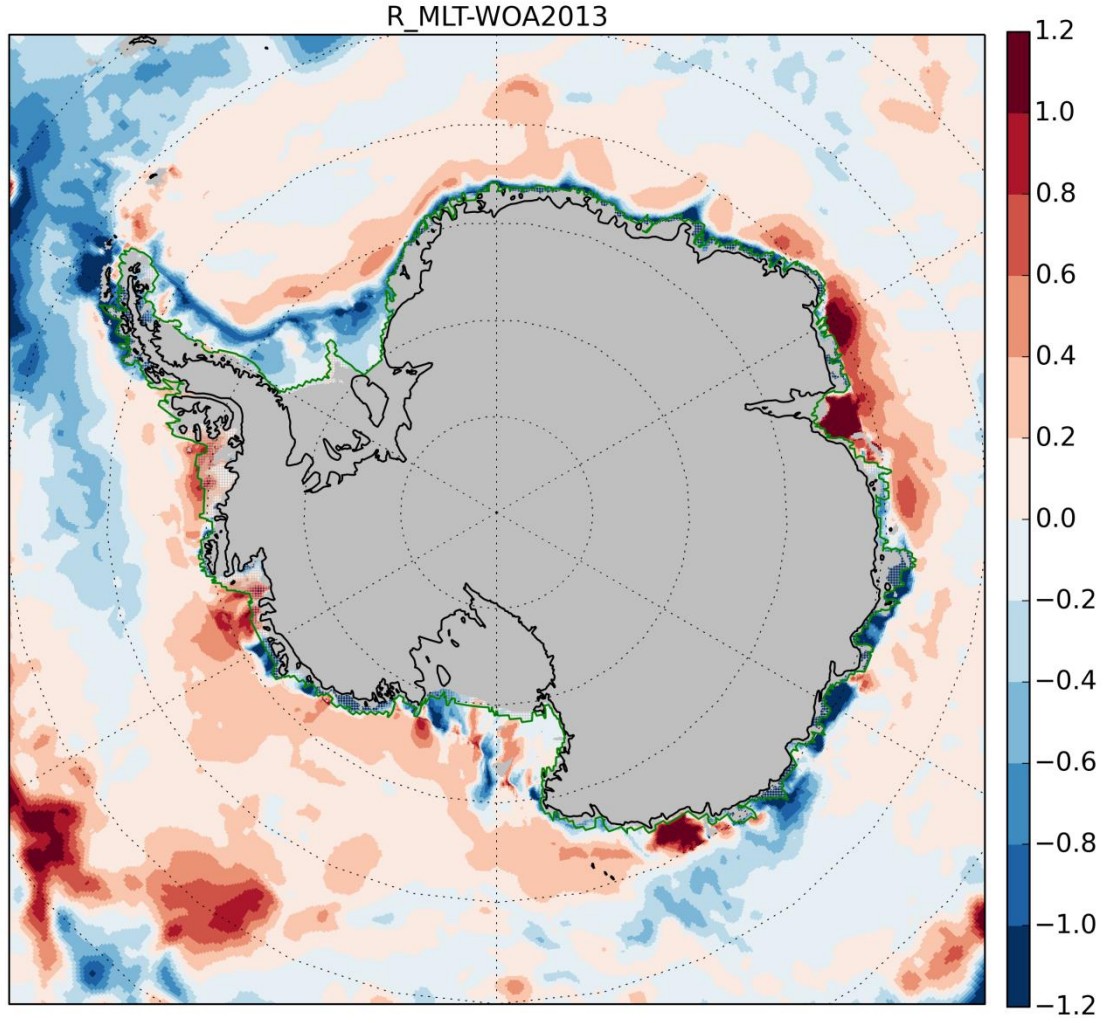

**Figure 14: 300-1000m mean temperature differences between R_MLT and observations from WOA3013. Grey area represents ice sheet, ice shelves or ocean shallower than 300m. The hatched area limited by the green line represents where the observational dataset is obtained by extrapolation.**





**Table 1: Parameters used in the ice shelf/ocean interaction formulation.**

| Symbol | Description | Value | Unit |
|---|---|---|---|
| Cp | Ocean specific heat | 3992 | $J.K^{-1}$ |
| Lf | Ice latent heat of fusion | $3.34 \times 10^5$ | $J.kg^{-1}$ |
| Cpi | Ice specific heat | 2000 | $J.K^{-1}$ |
| Rhoi | Ice density | 920 | $Kg.m^{-3}$ |
| K | Heat diffusivity | $1.54 \times 10^{-6}$ | $m^2.s^{-1}$ |
| Cd | Top drag coefficient | $10^{-3}$ | |
| $\sqrt{C_d}\Gamma_T$ | Thermal Stanton number | $1.1 \times 10^{-3}$ | |
| $\sqrt{C_d}\Gamma_S$ | Diffusion Stanton number | $3.1 \times 10^{-5}$ | |





**Table 2: List of model runs. Expl. means the ice shelf melt rate is explicitly calculated. Presc. means the ice shelf melt rate is prescribed (i.e. independent of ocean temperature and salinity and constant in time).**

| Name | Vertical resolution in the cavity | Losch boundary layer thickness | Melt rate formulation |
|---|---|---|---|
| 5M | 5 m | 5 m | Expl. |
| 10M | 10 m | 10 m | Expl. |
| 30M | 30 m | 30 m | Expl. |
| 60M | 60 m | 30 m | Expl. |
| 100M | 100 m | 30 m | Expl. |
| 150M | 150 m | 30 m | Expl. |
| 31L | 40-240 m | 30 m | Expl. |
| 46L | 40-110 m | 30 m | Expl. |
| 75L | 20-80 m | 30 m | Expl. |
| A_ISF | 30 m (Fig. 1b) | 30 m | Presc. |
| A_PAR | No cavity (Fig. 1d) | N/A | Presc. |
| A_BG03 | No cavity (Fig. 1c) | N/A | Presc. |
| R_ISF | 20-80 m (Fig. 1b) | 30 m | Presc. |
| R_PAR | No cavity (Fig. 1d) | N/A | Presc. |
| R_noISF | No cavity (Fig. 1a) | N/A | Presc. |
| R_MLT | 20-80 m (Fig. 1b) | 30 m | Expl. |





**Table 3: Basal melt in Gt.y⁻¹ for the last year of simulation in R_MLT. Observations come from Rignot et al. (2013). Geometry column indicates the main modification to the BEDMAP2 bathymetry/ice shelf draft as follows: GL means the GL is moved seaward, "shallow" means the ice shelf is too shallow away from the grounding line and "narrow" means the narrowest passage into the cavity is one cell wide. Blue background indicates "cold water ice shelf" and orange background indicates "warm water ice shelf". +/0/- is a summary of the ocean temperature condition at the closest non-extrapolated cell in the WOA2013 observational dataset (Fig. 14).**

| Ice shelf | Model | Obs (Rignot 2013) | Temperature error at the ice shelf edge (observation: WOA2013) | Geometry |
|---|---|---|---|---|
| Amery | 207 | 13 – 59 | ++ | GL |
| West | 26 | 17 – 37 | - | |
| Shackleton | 14 | 58 – 88 | -- | GL |
| Ross | 111 | 14 – 81 | 0 | GL, shallow |
| Larsen C | 46 | -46 – 87 | 0 | |
| FRIS | 123 | 111 – 210 | - | GL |
| Brunt + Riiser | 39 | -6 – 26 | - | shallow |
| Fimbul | 42 | 13 – 43 | - | GL |
| Cold ice shelves | 818 | 531 – 1033 | | |
| Getz | 337 | 131 – 159 | + (east) -- (west) | shallow |
| Thwaites | 74 | 91 – 105 | + | |
| Pine Island | 87 | 93 – 109 | + | |
| Abbot | 52 | 32 – 72 | + | |
| George VI | 298 | 72 – 106 | + | narrow |
| Warm ice shelves | 1142 | 452 – 630 | | |
| Others | 409 | 214 – 425 | | |
| Total | 1865 | 1263 – 1737 | | |