# Peer review of "Explicit representation and parametrised impacts of under ice shelf seas in the z\* coordinate ocean model NEMO 3.6"

_Geoscientific Model Development, 2017_

## Referee Comment (RC1) · Anonymous Referee #1 · 2 Apr 2017

Recommendation: Minor revision

The authors describe the implementation of ice-shelf cavities in the NEMO ocean model, and assess its behaviour in (1) the idealized ISOMIP framework, including sensitivity experiments and comparisons to previous modelling results, and (2) "real ocean" 0.25° simulations around Antarctica, including comparison to observational data and sensitivity analyses. Interestingly, they also present a new way to parameterize the input of ice-shelf melt water into ocean models with no explicit representation of cavities. They show that such parameterization is able to capture the ice-shelf influence on sea-ice thickness and on the ocean circulation over the continental shelf, which sounds promising for coarse climate models.

[Figure]

This is a substantial piece of work that clearly describes the implementation of ice shelves in NEMO, but that is also very useful beyond the NEMO community. The comparisons and sensitivity tests are conducted in a robust way, and the results are generally well presented and discussed. I have a bunch of minor comments that will hopefully contribute to improve the paper, but no major objection, and from my point of view, the paper is already quite good as it is.

——

Minor comments:

- The authors should make clear that their "parameterization" parameterizes the way to distribute ice-shelf melt water and therefore the circulation induced by ice shelves, but does not provide the amount of melt water. It is important to clarify this because readers from the ice-sheet or paleo-climate communities would probably expect an "ice shelf parameterization" to provide melt rates or melt fluxes. This is currently very clear in the conclusion, but maybe not enough in the text, and the title might be misleading.

- In section 2.2.1, it is assumed that the ice shelf is "in hydrostatic equilibrium in water at the reference density isf, taken to be the density of water at a temperature of $-1.9°$C and a salinity of 34.4". Can the authors explain why they make such assumption?

- Section 3.3: what would happen in case of a "Losh TBL" thicker than the vertical resolution? Then, in Fig. 3, the authors show the effect of using 31, 46 and 75 levels based on standard stretching parameters. They conclude that 75 levels might not be enough, but they don't issue any recommendation on how many levels should be used in standard NEMO simulations. Including greater values in Fig. 3 (e.g. L100, L150) would be useful for the community. Finally, these sensitivity results likely depend on the slope of the ISOMIP ice draft, and the authors should probably discuss the generalization of these results.

- The year/time-period represented in the "real ocean application" is not clearly stated.

As far as I understand, the results represent 1985, which is presented as sufficient to complete a 10-year spin-up and to give the first-order response to changes in ice shelf representation. Does it mean that the interannual variability is of secondary importance compared to the sensitivity to the representation of ice shelves? What about the comparison to the ice-shelf melt estimated by Rignot et al. (2013) that is undertaken in section 5.6? How strong is the interannual variability in basal melt, and can we expect melt rates in 1985 to resemble those in the 2000s? I do not expect a perfect match here, but at least, the possible limitations should be stated.

———

Other very minor suggestions & typos:

- Abstract, 5th sentence: "decrease" -> "decreased" or "decrease in".

- Section 2.2.2: expand "ISOMIP".

- Section 2.2.2, after equ. 14, expand "tbl" and mention that it's defined further in the text.

- Tab.1: heat capacities should be in J/kg/K. And it would be better to use the Greek letter for Rhoi (as in the equations).

- Section 2.3, 3rd paragraph: I would replace "equilibrium depth" with something clearer like "floatation depth" if it's what the authors mean.

- Section 2.3, last paragraph: expand "fwf".

- Section 3.1: please add some information about the initial state and T,S restoring if any.

- It would be better to have the labels for the x-axes in Fig.2a,b . Also, (a) and (c) are swapped in the figure caption.

- Section 3.2, about refreezing: is there any frazil formation in the water column?

- Last sentence of section 3.3 (about Fig.3b): another reason could be that overturning and barotropic circulations have physically the same dependence on total melt rates.

- Section 5.1: the authors need to tell a bit more about how tidal mixing data from FES 2012 are used in NEMO, and maybe how it accounts (or not) for the effects of tides on ice shelf melt rates.

- Section 5.1, about "The model is run for 10 years starting in 1976, and the first order response is investigated using output from the last year of the simulation": given that there seems to be interannual variability in these simulations, why analysing only one year? Isn't there "first order" variability at the interannual time scale?

- Last sentence of section 5.2: "results from R_MLT are used to evaluate the 3 equation ice shelf melting formulation in NEMO" -> I think it's not only the 3 equation formulation that is evaluated, but also the bathymetry, the ocean thermal forcing, vertical mixing, etc, etc. Same comment for the first paragraph of section 5.6 (although it is clear in 5.6.3).

- Fig.9: could the difference between Dutrieux et al. (2014) and NEMO simulations come from different periods under consideration?

- Section 5.6.3: a reference to Millan et al. (GRL, 2017) could be included to highlight uncertainties in bathymetry and ice drafts.

- Section 5.6.3: Makinson et al. (GRL 2011) estimate that tides double the net melt rate underneath Filchner–Ronne Ice Shelf.

---

## Referee Comment (RC2) · X. Asay-Davis (Referee) · 3 Apr 2017

Review of Mathiot et al. "Explicit and parametrised representation of under ice shelf seas in a z* coordinate ocean model"

Reviewer: Xylar Asay-Davis

I wish my name to be relayed to the authors, as I do not support the practice of anonymous review.

**General comments:**

This paper discusses the addition of ice-shelf cavities into the NEMO ocean model, and a series of tests used to gain confidence in the model's behavior through idealized simulation results that can be compared with those from other models, to understand sensitivity to certain parameters (notably ocean vertical resolution), and to validate the model against observations in a realistic configuration. The paper also presents a parameterization for prescribing known melt fluxes in the absence of ice shelf cavities, and shows that the parameterization captures many of the features of the flow produced with ice shelf cavities.

Overall, I find that this paper does a very good job at documenting the new NEMO features. The work is important for the field, especially because NEMO's capability to simulate ice shelf cavities is already used by several groups and NEMO is likely to become one of the most widely used models with this capability. The paper is well organized and the experiments are sensible and appropriate, exploring both the potential of the model and making clear some if its limitations and biases.

I recommend a number of minor revisions to the manuscript. If these are addressed, I would recommend the manuscript for publication.

**Specific comments:**

Title: I believe that GMD requires the model name and version to be part of the title for model description papers.

P. 1:
L. 10 I would suggest defining the acronym NEMO here, the first time you use it (other than the title, assuming you follow my previous suggestion)

L. 15 HSSW needs to be defined the first time it is used here.

L. 20 "that has been assessed": It is not clear to me what this phrase means in this context. Perhaps you could replace it with something more specific and meaningful?

L. 28 "very specific": I would suggest a different phrase, like "different from other freshwater sources".  The whole sentence may need to be reworded to avoid too many redundant references to "freshwater".

P. 2:
L. 1-2 Does the freshwater forcing from melting sea ice also deserve to be mentioned in this context?

L. 11 "spread": To me, this verb implies that water masses arrive at the continental shelf in specific locations and spread out from there.  For some water masses like CDW, this may be correct but it strikes me as strange to think of all water masses spreading across the continental shelf in this way.

L. 13 "towards the surface" I would be careful about this phrase because you make a point that it is important to correctly model the fact that melt water doesn't necessarily reach the surface.  Perhaps just saying "upward" instead of "towards the surface" would avoid implying that the meltwater reaches the surface.

L. 17 "...a temperature close to that of the surface freezing point..." should be "...a temperature close to the surface freezing point...", since the freezing point *is* a temperature.

L. 17-22 This sentence is *way* too long.  I would suggest one sentence for each mode.

L. 22-23 "The process is usually referred to as the ice pump." Because you've just mentioned refreezing in the previous sentence, it seems like the refreezing process is referred to as the ice pump.  I would suggest clarifying which process it is that you are referring to.

P. 3:
L. 8-9 "So whatever the resolution, some cavities remain unresolved." This is clearly not true, since 1 km resolution (for example) should be sufficient to resolve even Ferrigno ice shelf.  In the remainder of the manuscript, you are careful to state that some ice shelves will remain unresolved at any *practical* resolution for global modeling.  Please include those caveats here, too.  Keep in mind that what seems impractical may become practical in the coming years to decades.

L. 11-16 Please include section numbers as part of describing the structure.  This helps the reader to better navigate the paper based on this outline.

L. 19 The definition of the NEMO acronym should go earlier (in the abstract).  The version number should go in the title, meaning it may not need to be mentioned again here.

L. 20 Please explain what an Arakawa C-grid is.  It also seems awkward to mention the "nonlinear filtered free surface option" without giving some sort of explanation of what this is and why it is the appropriate choice for this work.

L. 26 Maybe z* needs to be explained first, at least qualitatively?

L. 26 "can be used with sea ice model" should probably be "can be used, unmodified, with a sea ice model". (Note also the missing "a" in front of "sea ice model" After all, you are saying that z* can also be used with an ice shelf but it requires modification.

L. 30 What about shallow regions covered in thick sea ice? Is there a provision in NEMO for preventing sea ice thickness from becoming of the same order as the ocean column thickness?

P. 4
Eqs. (1)-(4) Many problems here:
- The indexing here isn't entirely clear. Either a diagram or further explanation in the text would be helpful. k=1 would appear to be the surface, but this isn't explicitly stated. kz=1 would seem to be the first level.
- In (2), the upper limit on the sum should be k-1, not $k_{isf}$-1, I think
- $Z_w$ appears to be an inconsistent mix of positive up in (1) (assuming eta is positive up, which (4) seems to imply) and positive down in (2) and (3). You use a negative sign for depth in most subsequent figures and text, so I think (2) and (3) need a minus sign
- It seems like eta needs to be added to (2) and (3) for them to be consistent with (1) and (4). That is, If I plug in k=1 into (2), I would expect to get (1). If I plug in $k_{max}$ into (3), I would expect to hit the sea floor, $Z_w(k_{max}) = -H_{isf} - H$.
- It would be helpful to have H be written explicitly as a sum of $dz_{0,T}$

Eqs. (5)-(6) It might be relevant to indicate how these equations are discretized in the model. For example, presumably density is piecewise constant in layers and pressure gradients are explicitly substituted with density gradients?

L. 32 "The same limitation is expected…" You just said that partial cells compare favorably to sigma coordinate. You mentioned that bottom topography can be "challenging" but you haven't mentioned any specific "limitation" that should apply along the ice shelf base.

P. 5:
L. 7 "it is sometime necessary" should probably be "it would sometimes be necessary" since you state right after this sentence that you don't do this.

L. 8-10 You talk about the importance of parameterizing melt from ice shelves that are too small to resolve otherwise. You indicate later on that significant regions close to grounding lines and even most of some ice shelves are removed they are too steep and/or too thin to be easily represented in a z* coordinate without significant "digging" into bathymetry and/or ice draft. Might this not be having a comparable effect to the unresolved ice shelves and require some alternative treatment or parameterization? In many cases, the largest melt rates are at or near the grounding line. By moving the grounding line, particularly by moving it systematically toward the ocean, don't you think you are biasing the model toward lower

total melt (perhaps offset by other biases to produce the higher total melt you find in the end)?

L. 13-14 It would be nice to have a more mathematical description of what is meant by "correction" here.

L. 23 You don't explain until the middle of the next page how $T_w$ is computed from model temperature. You have also not mentioned anything about a boundary layer so far. It might be worth saying something like "$T_w$ the temperature averaged over a boundary layer below the ice shelf (explained below),"

L. 36 "ISOMIP formulation". You should define the ISOMIP acronym here and explain why the formulation is named this (since ISOMIP has not been mentioned previously).

L. 27 "Jenkins et al. (2001)" There are 2 formulations for the heat and freshwater fluxes in this paper. Could you point to the specific equations in the paper you used? I presume you use (24) from that paper, not (25) because the volume flux of freshwater at the surface is explicitly modeled.

Eq. (10):
- There is no Eq. (9), so you need to renumber.
- What about vertical advection, a la Holland and Jenkins (1999). Advection typically dominates diffusion except where melt/freezing rates are very small in magnitude.

P. 6:
Eq. (12) Units are needed. It would be cleaner to define liquidus coefficients with symbols and put them in Table 1.

L. 28-29 "averaged over the first cell and that part of the second wet cell required to make up the constant boundary layer thickness" You never need a third cell to get to 30 m? You don't allow the user to set a boundary layer thickness that is significantly larger than the resolution (e.g 30 m for 5 m resolution)? What would happen in your variable resolution case if you had a shallow ice shelf so that dz < 25 m but you specified a 30-m boundary layer thickness? Is the BL thickness never allowed to be thicker than the local resolution of a full cell?

P. 7:
L. 1-2 This could use some clarifying, especially for the non-constant layer thicknesses with variable numbers of levels. How are the layer thicknesses determined? What is the range of values?

L. 4-6: This point has been made almost word for word in the intro. I would suggest shortening or removing this sentence.

L. 9: "Figure 1c" should be "Fig. 1c". It's slightly odd to mention only this panel of the figure and get to the others only much later in 5.2. Maybe you could mention here that you will explain the remaining panels in Fig. 1 in Sect. 5.2.

L. 12-13: "spread the freshwater due to ice shelf melting evenly between the grounding line depth and the depth of the calving front" This maybe deserves a bit more emphasis and explanation. One might think the plume-like structure of melting would be more consistent with melt fluxes exiting closer to the surface. My ISOMIP+ simulations, for example, and those of most other models suggest that T and S are not strongly affected by melting except close to the interface. Why, then, is a uniformly distributed flux (both horizontally and vertically) the preferred choice? Is this for simplicity and because results are reasonable, or is there a deeper physical reason? Comparison with observations near PIG in Fig. 9 suggest that neither the explicit modeling nor the parameterization is mixing deep enough. This suggests maybe not enough entrainment when explicitly modeling the ice shelf cavity, and may suggest that more melt flux should actually go in at depth than closer to the surface.

L. 26 ISOMIP was never defined so should be defined either here or above when you speak of the "ISOMIP formulation".

L. 26 Also, you should mention that you are performing ISOMIP experiment 1.01 (there were 3 experiments defined).

P. 8:
L. 6-7 "at which time the system is in quasi-steady state". Two things: first, to me "quasi-steady state" is used for systems that oscillate (possibly chaotically) around a steady state, whereas this system has enough viscosity to relax toward what I would expect is a true steady state. Second, I have found in my own simulations that even 30 years isn't really enough time to be that close to steady state. Since NEMO's behavior is typically similar to POP2x, the mode I use, I would expect that the same is true for you. Do you have reason to believe that it *is* close to steady state (e.g. you ran another 5 or 10 years without much appreciable change)?

L. 31 Each ISOMIP experiment named in Table 2 needs to be explained, either in the caption or in the text. Also, adding a prefix like "I_" for ISOMIP would make the division between the different experiment categories clearer and would be more consistent with other experiment names in the table.

L. 31 Also, why no experiment with (say) a 5-m vertical res. but a 30-m TBL? Is this not supported? Such an experiment would better demonstrate whether the model converges with increasing vertical resolution. In my experience with POP2x, it does. This means that, if we want a boundary layer to be present, we should use a physical length scale to determine its depth (like KPP does, for example), rather than tying it to the vertical resolution. More discussion of this point below.

P. 9

L. 9 "coarsest resolution in the cavity seems to determine the total melt." I think this needs some more discussion. This is likely because the coarsest resolution corresponds to the deepest part of the ice shelf where melt rates are typically highest (thus having the greatest effect on the total flux), right?

L. 11-12 This is almost identical to text in 5.1, so maybe trim one or the other (probably trim here and refer to that section). Also, there you say that the maximum resolution is 150 m, which is very different from 40 m.

L. 26 This geometry is the one for ISOMIP expt. 2.01. Maybe change to "The geometry is the same as ISOMP expt. 2.01, which is the geometry from ISOMIP expt. 1.01 except …"

L. 27 Maybe give a specific equation and/or figure number from Asay-Davis et al. (2106)?

L. 29-30 The viscosity used in ISOMIP is already quite large compared to what we use in realistic simulations with similar resolution. It seems like increasing this value by another factor of 5 renders any comparisons with a realistic configuration nearly impossible. Also, can you talk about the cause of the noise at the ice shelf front? That sounds troubling and viscosity may not have been the best way to handle it. Why didn't similar noise show up in the realistic simulations?

P. 10
L. 21 "so the behaviour described above may differ in a realistic configuration". I'm not sure which "behaviour described above" is being referred to here for sure. Do you mean that a good comparison in an idealized context doesn't necessarily imply good behavior under realistic conditions?

L. 28-30 This is a brilliant solution for coarsening the grid resolution close to the South Pole!

P. 11
L. 1-2 As mentioned, this is almost identical to text in 3.3, where you claim instead a vertical resolution as coarse as 40 m in cavities. Please trim the text in 3.3 and refer to here, making sure the sections are consistent with each other.

L. 3-5 How did you blend these data sets together?

L. 5 It might be worth mentioning the strange choice made in Sect. 5.2 of Fretwell et al. (2013) for ice shelf cavities with poorly sampled bathymetry, since you state later on that you had trouble with many of these ice shelves. Their choice might have been appropriate for ice-sheet modeling but it has the effect of making the bathymetry closely follow the ice draft with a very thin water column between them in many places. The resulting ocean circulation is likely completely false because the cavity geometry is essentially nonsensical. Becaue of this, many of us have resorted back to RTOPO1 or adopted newer gravity-inversion data sets for the ice shelves where this technique was applied.

> "We tested for areas where ice-shelf thickness and sub-shelf bathymetry falsely indicated grounded ice, and where necessary, enforced flota- tion by lowering the

(poorly sampled) sea bed. We did this by interpolating the thickness of the sub-ice-shelf water column between the point where cavity thickness declined to 100m and the grounding line where cavity thickness is 0 m. This approach was required for Getz, Venable, Stange, Nivlisen, Shackleton, Totten and Moscow University ice shelves, for some of the thickest areas of the Filchner, Ronne, Ross, Amery ice shelves and for the ice shelves of Dronning Maud Land."

L. 9 "Moscow University" There seems to be confusion in the literature about which is Dalton and which is Moscow University.  The Australian groups whose research seems to be most focused on these shelves prefer to call the larger 2 shelves Totten and Dalton, with Moscow University as the small shelf between the 2.  Rignot et al. instead use Moscow University to refer to Dalton.  While I don't know for sure who is correct, I would tend to defer to the Australians and call this Dalton.

L. 10-11. This was covered above and can be removed.

P. 13
L. 11 "associated with" would maybe be more correct as "in addition to".  To my knowledge, Nakayama et al. (2014) attributes the fresher coastal current to sub-ice-shelf melting and does not draw any direct causal connection to weak winds in the atmospheric forcing.

P. 14
L. 25 "the deficiency in representing the giant ice shelves..." It is not clear from the context here which deficiencies you mean.  You presumably mean the lack of horizontal circulation due to not explicitly representing the ice shelf cavities.

L. 31 I would change "Nevertheless, current coarse resolution..." to "This may not be a significant problem because current coarse resolution..."

P. 15.
L. 30-31.  Any idea why the water on the continental shelf has such a large warm bias here?

P. 16
L. 4-5 "integrated melt rate" Elsewhere, you use  "total melt" for this concept, so maybe here as well.

L. 11 In the case of Getz, might the problem be the bad BEDMAP2 bathymetry?  George VI is more complicated but the BAS observations (which you could cite here -- Kimura and Venerable papers come to mind -- contact me if you don't know which ones I mean) show stairstep stratification that is likely poorly represented by the boundary-layer formulation assumed in the 3 equations and related heat and freshwater fluxes.

L. 13-15 These studies used RTOPO1 bathymetry for Getz.  They may have reasonable melt rates at George VI for the wrong reasons (e.g. cold water masses than observed or poor circulation).

L. 22 FESOM uses a sigma coordinate only near continental margins. In the deep ocean, it is a z-level model. Maybe state this as "while FESOM uses a sigma-coordinate around the Antarctic continental margin."

P. 17
L. 5-7 The bathymetry is not extrapolated from the surrounding region. Instead, the cavity thickness is extrapolated. This leads to ridiculously thin cavities in many, many places. The ice draft and the (completely made up) bathymetry may vary in tandem in the vertical over many ocean thicknesses, maintaining a thin ocean cavity between them. Nothing like this happens in any of the sub-ice-shelf cavities where observations are available, so (to beat a dead horse) this choice of interpolation was not appropriate for ocean modeling applications.

L. 15 I'm not sure what is meant by "the friction law directly". I would take out the word "directly" or replace it with a clearer explanation of what, besides the friction coefficient, is meant here.

L. 17 "is very sensitive" I would recommend against using subjective phrases like "very sensitive" if you can be more quantitative. Maybe just drop "very".

P. 18
L. 2 "very sensitive" again, I would drop "very" (or be more quantitative).

L. 2-5 You imply that the finer resolution solution is the more realistic but this assumes that the true boundary layer is correctly represented at high vertical resolution. It is not clear to me that this is the case, at least for realistic configurations. Unless the vertical viscosity and diffusivity (or another parameterization of turbulent mixing) are being adapted in such a way as to correctly represent the physics of turbulent mixing below the ice shelf, the finer resolution solution may markedly underestimate mixing and entrainment. Indeed, your Fig. 9 seems to suggest that this might be the case in NEMO (though processes outside of ice shelf cavities may also be responsible for the biases, of course).

L. 24 Once again, maybe replace "very dependent" with something more quantitative.

P. 19
L. 12 "prescribe the melt rate" might be clearer if it were replaced with "distribute the melt fluxes". It is unclear if "prescribe the melt rate" refers to computing is or distributing it, and you *don't* explain how to compute the melt rate.

L. 31-32 While these processes might very well be important elsewhere (e.g. Greenland), it's not clear that melting on ice faces can be a first-order effect in Antarctica. The areas of calving faces are so small compared with ice-shelf bases that the melt rates would need to be many orders of magnitude higher than those at the base of the ice for them to play a significant role in ice loss. Furthermore, melting at calving faces can indirectly be accounted for in the calving flux of icebergs. So I don't see these effects being of primary importance for coupled ice sheet/ocean modeling in the Antarctic.

P. 24
L. 6-7 It would be good to provide a URL, since this is not a journal article. Unfortunately, I am not aware of a working link so you may need to email Ben Galton-Fenzi to get him to put it somewhere permanent (like the other ISOMIP link I mentioned above).

Fig. 2: Sign of panel c) is wrong.

Fig. 4: Sign of both panels is wrong.

Fig. 5: Sign of depth is wrong (should be negative to match other plots). Titles of b) and c) are a bit misleading because the difference only applies to the temperature (colormap) not the overturning. In the caption, I would explicitly state that the MOC is in contours. As it is, it seems like you assume the reader will notice the MOC first (and that this is the primary piece of information being shown) and that the temperature is secondary. For me, the opposite was true: I noticed the temperature first.

Fig. 6: Nice figure!

Figs. 7-8: Maybe remind the reader that the model data is averaged over simulation year 10. Add citations for WOA. (I think they might be different for PT and for S.)

Fig. 13: White was not the best color choice for zero melting because it is hard to tell the difference between absence of ice shelves and presence but with zero melting. This figure is the only one zoomed in enough to give us a sense of how well resolved the smaller ice shelves are. I would suggest using light gray either from zero melting or for the background of each panel so the two can be distinguished (with slight preference for the latter).

**Typographic and grammatical corrections:**

Line numbering: For future manuscripts, it would be more helpful if line numbering continues through the whole manuscript (as in Latex) rather than being for each page. This makes the review process easier.

P. 1:
L. 13 comma needed after "at the surface)"

L. 16 "...under ice shelf seas overturning circulation…" This is an awkward phrase. Might I recommend, "...overturning circulation under ice shelves…"?

L. 17 comma missing after "at the surface"

L. 17-18 "It yields similar improvements… than the explicit…" In this sentence, "than" should be replace with something like "to those from" (i.e. "similar to", rather than "similar than").

L. 19 "widely used" does not need a hyphen; "3 equations" should be "3 equation" or possibly "3-equation"

P. 2:
L. 14 "...inflowing water mass that could..." should be "...inflowing water mass, which could..."

L. 18 no comma needed after "high"

L. 18 This is kind of picky, I know, but I would change "...meling can be high..." to "...melt rates can be high…", since melting is kind of a state of being that, to me at least, isn't really high or low.

L. 30-33 "Furthermore" seems to imply that the second of these two sentences follows from the first, but they are not really related. I would suggest changing the second sentence to something like, "Global conservation is also an important issue, as the ocean/sea-ice model is used as a component within Earth System Models."

L. 32 comma needed after "this issue"

L. 33-34 I would change "the z* vertical coordinate" to "a z* vertical coordinate".

P. 3:
L. 20 "nonlinear" does not need a hyphen

P. 4:
L. 14 "the z axis in" should be "the z axis of"

L. 31 "sigma coordinates models" should be "sigma coordinate models"

P. 5:
L. 20 I would much prefer TBL to tbl (and FWF to fwf later on). It is much easier to read and to spot the definition if you encounter the acronym later on and need to be reminded what it stood for.

L. 23 Here and *many* other places, you use a period to add spacing to your units. In Latex, the correct way to do this is with a half-space (\,). I suspect this manuscript was written in Word, so I don't know how a half-space is achieved and would recommend a full space instead. In any case, a period is not correct.

P. 6:
L. 8 "($Q_{h)}$" the second parenthesis should not be subscript.

L. 12 "tbl" again better as "TBL"

L. 13-14 "(Jenkins et al., 2010)" should be "Jenkins et al. (2010)"

L. 17 move "(Jenkins et al., 2010)" (no comma) to after "their values" and change "are based on" to "is based on". Should now read: "Furthermore, uncertainties in the Stanton numbers are also large, as the study used to determine their values (Jenkins et al., 2010) is based on data from a single borehole."

L. 18 "Eq. 7-12" should probably be "Eqs. 7-12" and "Eq. (10) to (12)" should be "Eqs. 10-12".

L. 25 "smallest" should be "thinnest" (or "thickest" should be "largest") for consistency.

P. 7:
L. 10 comma missing after "...parameterisation is that"

L. 12 "fresh water" should be "freshwater"

L. 16 I would prefer "FWF" to "fwf". Please define the acronym FWF here.

L. 17 Use "BG03" instead of the full citation, since you took the trouble to define a shorthand.

P. 8:
L. 10-11 In my experience, URLs are most cleanly done as footnotes. They could also be done as citations, in which case you need an author and the last date they were accessed. Also, John Hunter's website is now down. I had asked Ben Galton-Fenzi to post it on a more permanent place. That place ended up being Ben's staff website, which also now seems to have gone down. I would suggest contacting Ben to get this website posted somewhere permanent (once again!).

L. 11 "(Asay-Davis, 2013)" this citation isn't in the bibliography. Is this my EGU presentation?

L. 19 Here and elsewhere, "m/y" should probably be "m y$^{-1}$" or "m a$^{-1}$".

L. 21 "Figure 2" should be "Fig. 2". Also, you are showing melt rates but you have never explicitly said how melt rates are computed from q. Presumably they are in m a$^{-1}$ of freshwater and are positive for melting (as stated in the figure caption). In this case, the field plotted in Fig. 2 needs to be multiplied by -1 (i.e. you're plotting positive freezing).

L. 21 "similar to the one" would be slightly better as "similar to that"

L. 25 "Losch" should probably be "L08"

L. 29 "top boundary layer" could be "TBL"

L. 30 "9 simulations" should, I think, be "nine simulations". The rule I learned was to write out numbers ten or smaller.

P. 12:
L. 20 and 22 "Fig. 7-8" and "Fig. 7, 8 and 10" should be "Figs. 7-8" and "Figs. 7, 8 and 10"

P. 13
L. 1 "Figure 7-8" should be "Figs. 7-8".

L. 8 "10y" should be written out as "ten years"

L. 17-19 Consider reorganizing for clarity: "The position of the ice edge, being too far south in the Amundsen Sea and too far north in the Weddell Sea and around East Antarctica in both simulation, is not changed significantly by the presence of ice shelf cavities (Fig. 11)." When I first read this, I thought the presence or absence of ice shelves was related to the location of the ice edge, whereas you want to point out that these biases exist regardless.

L. 25-26 "Sea ice is thus thinner in R_ISF than in R_noISF…" This was already stated above.

L. 30 "as the impact…" should be "as is the impact…"

P. 14
L. 12 "similar in both" should be "similar between" ("similar" implies a relationship between two things, not a property of both things.)

L. 18 More grammatical would be "In R_PAR, this is due to the lack of a HSSW circulation..."

L. 32: "1° x cos (latitude)" This is some strange formatting with a mix of math and text as well as notation that is not very standard. Maybe "a nominal resolution of $1° \cos(\theta)$, where $\theta$ is the latitude, which is sufficient to..."

P. 15
L. 26 No spaces in "(51-260 Gt y$^{-1}$)".

P. 17
L. 3 "our model setup as the large…" should probably be "our model setup as well as the large…"

P. 18
L. 16 "observed" I would opt of another word like "seen" because "observed" seems to imply "observations" to me, which is not your intent here.

P. 19
L. 13 "physically sensible" does not need a hyphen.

Table 1: many symbols have not been properly subscripted (Cp, Lf, Cpi, Rhoi, Cd).  Rhoi should be the Greek symbol rho, right?  Remove dots in the units.

Table 2: Many of the ISOMIP experiments are not explicitly mentioned in the text.  The names need to be explained either here in the caption or in the text, particularly 31L, 46L and 75L.

Table 3: In the caption, it would be good if you could define ++/+/0/-/-- more quantitatively.  Otherwise, this seems rather subjective.  Regarding the table itself, I don't think GMD is likely to let you format the table the way you have it here (see instructions for authors).  Specifically, they are unlikely to support color like this.  You do have some control over horizontal lines, and this may be the best way to differentiate the different regions.  I don't think you explicitly discuss the last two columns of the table in the text, which it seems like you should.

---

## Short Comment (SC1) · 4 Apr 2017

Dear authors,

in my role as Executive editor of GMD, I would like to bring to your attention our Editorial version 1.1:

http://www.geosci-model-dev.net/8/3487/2015/gmd-8-3487-2015.html

This highlights some requirements of papers published in GMD, which is also available on the GMD website in the 'Manuscript Types' section:

http://www.geoscientific-model-development.net/submission/manuscript_types.html

In particular, please note that for your paper, the following requirements have not been met in the Discussions paper:

- "The main paper must give the model name and version number (or other unique identifier) in the title."

- "If the model development relates to a single model then the model name and the version number must be included in the title of the paper. If the main intention of an article is to make a general (i.e. model independent) statement about the usefulness of a new development, but the usefulness is shown with the help of one specific model, the model name and version number must be stated in the title. The title could have a form such as, "Title outlining amazing generic advance: a case study with Model XXX (version Y)"."

Please add a reference to the NEMO model including a version number in the title of your article in your revised submission to GMD.

Yours,

Astrid Kerkweg
* * *

---

## Author Comment (AC1) · 30 May 2017

Thank you for the clarification. The title has been changed to Explicit representation and parametrised impacts of under ice shelf seas in the $z^*$ coordinate ocean model NEMO 3.6.

---

## Author Comment (AC2) · 30 May 2017

**Review of Mathiot et al. "Explicit and parametrised representation of under ice shelf seas in a z ∗ coordinate ocean model"**

***Reviewer: Anonymous Referee #1***

**Recommendation: Minor revision**

**General comments:**

The authors describe the implementation of ice-shelf cavities in the NEMO ocean model, and assess its behaviour in (1) the idealized ISOMIP framework, including sensitivity experiments and comparisons to previous modelling results, and (2) "real ocean" 0.25 ◦ simulations around Antarctica, including comparison to observational data and sensitivity analyses. Interestingly, they also present a new way to parameterize the input of ice-shelf melt water into ocean models with no explicit representation of cavities. They show that such parameterization is able to capture the ice-shelf influence on sea-ice thickness and on the ocean circulation over the continental shelf, which sounds promising for coarse climate models. This is a substantial piece of work that clearly describes the implementation of ice shelves in NEMO, but that is also very useful beyond the NEMO community. The comparisons and sensitivity tests are conducted in a robust way, and the results are generally well presented and discussed. I have a bunch of minor comments that will hopefully contribute to improve the paper, but no major objection, and from my point of view, the paper is already quite good as it is.

**Authors:** We would like to thank you for the very constructive, positive and encouraging comments. You will find below a reply to each point in *italic blue* with, when necessary, the new text between quote.

**Minor comments:**

- The authors should make clear that their "parameterization" parameterizes the way to distribute ice-shelf melt water and therefore the circulation induced by ice shelves, but does not provide the amount of melt water. It is important to clarify this because readers from the ice-sheet or paleo-climate communities would probably expect an "ice shelf parameterization" to provide melt rates or melt fluxes. This is currently very clear in the conclusion, but maybe not enough in the text, and the title might be misleading.

*About your point on the misleading title, the title has been changed to mentioned we parametrised the impact of under ice shelf seas. The second reviewer and the editor mentioned the text should include the model name and version. The new title is "Explicit representation and parametrised impacts of under ice shelf seas in the z\* coordinate ocean model NEMO 3.6".*

*We do not modify the rest of the text as we think it is quite clear in the text as it is:*

*Abstract:" Mimicking the overturning circulation under the ice shelves by introducing the meltwater over the depth range of the ice shelf base, rather than at the surface, is also assessed." In this part we do not mention we will assess a parametrisation of the melt rate.*

*Section 2.3: "In this part of the study we focus on how to inject the observed ice shelf meltwater flux into the ocean model. Therefore, the ice shelf melting is prescribed and the heat flux is derived from the freshwater flux using Eq. 8. The computation of the melt rate from the off-shore ocean properties and ice shelf geometry could be included using the BG03 parametrisation or some adaptation of the Jenkins (2011) plume model, but testing these interactive melt parametrisations is beyond the scope of the study."*

*Section 5.5: "The parametrisation directly addresses this latter feature of the sub-ice-shelf ocean circulation and so is able to represent the ocean dynamics associated with the overturning circulation within the cavity." As for the abstract we do not mention that it is a melt parametrisation.*

*And as you mentioned, it is quite clear in the conclusion: "We do not describe a way to compute the melt rate itself. To tackle this issue, this work needs to be combined with a parameterisation of ice shelf melting (for example: Beckmann and Goosse, 2003; Jenkins et al., 2011)."*

- In section 2.2.1, it is assumed that the ice shelf is "in hydrostatic equilibrium in water at the reference density isf, taken to be the density of water at a temperature of − 1.9 ◦ C and a salinity of 34.4". Can the authors explain why they make such assumption?

*In the ISOMIP case, this assumption is used as the initial condition of ISOMIP are -1.9C and 34.4 PSU. In realistic case, we kept this value by simplicity. -1.9°C is a good estimate of the water temperature at an ocean/ice interface. 34.4 PSU is a good estimate of the mean salinity over the Antarctic continental shelf. The mean salinity from WOA2013 between 0-1000m everywhere the bathy is shallower than 1000m and south of 55S is 34.42. Text is now: "We assume the ice shelf to be in hydrostatic equilibrium in water at the reference density $\rho_{isf}$, taken to be the density of water at a temperature of −1.9°C (freezing point) and a salinity of 34.4 PSU (mean salinity over Antarctic continental shelves)."*

- Section 3.3: what would happen in case of a "Losh TBL" thicker than the vertical resolution? Then, in Fig. 3, the authors show the effect of using 31, 46 and 75 levels based on standard stretching parameters. They conclude that 75 levels might not be enough, but they don't issue any recommendation on how many levels should be used in standard NEMO simulations. Including greater values in Fig. 3 (e.g. L100, L150) would be useful for the community. Finally, these sensitivity results likely depend on the slope of the ISOMIP ice draft, and the authors should probably discuss the generalization of these results.

*The text to described what is happening in the Losh boundary layer is described in section 2.2. It has been reformulate like this: "Following L08, the noise due to the spatially varying size of the top cells is suppressed by computing $T_w$ and $S_w$ in Eq. 7,9 and 10 as the mean value over a constant thickness, assumed to represent the top boundary layer thickness ($H_{TBL}$, i.e. properties are averaged over the cells entirely included in the top boundary layer and a fraction of the deepest wet cell partly included in the top boundary required to make up the constant $H_{TBL}$)."*

*The case where HTBL larger than the horizontal resolution has also been highlighted by the reviewer 2. Result of experiment with a resolution of 5m and 10m with a Losh boundary layer set to 30m are now described in the text.*

*The new text is "The choice of vertical resolution and Losh $H_{TBL}$ strongly affects the ice shelf melting. When $H_{TBL}$ is tied to the vertical resolution, finer resolution gives lower melting. Under melting conditions, a thin, fresh and cold top boundary layer appears in the top metres of the ocean next to the ice shelf base. With finer vertical resolution, a thinner and colder top boundary layer can be resolved, resulting in weaker melting (Fig. 3a). Our sensitivity experiments show a maximum melt rate 4 times higher in the I_150M simulation (4.3 m y$^{-1}$) and 3 times higher in the I_60M simulation (3.1 m y$^{-1}$) than in the I_5M simulation (0.9 m y$^{-1}$). In analogous experiments, L08 found a similar sensitivity, with maximum melting 3 times larger at 45 m resolution than at 10m resolution. However, when $H_{TBL}$ is kept constant (I_5M30M, I_10M30M and I_30M), the total melt is insensitive to the vertical resolution. The total melt at high vertical resolution (5 m or 10 m) with a 30 m Losh top boundary layer thickness (respectively I_5M30M and I_10M30M) is converging toward I_30M (Fig. 3a). This suggests that a more physical definition of $H_{TBL}$ (based on stratification, melt rate, etc …), rather than a constant $H_{TBL}$, could significantly change the melt rate with a high resolution model (beyond the scope of the paper)."*

*About the recommendation and the test of other vertical resolution, we tested 31L, 46L and 75L because these resolutions are (or were) commonly used for global hindcast. We are not aware of higher vertical resolution commonly used with NEMO. Extra sensitivity test will be pertinent only if the stretching function used to build the vertical coordinates is suitable for a global configuration (such test are out of the scope of this study). References to the 3 variable level configurations mentioned are added:*

*"With variable vertical resolution (I_31L, I_46L and I_75L), such as is typically used in global configurations of NEMO (Timmermann et al, 2005, Drakkar group, 2007 and Megann et al., 2014), the coarsest resolution in the cavity seems to determine the total melt."*

*Generalisation of this work is really not straight forward as many factor could influence the results (slope of the ice shelf, coordinate system used, melt formulation …). This kind of generalisation will be maybe be tackle in the paper describing the results of the ISOMIP+ experiment (Asay-Davis et al., 2016).*

- The year/time-period represented in the "real ocean application" is not clearly stated. As far as I understand, the results represent 1985, which is presented as sufficient to complete a 10-year spin-up and to give the first-order response to changes in ice shelf representation. Does it mean that the interannual variability is of secondary importance compared to the sensitivity to the representation of ice shelves? What about the comparison to the ice-shelf melt estimated by Rignot et al. (2013) that is undertaken in section 5.6? How strong is the interannual variability in basal melt, and can we expect melt rates in 1985 to resemble those in the 2000s? I do not expect a perfect match here, but at least, the possible limitations should be stated.

*The end year has been added. And by the way we found a typo in the start date. The run started in 1979 and run for 10 years. The new text is : 'The model is run for 10 years starting in 1979 and ending*

*in 1988, and the first order response is investigated using output from the last year of the simulation.'*

*In Jourdain et al. (2017), figure 2 shows clearly that after 5 years the fresh water flux from the melting reach an equilibrium state. Similar behaviour is found for cold and warm ice shelves (Fig. 5 in Timmerman et al., 2012). In R_MLT, the same is happening for Ross and Pine Island Glacier ice shelves (Fig. 1 in this review). After 5 years, the ice shelf melting is well span up (even after the first year it is mostly spin-up).*

[Figure]

*Figure 1: Monthly melt beneath Ross and Pine Island Glacier ice shelves in Gt/y.*

*About the melt rate in the last year and comparison with recent estimate, as the geometry used is a recent geometry (Fretwell et al., 2013) and the ice shelf regime (cold or warm) did not change over the last 40 years, we can reasonable assume that the model melting should match the Rignot estimates. Text has been added in section 5.6:*

*'The total ice shelf melting simulated in R_MLT (1865 Gt $y^{-1}$) is slightly above the range of the observational estimate of Rignot et al. (2013) (Table 3). In R_MLT, as in the observations, we can separate the ice shelves into two different regimes based on the temperature of the water masses on the continental shelves (Fig. 7d) and the average melt rate: the cold water (Fig. 13b-d) and the warm water (Fig. 13a) ice shelves. As the ice shelf cavity geometry is based on recent estimates (Fretwell et al., 2013) and the ice shelf regimes modelled in R_MLT are similar to those in the observations, the modelled ice shelf melting are expected to match the Rignot et al. (2013) estimates.'*

*About the ocean properties, in figure 2, we clearly show that the mean salinity over Amundsen Sea in front of Pine Island Bay is spin up after 7 years. Differences between R_noISF and R_ISF or R_PAR are much larger than the inter-annual variability. Comparison between R_ISF and R_PAR shows the same inter-annual variability in both runs. Therefore, it do not rule out the analysis and the conclusion we made in the run comparison. No change in the text.*

[Figure]

*Figure 2: Monthly salinity (average between 300 and 1000 m depth) in front of PIG ice shelf.*

*Jourdain, N. C., P. Mathiot, N. Merino, G. Durand, J. Le Sommer, P. Spence, P. Dutrieux, and G. Madec (2017), Ocean circulation and sea-ice thinning induced by melting ice shelves in the Amundsen Sea, J. Geophys. Res. Oceans, 122, 2550–2573, doi:10.1002/ 2016JC012509.*

*Timmermann, R., Wang, Q. and Hellmer, H. (2012): Ice shelf basal melting in a global finite-element sea ice/ice shelf/ocean model , Annals of Glaciology, 53 (60) . doi: 10.3189/2012AoG60A156*

**Other very minor suggestions & typos:**

- Abstract, 5th sentence: "decrease" -> "decreased" or "decrease in". *DONE*

- Section 2.2.2: expand "ISOMIP". *DONE*

- Section 2.2.2, after equ.  14, expand "tbl" and mention that it's defined further in the text. *Based on the comments from Xylar Asay Davis and your, we decided to expand the Tbl acronym in the text and we define for the section 2.2 the acronym $H_{TBL}$ for top boundary layer thickness.*

- Tab.1: heat capacities should be in J/kg/K. And it would be better to use the Greek letter for Rhoi (as in the equations). *DONE*

- Section 2.3, 3rd paragraph: I would replace "equilibrium depth" with something clearer like "floatation depth" if it's what the authors mean. *By equilibrium depth, we mean the depth where the plume density equal the density of the ambient water. So, we think equilibrium depth is the correct word. "Floatation depth" could lead to confusion with the base of the ice shelf. Precisions are added into the text: "… thus an overturning between the grounding line depth and the equilibrium depth (the depth where the density of the plume is equal to the density of the ambient water)"*

- Section 2.3, last paragraph: expand "fwf". *DONE*

- Section 3.1: please add some information about the initial state and T,S restoring if any. *We add the information in the first paragraph of section 3.1: "The water is initially at rest and has a potential temperature of -1.9◦ C and a salinity of 34.4 PSU. No restoring is applied to either the temperature and salinity."*

- It would be better to have the labels for the x-axes in Fig.2a,b . Also, (a) and (c) are swapped in the figure caption. *DONE, fontsize was also changed to ease the reading and extra simulation point added.*

- Section 3.2, about refreezing: is there any frazil formation in the water column?

*The refreezing occurs only at the ice/ocean interface. Furthermore, the properties at the ice/ocean interface in case of freezing (drag, exchange coefficient …) are the same as under melting conditions. Text in section 2.2 has been modified. New text is:*

*". Parameter values used in Eqs. 7-12 are defined in Table 1. Hereafter, Eqs. 10-12 are referred to as the "three equation" ice shelf melting formulation. At the differences of more sophisticated model (Galton-fenzi et al., 2012), the parameter used in the "three equation" formulation are not dependent of the surface state (freezing or melting) and the freezing only occurs at the ice/ocean interface."*

- Last sentence of section 3.3 (about Fig.3b): another reason could be that overturning and barotropic circulations have physically the same dependence on total melt rates.

*It could be, but it does not explain why the sensitivity of overturning and stream function are weak. No text change.*

- Section 5.1: the authors need to tell a bit more about how tidal mixing data from FES 2012 are used in NEMO, and maybe how it accounts (or not) for the effects of tides on ice shelf melt rates.

*The internal energy wave used in the parametrisation is derived from a barotropic model of the tides utilizing a parameterization of the conversion of barotropic tidal energy into internal waves. Under the ice shelves, the internal energy wave map is set to 0 by simplicity.*

*The new text: "The geothermal heat flux is assumed to be constant and set to 86 mW/m² (Emile-Geay and Madec, 2010), while the internal wave energy used in the tidal mixing parametrisation (0 under the ice shelf by simplicity) is derived from the tide model FES 2012 (Carrère et al., 2012)."*

- Section 5.1, about "The model is run for 10 years starting in 1976, and the first order response is investigated using output from the last year of the simulation": given that there seems to be interannual variability in these simulations, why analysing only one year? Isn't there "first order" variability at the interannual time scale?

*There is inter-annual variability in our model, that is right. Figure 2 show clearly that the differences between run are larger than the inter-annual variability (R_noISF vs R_ISF or R_PAR) or that the inter-annual variability is very similar (R_ISF vs R_PAR). In the first case the signal we are looking at is much larger that the interannual variability. In the second case, this means that there is no "first order" variability at the interannual time scale. This does not rule out the conclusion we made on the performance of the simple parametrisation (R_PAR) compare to the standard case (R_noISF). No change in the text.*

- Last sentence of section 5.2: "results from R_MLT are used to evaluate the 3 equation ice shelf melting formulation in NEMO" -> I think it's not only the 3 equation formulation that is evaluated, but also the bathymetry, the ocean thermal forcing, vertical mixing, etc, etc. Same comment for the first paragraph of section 5.6 (although it is clear in 5.6.3).

*We agree. The text has been changed to make it clear.*

*'Finally, results from R_MLT are used to evaluate the modelled ice shelf melting in our circum-Antarctic configuration using the "three equation" ice shelf melting formulation.'*

*"To compute melt rates for other oceanic states interactively, and eventually to couple the ocean model to an evolving ice sheet model, requires the "three equation" formulation for ice shelf melting. Next, we evaluate the ability of the described circum-Antarctic configuration with the "three equation" ice shelf melting formulation to modelled ice shelf melting."*

- Fig.9: could the difference between Dutrieux et al. (2014) and NEMO simulations come from different periods under consideration?

*Yes it could. Precision on the period of the climatology used and on the model year has been added in Figure 9 caption. New caption is: "Profiles (year 10, 1988) in Pine Island Bay in R_noISF (blue), R_ISF (red) and R_PAR (green) of a) salinity and b) temperature. Climatology from 1994 to 2012 (Dutrieux et al, 2014) is in black."*

- Section 5.6.3: a reference to Millan et al. (GRL, 2017) could be included to highlight uncertainties in bathymetry and ice drafts. *We add a sentence to mention that bathymetric features are missing in the BEDMAP2 data set.*

*The new text is : "The most recent bathymetry and ice shelf draft reconstruction of Amundsen Sea (Millan et al., 2017) shows large missing features in the BEDMAP2 data set. In BEDMAP2, for many ice shelves, there are only indirect observations of ice draft, based on satellite surface elevation data, while the sub-ice bathymetry is often poorly constrained. For some ice shelves (Getz, Venable, Stange, Nivlisen, Shackleton, Totten and Dalton ice shelves, for some of the thickest areas of the Filchner, Ronne, Ross, Amery ice shelves and for the ice shelves of Dronning Maud Land), the*

*floatation needs to be enforced by lowering the sea bed based on nothing more than extrapolation of cavity thickness from surrounding regions of grounded ice and 100 m thick cavity. Consequently, more data are needed for effective modelling (Fretwell, et al., 2013), because cavity geometry has a major impact on the simulated melting by controlling the water mass structure and circulation within the cavity (Rydt, et al., 2014)."*

- Section 5.6.3:  Makinson et al.  (GRL 2011) estimate that tides double the net melt rate underneath Filchner and Ronne Ice Shelf. *The citation we used in the text in section 5.6.3 and 5.6.1 was wrong. Instead of Makinson et al., 2012 we now used Makinson et al. (2011) (Makinson, K., Holland, P. R., Jenkins, A., Nicholls, K. W., and Holland, D. M.: Influence of tides on melting and freezing beneath Filchner-Ronne Ice Shelf, Antarctica, Geophys. Res. Lett., 38, L06601, doi:10.1029/2010GL046462, 2011.)*

---

## Author Comment (AC3) · 30 May 2017

**Review of Mathiot et al. "Explicit and parametrised representation of under ice shelf seas in a z* coordinate ocean model"**

*Reviewer: Xylar Asay-Davis*

**General comments:**

This paper discusses the addition of ice-shelf cavities into the NEMO ocean model, and a series of tests used to gain confidence in the model's behavior through idealized simulation results that can be compared with those from other models, to understand sensitivity to certain parameters (notably ocean vertical resolution), and to validate the model against observations in a realistic configuration. The paper also presents a parameterization for prescribing known melt fluxes in the absence of ice shelf cavities, and shows that the parameterization captures many of the features of the flow produced with ice shelf cavities. Overall, I find that this paper does a very good job at documenting the new NEMO features. The work is important for the field, especially because NEMO's capability to simulate ice shelf cavities is already used by several groups and NEMO is likely to become one of the most widely used models with this capability. The paper is well organized and the experiments are sensible and appropriate, exploring both the potential of the model and making clear some if its limitations and biases. I recommend a number of minor revisions to the manuscript. If these are addressed, I would recommend the manuscript for publication.

**Authors:** We would like to thank you for the very constructive, positive and encouraging comments. You will find below a reply to each point in *italic blue* with, when necessary, the new text between quote.

**Specific comments:**

Title: I believe that GMD requires the model name and version to be part of the title for model description papers.

*We change the title "Explicit representation and parametrised impacts of under ice shelf seas in the z\* coordinate ocean model NEMO 3.6"*

P. 1:

L. 10 I would suggest defining the acronym NEMO here, the first time you use it (other than the title, assuming you follow my previous suggestion) *DONE*

L. 15 HSSW needs to be defined the first time it is used here. *We expanded HSSW in the abstract to limit the number of acronym in it.*

L. 20 "that has been assessed": It is not clear to me what this phrase means in this context. Perhaps you could replace it with something more specific and meaningful?

*In the abstract, we reformulate this sentence and slightly change the one before to avoid repetition:*

*"Mimicking the overturning circulation under the ice shelves by introducing a prescribed meltwater flux over the depth range of the ice shelf base, rather than at the surface, is also assessed. It yields similar improvements in the simulated ocean properties and circulation over the Antarctic continental shelf to those from the explicit ice shelf cavity representation. With the ice shelf cavities opened, the widely used "three equation" ice shelf melting formulation, which enables an interactive computation of melting, is tested. Comparison with observational estimates of ice shelf melting indicates realistic results for most ice shelves. However, melting rates for Amery, Getz and George VI ice shelves are considerably overestimated."*

L. 28 "very specific": I would suggest a different phrase, like "different from other freshwater sources". The whole sentence may need to be reworded to avoid too many redundant references to "freshwater".

*We reformulate the sentence like this: "The ice shelf melting contribution to the Southern Ocean freshwater forcing is different from the iceberg melting and precipitation. Ice shelf melting is injected into the ocean at depth whereas precipitation is input at the surface and icebergs inject melt water at a range of depths, but primarily in the top ~100 m."*

P. 2:

L. 1-2 Does the freshwater forcing from melting sea ice also deserve to be mentioned in this context?

*We don't want to mention this point because, if we do so we need also to mitigate it by the fact that sea ice is also a sink of fresh water for the ocean in winter (ice formation in winter) and this information is not useful for the rest of the paper. The text is not changed.*

L. 11 "spread": To me, this verb implies that water masses arrive at the continental shelf in specific locations and spread out from there. For some water masses like CDW, this may be correct but it strikes me as strange to think of all water masses spreading across the continental shelf in this way.

We suggest "are present" instead of spread. The sentence is now: *"Basal melting of ice shelves is driven by the properties of the water masses that are present over the continental shelves, enter the ocean cavities and reach the grounding line where they initiate melting."*

L. 13 "towards the surface" I would be careful about this phrase because you make a point that it is important to correctly model the fact that melt water doesn't necessarily reach the surface. Perhaps just saying "upward" instead of "towards the surface" would avoid implying that the meltwater reaches the surface. *DONE*

L. 17 "...a temperature close to that of the surface freezing point..." should be "...a temperature close to the surface freezing point...", since the freezing point is a temperature. *DONE*

L. 17-22 This sentence is way too long. I would suggest one sentence for each mode.

*DONE, the new sentences are: 'Mode 1 melt is low, because HSSW has a temperature close to the surface freezing point and can melt ice at depth only because of the lowering of its freezing point*

*with increasing pressure. Mode 2 melt can be high if almost unmodified CDW has access to the sub-ice-shelf cavities. Mode 3 melt is intermediate and variable, depending on whether only the near-freezing core of ASSW, often designated Winter Water (WW), or the seasonally warmer upper layers can access the cavities.'*

L. 22-23 "The process is usually referred to as the ice pump." Because you've just mentioned refreezing in the previous sentence, it seems like the refreezing process is referred to as the ice pump. I would suggest clarifying which process it is that you are referring to.

*The definition of the ice pump was moved before the definition of the modes. The new text is: 'Basal melting of ice shelves is driven by the properties of the water masses that are present over the continental shelves, enter the ocean cavities and reach the grounding line where they initiate melting. The associated input of buoyancy triggers an overturning circulation with inflow at depth and outflow along the ice shelf base that carries meltwater upward. The process is referred to as an ice pump when the ascending waters cause refreezing (Lewis and Perkin, 1986).'*

P. 3:

L. 8-9 "So whatever the resolution, some cavities remain unresolved." This is clearly not true, since 1 km resolution (for example) should be sufficient to resolve even Ferrigno ice shelf. In the remainder of the manuscript, you are careful to state that some ice shelves will remain unresolved at any practical resolution for global modeling. Please include those caveats here, too. Keep in mind that what seems impractical may become practical in the coming years to decades.

*As suggested we stay focus on the global model resolution and remove this sentence as the main point was already mentioned the sentence before. The new text is:*

*"Ice shelves range in size from the giant Ross ice shelf (500,000 km²) to the tiny Ferrigno ice shelf (117 km²). This means that current global ocean model configurations are not able to resolve explicitly all the ice shelf cavities. For this reason, a simple way to include unresolved ice shelf melting in the ocean model that mimics the circulation driven by ice shelf melting at depth is also presented here."*

L. 11-16 Please include section numbers as part of describing the structure. This helps the reader to better navigate the paper based on this outline. *DONE*

L. 19 The definition of the NEMO acronym should go earlier (in the abstract). The version number should go in the title, meaning it may not need to be mentioned again here.

*We defined the acronym in the Abstract and in the introduction the first time it appeared. We decided to keep the version number in the model description.*

L. 20 Please explain what an Arakawa C-grid is. It also seems awkward to mention the "nonlinear filtered free surface option" without giving some sort of explanation of what this is and why it is the appropriate choice for this work.

L. 26 Maybe z* needs to be explained first, at least qualitatively?

*About the 2 previous point, basic precision on the Arakawa C grid has been added as well as on z\* coordinates. 'Filtered' was suppressed as it do not bring extra information. 'Non-linear free surface' was used as a synonym of z\* coordinate, but as it can lead to confusion, we replace it by z\* coordinate. The new text is:*

*"2 Model description*

*2.1 Ocean model*

*NEMO is a primitive equation ocean model, and this study uses version 3.6 of the code. The variables are distributed on an Arakawa C-grid; i.e. the scalar point (temperature, salinity) is defined on the centre of the cell and the vector points (zonal, meridional, vertical velocity) are defined on the centre of each face (Arakawa, 1966). We also make use of the time varying z\* vertical coordinate; i.e. the variation of the water column thickness due to sea-surface undulations is not concentrated in the surface level, as in the z-coordinate formulation, but is distributed over the full water column (Adcroft and Campin, 2004).*

*A complete description of the schemes and options available in NEMO is available in the documentation (Madec, 2012). A full description of the configurations used in this study is presented in Sect. 3.1 for the idealised configuration and in Sect. 4.1 for the realistic configuration.*

*2.2 Ice shelf/ocean interaction description*

*2.2.1 Ocean dynamics*

*The z\* vertical coordinate can be used with a sea ice model (Campin et al., 2008) in NEMO (Madec et al., 2012)."*

L. 26 "can be used with sea ice model" should probably be "can be used, unmodified, with a sea ice model". (Note also the missing "a" in front of "sea ice model" After all, you are saying that z\* can also be used with an ice shelf but it requires modification.

*We think the mention of modification for the ice shelf case is enough. Minor text change, the new text is:' The z\* vertical coordinate can be used with a sea ice model (Campin et al., 2008) in NEMO (Madec et al., 2012). However, modelling the ocean circulation within an ice shelf cavity in z\* coordinates requires some modification of the existing code.'*

L. 30 What about shallow regions covered in thick sea ice? Is there a provision in NEMO for preventing sea ice thickness from becoming of the same order as the ocean column thickness?

*There is nothing in NEMO to prevent this case to happen. As this is a little bit out of the scope of the paper, the text is not modified.*

P. 4

Eqs. (1)-(4) Many problems here:

● The indexing here isn't entirely clear. Either a diagram or further explanation in the text would be helpful. k=1 would appear to be the surface, but this isn't explicitly stated. kz=1 would seem to be the first level.

● In (2), the upper limit on the sum should be k-1, not kisf-1, I think. *That's right, DONE*

● Zw appears to be an inconsistent mix of positive up in (1) (assuming eta is positive up, which (4) seems to imply) and positive down in (2) and (3). You use a negative sign for depth in most subsequent figures and text, so I think (2) and (3) need a minus sign

● It seems like eta needs to be added to (2) and (3) for them to be consistent with (1) and (4). That is, If I plug in k=1 into (2), I would expect to get (1). If I plug in k max into (3), I would expect to hit the sea floor, Zw(kmax) = -Hisf - H.

● It would be helpful to have H be written explicitly as a sum of dz0,T

*Sign convention is now state on the text. The sign convention in the figure is completely disconnected from the convention used in these equations. Eq. 1 is changed to Zw = 0 (as it is the depth of the ocean/atmosphere interface). So with this, in eq 2, Zw(k=kisf)=Hisf (or ice shelf draft) and eq. 3, the depth of the sea bed/ocean Zw(k=kbot+1) = Hisf + H + η. Precision has been added for H: "H the total water column thickness (sum of all the wet cell vertical thicknesses at time 0)"*

Eqs. (5)-(6) It might be relevant to indicate how these equations are discretized in the model. For example, presumably density is piecewise constant in layers and pressure gradients are explicitly substituted with density gradients?

*Precision has been added:*

*"The hydrostatic pressure gradient at a given level, k, (first term in Eq. 6) is computed by adding the pressure gradient due to the ice shelf load (defined as the first term of Eq. 5) with the vertical integral of the in-situ density gradient along the model level from the surface to that level."*

L. 32 "The same limitation is expected..." You just said that partial cells compare favorably to sigma coordinate. You mentioned that bottom topography can be "challenging" but you haven't mentioned any specific "limitation" that should apply along the ice shelf base.

*This part is really confusing as you mention. The text has been modified: "Representation of the bottom topography is difficult in z coordinate models. The partial cell scheme allows a more accurate representation of bottom topography through the use of partially wet cells (Adcroft et al., 1997). Solutions obtained with this scheme compare favourably with those obtained with sigma coordinate models (Adcroft et al., 1997) and also with more realistic solutions (Barnier et al., 2006). Following L08, we apply the partial cell scheme developed for the bottom topography to the top cells beneath the ice shelf base. For stability reasons, the minimum thickness of the bottom and top cells is set to the smaller of 25m or 20% of a full cell. However, representation of density driven flow in a z coordinate model (even with partial cells), like the overflow, is challenging (Legg et al., 2006). Thus the representation of the buoyancy driven flow along an ice shelf base is expected to present analogous problems."*

P. 5:

L. 7 "it is sometime necessary" should probably be "it would sometimes be necessary" since you state right after this sentence that you don't do this. *DONE*

L. 8-10 You talk about the importance of parameterizing melt from ice shelves that are too small to resolve otherwise. You indicate later on that significant regions close to grounding lines and even most of some ice shelves are removed they are too steep and/or too thin to be easily represented in a z* coordinate without significant "digging" into bathymetry and/or ice draft. Might this not be having a comparable effect to the unresolved ice shelves and require some alternative treatment or parameterization?

*We agree, this could have exactly the same effect than for the unresolved ice shelves. A parametrisation needs to be developed to compute the melt in this mask region (between the true grounding line and the model grounding line) and something similar to what is done in R_PAR could be used to mimic the ocean circulation in these missing cells. As this is more related to the perspective, the conclusion has been modified instead of the model description part.*

*New text in the conclusion is:*

*'To apply this work to a global coupled ice sheet/ocean model, we will need some further developments. First, a better knowledge of sub-ice-shelf cavity geometries and key processes that contribute to melting (drag, tides, boundary layer, etc ...) could lead to improvements in the ice shelf representation. Secondly, parameterisations need to be developed to represent the processes (melt and circulation) where the resolution is not fine enough to represent the ice shelf cavity geometry correctly as at the grounding line for example.. Finally, a conservative wetting and drying scheme needs to be developed to allow the grounding line (and calving front) to move back and forth.'*

In many cases, the largest melt rates are at or near the grounding line. By moving the grounding line, particularly by moving it systematically toward the ocean, don't you think you are biasing the model toward lower total melt (perhaps offset by other biases to produce the higher total melt you find in the end)?

*As you mention, the bathymetry and ice shelf modification process (to maintain 2 cell in the water column) could never open cell beyond the grounding line. So, the ice shelf area in NEMO could only be smaller or equal to the true area. So, the model could have a bias toward low melt. Evaluating this bias is not simple with the present simulation. Because of the temperature and velocity dependence at the ice shelf interface, it is difficult to estimate what the melt rate would be beneath the masking cell if this particular water column was opened. A minor modification has been made to mention this:*

*"Rather than making such extensive modifications to the topography, we regard the combination of vertical and horizontal resolution as too coarse to represent the sub-ice cavity geometry in these places, and instead we ground the ice shelf. Consequently some ice shelves have a reduced area."*

L. 13-14 It would be nice to have a more mathematical description of what is meant by "correction" here.

*We reformulate the sentence as this: "For regional configurations with open boundaries, the normal barotropic velocity around the boundary at each time step is corrected to force the total volume to be constant. The correction ensures that the net inflow (the combination of inflow at the open boundary, runoff, ice shelf melting and precipitation) and net outflow (the combination of outflow at the open boundary, ice shelf freezing and evaporation) are balanced."*

L. 23  You don't explain until the middle of the next page how Tw is computed from model temperature.  You have also not mentioned anything about a boundary layer so far.  It might be worth saying something like "Tw the temperature averaged over a boundary layer below the ice shelf (explained below)," *DONE*

L. 36 "ISOMIP formulation".  You should define the ISOMIP acronym here and explain why the formulation is named this (since ISOMIP has not been mentioned previously). *DONE*

L. 27 "Jenkins et al. (2001)" There are 2 formulations for the heat and freshwater fluxes in this paper. Could you point to the specific equations in the paper you used?  I presume you use (24) from that paper, not (25) because the volume flux of freshwater at the surface is explicitly modeled.

*Precision added.*

Eq. (10):

● There is no Eq. (9), so you need to renumber. *DONE*

● What about vertical advection, a la Holland and Jenkins (1999).  Advection typically dominates diffusion except where melt/freezing rates are very small in magnitude.

*The temperature profile through the ice shelf is assumed to be linear (case no advection, diffusion in Holland and Jenkins, 1999). The case advection and diffusion through the ice shelf has not been tested. No change in the text.*

P. 6:

Eq. (12) Units are needed.  It would be cleaner to define liquidus coefficients with symbols and put them in Table 1. *DONE*

L. 28-29 "averaged over the first cell and that part of the second wet cell required to make up the constant boundary layer thickness"  You never need a third cell to get to 30 m?  You don't allow the user to set a boundary layer thickness that is significantly larger than the resolution (e.g 30 m for 5 m resolution)? What would happen in your variable resolution case if you had a shallow ice shelf so that dz < 25 m but you specified a 30-m boundary layer thickness?  Is the BL thickness never allowed to be thicker than the local resolution of a full cell?

*With partial cell at the top, this case happens for all the simulation. For example in a shallow ice shelf, vertical resolution is around 10m at the ice shelf base. With top partial cell, it means the effective top cell thickness is between 1 and 10m (depending on the ice shelf draft). In case of a 30m $H_{TBL}$, the properties in the top boundary will include, the top cell (let say) 5m at level kt, the full cell at level kt+1 (10m), the full cell at level kt+2 (10m) and half of the cell at level kt+3. The text is changed like this: 'properties are averaged over the cells entirely included into the top boundary layer and a fraction of the deepest wet cell partly included in the top boundary required to make up the constant $H_{TBL}$'*

P. 7:

L. 1-2 This could use some clarifying, especially for the non-constant layer thicknesses with variable numbers of levels. How are the layer thicknesses determined? What is the range of values?

*Varying was replaced by various as varying was misleading. In case of variable vertical resolution with partial cell, there is 2 cases: top cell smaller than $H_{TBL}$ and top cell larger than $H_{TBL}$. In the first case, 'properties are averaged over the first cell and that part of the second wet cell required to make up the constant $H_{TBL}$'. In the second case, only the top cell is concerned by HTBL '$H_{TBL}$ is set to the top cell thickness' (as if Losh Top Boundary Layer Scheme not used for this cell). In z\*, the cell included in the Losch Boundary Layer Scheme (and the respective portion of the second cell) are computed at every time step.*

L. 4-6: This point has been made almost word for word in the intro. I would suggest shortening or removing this sentence.

*We decide to shorten the sentence by removing the references to the different ice shelf size: 'which range in size from the vast Filchner-Ronne and Ross ice shelves to the much smaller ice shelves of the Bellingshausen and Amundsen Seas'.*

*The new text is:*

*'Global ocean model configurations are typically unable to resolve all the ice shelves around Antarctica. Despite their limited extent, the smaller ice shelves nevertheless make a significant contribution to the total meltwater flux from the ice sheet. We therefore need a way to mimic the impact of unresolved cavities on the ocean.'*

L. 9: "Figure 1c" should be "Fig. 1c". It's slightly odd to mention only this panel of the figure and get to the others only much later in 5.2. Maybe you could mention here that you will explain the remaining panels in Fig. 1 in Sect. 5.2.

*DONE, Fig. 1d is mentioned few line below and we add a sentence at the end of the paragraph describing Fig. 1d. New text is: "The idea tested in this paper is to spread the freshwater due to ice shelf melting evenly between the grounding line depth and the depth of the calving front. In this case, the model creates its own plume along the vertical wall (Fig. 1d, no cavity in this case) and thus an overturning between the grounding line depth and the equilibrium depth (depth where the density of the plume reaches the density of the ambient water). Fig. 1a and 1b are discussed in Sec. 5.2."*

L. 12-13: "spread the freshwater due to ice shelf melting evenly between the grounding line depth and the depth of the calving front" This maybe deserves a bit more emphasis and explanation. One might think the plume-like structure of melting would be more consistent with melt fluxes exiting closer to the surface. My ISOMIP+ simulations, for example, and those of most other models suggest that T and S are not strongly affected by melting except close to the interface. Why, then, is a uniformly distributed flux (both horizontally and vertically) the preferred choice? Is this for simplicity and because results are reasonable, or is there a deeper physical reason? Comparison with observations near PIG in Fig. 9 suggest that neither the explicit modeling nor the parameterization is mixing deep enough. This suggests maybe not enough entrainment when explicitly modeling the ice shelf cavity, and may suggest that more melt flux should actually go in at depth than closer to the surface.

*We are aware that one might think that the fwf should be put at the exit level (as in BG03) or even at higher level (equilibrium depth) as this is the depth where the fresh water ends. But by doing this, we completely miss the effect on the fresh water melting on the ocean circulation, ie the triggering of the ice pump. So to trigger an "ice-pump" in a model without cavity similar to the ice-pump generated in a model with ice shelf cavity, the melt water has to be spread in depth up to the grounding line. The fwf is distributed uniformly horizontally and vertically by simplicity and because this simple distribution give reasonable results. The effect and the limit of both approach are very clear in section 4, Figure 5 and section 5.5, so the text is not changed.*

*Obviously, this could be improved (other distribution over the vertical based on the ice shelf geometry or the location of the ice shelf melt …) to fit again better the simulation R_ISF. Or play with the horizontal distribution to fit the horizontal circulation beneath the ice shelf for the Ross and Filschner-Ronne ice shelf. But what we really want for wider use is a parametrisation of the circulation beneath the ice shelf as simple as possible to use (ie input file easy to build). In this case, the input file contains only for each cell in-front of the ice shelves the grounding line depth, the calving front depth and the total amount of melt water to inject in depth).*

*A sentence has been added at the end to explain where are the limit of our study: "The computation of the melt rate from the off-shore ocean properties and ice shelf geometry could be included using the BG03 parametrisation or some adaptation of the Jenkins (2011) plume model. The parametrisation tested in this study is kept as simple as possible for ease of use in a wide range of applications. Further testing of other interactive melt parametrisations or fresh water distributions that are functions of the ice shelf geometry or melt rate is beyond the scope of this study."*

L. 26 ISOMIP was never defined so should be defined either here or above when you speak of the "ISOMIP formulation". *DONE in section 2.2.2 when we described the ISOMIP formulation.*

L. 26 Also, you should mention that you are performing ISOMIP experiment 1.01 (there were 3 experiments defined). *DONE, section 3.1 start now like this: The ISOMIP setup follows the recommendations of the inter comparison project for experiment 1.01 (Hunter, 2006)*

P. 8:

L. 6-7 "at which time the system is in quasi-steady state". Two things: first, to me "quasi-steady state" is used for systems that oscillate (possibly chaotically) around a steady state, whereas this system has enough viscosity to relax toward what I would expect is a true steady state.

*Quasi steady state was replaced by the more generic term "steady state".*

Second, I have found in my own simulations that even 30 years isn't really enough time to be that close to steady state. Since NEMO's behavior is typically similar to POP2x, the mode I use, I would expect that the same is true for you. Do you have reason to believe that it *is* close to steady state (e.g. you ran another 5 or 10 years without much appreciable change)?

*The time date day 10 000 was chosen to fit the plot from L08. This allow an exact comparison. We agree, the trend on moc and stream function is not flat after 10 000 day. However, the trends are stable after 15 years. A different date will made the comparison harder.*

*About the validity of the conclusion we made, in Fig. 1 (in this document), an equivalent of the figure 3 of the paper made after 20 years of run instead of 27.4y. Comparison of Figure 1 with Figure 3 from the paper shows that the amount of melt and freezing are slightly larger, but very similar conclusion can be made. Behaviour of the high-resolution model with respect to the low resolution model are similar. Bsf and moc are larger after 20y than after 27.4y (10000 days) and the spread of the ensemble is slightly larger after 20y than after 27.4y. So, for the conclusion we made, we are not convince on the utility of 10 more years.*

*No change are made in the text.*

[Figure]

*Fig. 1: equivalent of the figure 3 of the paper after 20y of run instead of 27.4 years. a) Total melting rate versus total freezing rate, and b) meridional overturning circulation versus barotropic stream function (bsf) for all the ISOMIP sensitivity experiments (I_5M, I_10M, I_30M, I_60M, I_100M, I_150M, I_31L, I_46L and I_75L). The simulations I_XXM are with constant vertical resolutions and the simulation I_XXL are with variable vertical resolution. Details are given in Table 2.*

L. 31 Each ISOMIP experiment named in Table 2 needs to be explained, either in the caption or in the text.

*We found the explanation gave in table 2 to describe every experiments used in Figure 3 clear enough. Some experiment are only used in Figure 3 and not in the text in order to support the conclusion made, but we do not think there is a need to detailed the results of every single experiment as I_10M for example. Some minor change has been made in the text:*

*"To evaluate the impact of this choice on the ocean circulation beneath the ice shelf, nine simulations with vertical resolution ranging from 5m (I_5M) to 150m (I_150M) have been carried out (Table 2)."*

*"With variable vertical resolution (I_31L, I_46L and I_75L), such as is typically used in global configurations of NEMO (Timmermann et al, 2005, Drakkar group, 2007 and Megann et al., 2014), the coarsest resolution in the cavity seems to determine the total melt."*

Also, adding a prefix like "I_" for ISOMIP would make the division between the different experiment categories clearer and would be more consistent with other experiment names in the table. *DONE*

L. 31 Also, why no experiment with (say) a 5-m vertical res. but a 30-m TBL?  Is this not supported? Such an experiment would better demonstrate whether the model converges with increasing vertical resolution.  In my experience with POP2x, it does.  This means that, if we want a boundary layer to be present, we should use a physical length scale to determine its depth (like KPP does, for example), rather than tying it to the vertical resolution.  More discussion of this point below.

*NEMO is supporting a 30-m TBL with a 5-m resolution. The initial idea was to play with the vertical resolution and keep the Losch boundary layer to limit the noise (ie keeping it to as low as possible when possible and no more than 30m). The case ISOMIP with 5m resolution and a 30m L08 tbl is supported by NEMO. In this case, the temperature used to compute the melt will be the average of 1 partial cell at the top, 5 full cell plus one partial cell at the bottom of the top boundary layer. The suggested simulation and an other one with 10m resolution and 30m for the boundary layer has been run 10000 days and added in the discussion and in the figure 3.*

*The new text is: "The choice of vertical resolution and Losh $H_{TBL}$ strongly affects the ice shelf melting. When $H_{TBL}$ is tied to the vertical resolution, finer resolution gives lower melting. Under melting conditions, a thin, fresh and cold top boundary layer appears in the top metres of the ocean next to the ice shelf base. With finer vertical resolution, a thinner and colder top boundary layer can be resolved, resulting in weaker melting (Fig. 3a). Our sensitivity experiments show a maximum melt rate 4 times higher in the I_150M simulation (4.3 m y$^{-1}$) and 3 times higher in the I_60M simulation (3.1 m y$^{-1}$) than in the I_5M simulation (0.9 m y$^{-1}$). In analogous experiments, L08 found a similar sensitivity, with maximum melting 3 times larger at 45 m resolution than at 10m resolution. However, when $H_{TBL}$ is kept constant (I_5M30M, I_10M30M and I_30M), the total melt is insensitive to the vertical resolution. The total melt at high vertical resolution (5 m or 10 m) with a 30 m Losh top boundary layer thickness (respectively I_5M30M and I_10M30M) is converging toward I_30M (Fig. 3a). This suggests that a more physical definition of $H_{TBL}$ (based on stratification, melt rate, etc …), rather than a constant $H_{TBL}$, could significantly change the melt rate in a high resolution models (although investigation of this is beyond the scope of the paper)."*

P. 9

L. 9 "coarsest resolution in the cavity seems to determine the total melt." I think this needs some more discussion.  This is likely because the coarsest resolution corresponds to the deepest part of the ice shelf where melt rates are typically highest (thus having the greatest effect on the total flux), right?

*Yes, it is. Figure 2 is showing the cumulative melt from the grounding line (southern part of the domain) to the northern part of the domain. This figure shows that more than 50% of the melt occurs*

*where the ice shelf draft is between 700m and 550m (whatever the resolution). In case of variable resolution, it is where the resolution is the coarsest and thus constrain a lot the total melt. The figure is not added to the text. However, some precisions are added to the text:*

*'With variable vertical resolution (I_31L, I_46L and I_75L), such as is typically used in global configurations of NEMO (Timmermann et al, 2005, Drakkar group, 2007 and Megann et al., 2014), the coarsest resolution in the cavity seems to determine the total melt. This is because more than 50% of the melting occurs between 500 m and 700 m depth where the resolution is coarsest (not shown). This could be an issue for modelling ice shelf melting with the standard configuration used for climate applications because Dutrieux et al. (2013) show that, for some ice shelves with high melt rates, most of the melt may occur over a small area close to the grounding line, where the resolution is coarsest.'*

[Figure]

*Figure 2: Cumulative melt from the grounding line for every ISOMIP simulation.*

L. 11-12 This is almost identical to text in 5.1, so maybe trim one or the other (probably trim here and refer to that section).  Also, there you say that the maximum resolution is 150 m, which is very different from 40 m.

*We trim in the section 3.3.*

*The difference between 40m and 150m is coming from the ice shelves you are looking at. In section 3.3 we are looking at ISOMIP case (grounding line (GL) at 700m) and in section 5, the deepest ice shelf is Amery (GL around 2000m deep).  In the first case the vertical resolution at the GL is about 40 m and in the second case, the vertical resolution at the GL is 150 m. The description of the vertical resolution is now: "The model uses 75 vertical levels with thicknesses varying from 1 m at the surface to 200 m at 6000 m depth, giving a vertical resolution ranging from 10 to 150 m beneath the ice shelves. See Sec. 3.3 for the effect of this resolution on ice shelf melting in an idealised case."*

L. 26 This geometry is the one for ISOMIP expt. 2.01.  Maybe change to "The geometry is the same as ISOMP expt. 2.01, which is the geometry from ISOMIP expt. 1.01 except ..." *DONE*

L. 27 Maybe give a specific equation and/or figure number from Asay-Davis et al. (2106)?

*The precision of the figure number is added. New sentence is: "The simulations are initialised with a warm linear profile typical of conditions on the continental shelves of the Amundsen and Bellingshausen Seas (Fig. 6 in Asay Davis et al., 2016 with constant value between 720 m and 900 m)."*

L. 29-30 The viscosity used in ISOMIP is already quite large compared to what we use in realistic simulations.  It seems like increasing this value by another factor of 5 renders any comparisons with a realistic configuration nearly impossible.  Also, can you talk about the cause of the noise at the ice shelf front?  That sounds troubling and viscosity may not have been the best way to handle it.  Why didn't similar noise show up in the realistic simulations?

*We investigated this issue. It appears that using the same vertical viscosity and diffusivity for unstable conditions as in the realistic configurations give negligible noise.*

*The text is changed to: 'The vertical eddy viscosity (diffusivity), in unstable conditions,  are increased from 0.1 m$^2$.s$^{-1}$ to 10 m$^2$.s$^{-1}$ to reduce the noise generated along the ice shelf front.'*

*All the simulations (the one with interactive melt used to compute the steady pattern of basal melt/freeze used in A_ISF, A_ISF, A_PAR and A_BG03). In addition, the temperature profile at the northern boundary changed slightly. In the first version, the profile was stretch from the surface to 900m instead of 720m in Asay Davis et al. (2016). We modify it to be exactly as Asay Davis et al. (2016) (ie linear) between surface and 720m and constant beneath 720m. Figures and values are changed accordingly.*

P. 10

L. 21 "so the behaviour described above may differ in a realistic configuration".  I'm not sure which "behaviour described above" is being referred to here for sure.  Do you mean that a good comparison in an idealized context doesn't necessarily imply good behavior under realistic conditions?

*Yes it is. The text has been modified to be more clear: "Nevertheless, the bathymetry and ice shelf draft are smooth in these idealised cases and the heat transfer coefficient is constant, so the favourable comparison with other models in the idealised ISOMIP setup between models as well as the good match between the idealised A_ISF and A_PAR experiment might not be reproduced in a realistic configuration. In the next section, we assess both the explicit ocean cavity representation and the cavity parametrisation in a realistic circumpolar configuration."*

L. 28-30 This is a brilliant solution for coarsening the grid resolution close to the South Pole!

P. 11

L. 1-2 As mentioned, this is almost identical to text in 3.3, where you claim instead a vertical resolution as coarse as 40 m in cavities. Please trim the text in 3.3 and refer to here, making sure the sections are consistent with each other.

*The sentence in 3.3 is removed. The difference between 40m and 150m is coming from the ice shelves you are looking at. In section 3.3 we are looking at ISOMIP case (grounding line (GL) at 700m) and in section 5, the deepest ice shelf is Amery (GL around 2000m deep). In the first case the vertical resolution at the GL is about 40m and in the second case, the vertical resolution at the GL is 150m. The description of the vertical resolution is now: "The model uses 75 vertical levels with thicknesses varying from 1m at the surface to 200 m at 6000 m depth, giving a vertical resolution ranging from 10 to 150 m beneath the ice shelves. See Sec. 3.3 for the effect of this resolution on ice shelf melting in an idealised case."*

L. 3-5 How did you blend these data sets together?

*BEDMAP2 is included by default in IBCSO. So no blending to do for these 2 data sets. Between the open ocean data set and shelf seas, the blend is done along the continental slope. Details are added to the text:*

*'The bathymetry used for the model domain north of the Antarctic continental shelf is that described by Megann et al., (2014). Over the Antarctic continental shelves the IBCSO data set (Arndt et al., 2013) is used. The two bathymetry data sets are merged between the 1000 m and 2000 m isobath along the Antarctic continental slope. Under the ice shelves, bathymetry (included in the IBCSO data set) and ice draft are taken from BEDMAP 2 (Fretwell et al., 2013).'*

L. 5 It might be worth mentioning the strange choice made in Sect. 5.2 of Fretwell et al. (2013) for ice shelf cavities with poorly sampled bathymetry, since you state later on that you had trouble with many of these ice shelves. Their choice might have been appropriate for ice-sheet modeling but it has the effect of making the bathymetry closely follow the ice draft with a very thin water column between them in many places. The resulting ocean circulation is likely completely false because the cavity geometry is essentially nonsensical. Because of this, many of us have resorted back to RTOPO1 or adopted newer gravity-inversion data sets for the ice shelves where this technique was applied. "We tested for areas where ice-shelf thickness and sub-shelf bathymetry falsely indicated grounded ice, and where necessary, enforced flotation by lowering the (poorly sampled) sea bed. We did this by interpolating the thickness of the sub-ice-shelf water column between the point where cavity thickness declined to 100m and the grounding line where cavity thickness is 0 m. This approach was required for Getz, Venable, Stange, Nivlisen, Shackleton, Totten and Moscow

University ice shelves, for some of the thickest areas of the Filchner, Ronne, Ross, Amery ice shelves and for the ice shelves of Dronning Maud Land."

*More details on this has been added in the limitation section to reflect the large number of ice shelf concerned by this flotation enforcement and the basic method to do it (see comment later on). In the description part we add a mention to this issue. The new text is:" bathymetry (included in IBCSO data set) and ice draft are taken from BEDMAP 2 (Fretwell et al., 2013). The resulting model bathymetry is shown in Fig. 6. Note that for some ice shelves, Fretwell et al. (2013) need to enforced the flotation by lowering the sea bed. In addition, we impose a minimum of two vertical grid cells within the ocean cavities so that an overturning cell can develop."*

L. 9 "Moscow University" There seems to be confusion in the literature about which is Dalton and which is Moscow University. The Australian groups whose research seems to be most focused on these shelves prefer to call the larger 2 shelves Totten and Dalton, with Moscow University as the small shelf between the 2. Rignot et al. instead use Moscow University to refer to Dalton. While I don't know for sure who is correct, I would tend to defer to the Australians and call this Dalton.

*As the Australians are expert in this area, we will follow your advice. In the figure 13 as well as in the text, all the occurrences of Moscou University are replaced by Dalton. We add a note of Moscou University ice shelf in the text when we mention Dalton ice shelf the first time: "Totten and Dalton (Moscow University in Rignot et al., 2013) ice shelves"*

L. 10-11. This was covered above and can be removed. *DONE*

P. 13

L. 11 "associated with" would maybe be more correct as "in addition to". To my knowledge, Nakayama et al. (2014) attributes the fresher coastal current to sub-ice-shelf melting and does not draw any direct causal connection to weak winds in the atmospheric forcing. *DONE, you are right. We change the text as suggested.*

P. 14

L. 25 "the deficiency in representing the giant ice shelves..." It is not clear from the context here which deficiencies you mean. You presumably mean the lack of horizontal circulation due to not explicitly representing the ice shelf cavities. *We changed the text like this "the deficiency in representing the ocean circulation beneath the giant ice shelves…"*

L. 31 I would change "Nevertheless, current coarse resolution..." to "This may not be a significant problem because current coarse resolution..." *DONE*

P. 15.

L. 30-31. Any idea why the water on the continental shelf has such a large warm bias here?

*We do not looked in detailed on what is going wrong in this area. The figure 14 suggests that part of the error is coming from an error off shelf. In addition, Williams et al. (2016) mentioned that "only heavily modified mCDW is present on the continental shelf". The understanding on the processes responsible on the presence of CDW on the continental shelf is not yet fully understood and could*

*need eddy resolving resolution (St-Laurent et al., 2013). Furthermore, coastal polynyas are playing a key role in this area. Because polynya are small scale features and are really affected by local wind, it is possible that in our model this feature are not well represented. As this point is out of the scope of the paper, no change in the text are made.*

*St-Laurent, P., J.M. Klinck, and M.S. Dinniman, 2013: On the Role of Coastal Troughs in the Circulation of Warm Circumpolar Deep Water on Antarctic Shelves. J. Phys. Oceanogr., 43, 51–64, doi: 10.1175/JPO-D-11-0237.1.*

*Williams, G.D., Herraiz-Borreguero, L., Roquet, F., Tamura, T., Ohshima, K.I., Fukamachi, Y., Fraser, A.D., Gao, L., Chen, H., McMahon, C.R., et al. (2016). The suppression of Antarctic bottom water formation by melting ice shelves in Prydz Bay. Nature Communications 7, 12577.*

P. 16

L. 4-5 "integrated melt rate" Elsewhere, you use "total melt" for this concept, so maybe here as well. *DONE*

L. 11 In the case of Getz, might the problem be the bad BEDMAP2 bathymetry? *See point L. 13-15 below*

George VI is more complicated but the BAS observations (which you could cite here -- Kimura and Venerable papers come to mind -- contact me if you don't know which ones I mean) show stairstep stratification that is likely poorly represented by the boundary-layer formulation assumed in the 3 equations and related heat and freshwater fluxes.

*We added comments in the limitation section to highlight the possible deficiency of the "three equation" formulation in some case. The text added is: " Recent observations beneath George VI ice shelf exhibit thermohaline staircases in the top 20 m below the melting ice shelf base, due to double-diffusive convection (Kimura et al., 2015). These observations raise a doubt about the applicability of the widely used three-equation model to predict the melt rate in regions where the flow beneath the ice shelf is weak. More experiments, observations, and numerical simulations are needed to fully understand the role of turbulence and thermohaline staircases controlling the heat flux to melting ice shelves."*

L. 13-15 These studies used RTOPO1 bathymetry for Getz. They may have reasonable melt rates at George VI for the wrong reasons (e.g. cold water masses than observed or poor circulation).

*It is a very good point. We check the Schodlock et al. (2016) (which have similar melt as us in Getz and George VI) and they are using IBCSO + BEDMAP2 as in our NEMO configuration. So we added a 3$^{rd}$ point on the possible inter-model differences causes. The new text is:*

*"R_MLT estimates are also well above earlier estimates obtained with FESOM by Timmermann et al. (2012) and Nakayama, et al. (2014) with RTOPO1 bathymetry (Timmerman et al., 2010), respectively, 164 and 127 Gt y$^{-1}$ for Getz Ice shelf, and 86 and 88 Gt y$^{-1}$ for George VI Ice Shelf. However, Schodlok et al, (2016) obtained similar melt rates using MITgcm with IBCSO bathymetry (respectively 303.9 and 373.1 Gt y$^{-1}$).*

*These large inter-model differences could have three causes. Firstly, the bathymetry and ice shelf draft data used in Timmermann, et al. (2012) and Nakayama, et al. (2014) come from RTOPO1, whereas Schodlok et al. (2016) and the present study use bathymetry data from IBCSO and ice shelf draft data from BEDMAP2. Differences in ice shelf geometry and bathymetry, particularly the height of seabed sills, can strongly affect ice-shelf melting (Rydt, et al., 2014).*

*Secondly, the ability of off-shelf CDW to cross the shelf break …"*

L. 22 FESOM uses a sigma coordinate only near continental margins. In the deep ocean, it is a z-level model. Maybe state this as "while FESOM uses a sigma-coordinate around the Antarctic continental margin." *DONE*

P. 17

L. 5-7 The bathymetry is not extrapolated from the surrounding region. Instead, the cavity thickness is extrapolated. This leads to ridiculously thin cavities in many, many places. The ice draft and the (completely made up) bathymetry may vary in tandem in the vertical over many ocean thicknesses, maintaining a thin ocean cavity between them. Nothing like this happens in any of the sub-ice-shelf cavities where observations are available, so (to beat a dead horse) this choice of interpolation was not appropriate for ocean modeling applications.

*The limitation paragraph was corrected and rewrite to add useful information on the concerned ice shelves. The new text is:*

*"The most recent bathymetry and ice shelf draft reconstruction of the Amundsen Sea (Millan et al., 2017) shows features that are missing in the BEDMAP2 data-set. In BEDMAP2, for many ice shelves, there are only indirect observations of ice draft, based on satellite surface elevation data, while the sub-ice bathymetry data are often poorly constrained. For some ice shelves (Getz, Venable, Stange, Nivlisen, Shackleton, Totten and Dalton ice shelves, some of the thickest areas of the Filchner, Ronne, Ross and Amery ice shelves and for the ice shelves of Dronning Maud Land), the flotation condition had to be enforced by lowering the sea bed arbitrarily from a level that itself was based on nothing more than extrapolation of cavity thickness from surrounding regions of grounded ice and 100 m thick cavity. Consequently, more data are needed for effective modelling (Fretwell, et al., 2013), because cavity geometry has a major impact on the simulated melting by controlling the water mass structure and circulation within the cavity (Rydt, et al., 2014)."*

L. 15 I'm not sure what is meant by "the friction law directly". I would take out the word "directly" or replace it with a clearer explanation of what, besides the friction coefficient, is meant here.

*'Directly' is now removed*

L. 17 "is very sensitive" I would recommend against using subjective phrases like "very sensitive" if you can be more quantitative. Maybe just drop "very".

*We drop 'very'*

P. 18

L. 2 "very sensitive" again, I would drop "very" (or be more quantitative).

*We drop 'very'*

L. 2-5 You imply that the finer resolution solution is the more realistic but this assumes that the true boundary layer is correctly represented at high vertical resolution. It is not clear to me that this is the case, at least for realistic configurations. Unless the vertical viscosity and diffusivity (or another parameterization of turbulent mixing) are being adapted in such a way as to correctly represent the physics of turbulent mixing below the ice shelf, the finer resolution solution may markedly underestimate mixing and entrainment. Indeed, your Fig. 9 seems to suggest that this might be the case in NEMO (though processes outside of ice shelf cavities may also be responsible for the biases, of course).

*Based on the earlier comments on the result of the simulation I_5M30M (5 m resolution + 30m Losh top boundary layer) and your comments, the text has been changed. The reference to 'better' has been dropped as we do not assess in detailed the top boundary layer properties. Reference to the sensibility of the result to the definition of the top boundary layer has been added. New text is:*

*"Losch et al. (2008) using the MITgcm model. Ice shelf melting appears to be sensitive to vertical resolution and the top boundary layer definition. When the Losch top boundary layer thickness is fixed, results are independent of vertical resolution, converge toward those obtained with a vertical resolution equal to that of the top boundary layer. When top boundary layer thickness changes with the vertical resolution under melting conditions, models simulated a cold, fresh, top boundary layer that tends to decrease the thermal forcing and thus the simulated melt rate.. At coarse resolution, the cold, top boundary layer is absent, leading to much larger melt rates."*

L. 24 Once again, maybe replace "very dependent" with something more quantitative.

*We did not change the text as detailed are given in the 2 following sentences:*

*'The effects on sea ice are very dependent on the amount of ocean heat available at depth. Over warm water shelves, the CDW entrained into the cavity overturning circulation warms the surface layer all year long and thus restricts the sea ice formation. This warming of the surface layer leads to thinning of the sea ice by more than 1m in coastal regions of the Bellingshausen and Amundsen seas (2m locally). Over cold water shelves, including the sub-ice-shelf cavities has a smaller effect on sea ice thickness (less than 20 cm).'*

P. 19

L. 12 "prescribe the melt rate" might be clearer if it were replaced with "distribute the melt fluxes". It is unclear if "prescribe the melt rate" refers to computing is or distributing it, and you don't explain how to compute the melt rate. *DONE*

L. 31-32 While these processes might very well be important elsewhere (e.g. Greenland), it's not clear that melting on ice faces can be a first-order effect in Antarctica. The areas of calving faces are so small compared with ice-shelf bases that the melt rates would need to be many orders of magnitude higher than those at the base of the ice for them to play a significant role in ice loss. Furthermore, melting at calving faces can indirectly be accounted for in the calving flux of icebergs. So I don't see these effects being of primary importance for coupled ice sheet/ocean modeling in the Antarctic.

*The conclusion has been changed. As you mentioned, this piece of work (vertical ice wall) is not in the first priority for an Antarctic ice sheet/ocean coupling as the melt along this face is expected to be very small and as this work is not global and do not mentioned Greenland at all, we removed this sentence from the conclusion. Instead we highlight the fact parametrisation are needed to represent the grounding line processes if the resolution is not fine enough to represent the cavity in these location. The new text is:*

*'To apply this work to a global coupled ice sheet/ocean model, we will need some further developments. First, a better knowledge of sub-ice-shelf cavity geometries and key processes that contribute to melting (drag, tides, boundary layer, etc ...) could lead to improvements in the ice shelf representation. Secondly, parameterisations need to be developed to represent the processes (melt and circulation) where the resolution is not fine enough to represent the ice shelf cavity geometry correctly as at the grounding line for example.. Finally, a conservative wetting and drying scheme needs to be developed to allow the grounding line (and calving front) to move back and forth. '*

P. 24

L. 6-7 It would be good to provide a URL, since this is not a journal article. Unfortunately, I am not aware of a working link so you may need to email Ben Galton-Fenzi to get him to put it somewhere permanent (like the other ISOMIP link I mentioned above).

*The server is now back in service. URL has been modified accordingly*

Fig. 2: Sign of panel c) is wrong.

*We decide to keep the same sign convention as this is the one used in L08 figure. We adjusted the caption.*

Fig. 4: Sign of both panels is wrong.

*We decide to keep the same sign convention as this is the one used in L08 figure. We adjusted the caption.*

Fig. 5: Sign of depth is wrong (should be negative to match other plots). Titles of b) and c) are a bit misleading because the difference only applies to the temperature (colormap) not the overturning. In the caption, I would explicitly state that the MOC is in contours. As it is, it seems like you assume the reader will notice the MOC first (and that this is the primary piece of information being shown) and that the temperature is secondary. For me, the opposite was true: I noticed the temperature first.

*The sign of depth is changed. We modify the caption accordingly. No change in the title. The color background is first described and then the contour. The new caption is:*

*'Figure 5: (a) Zonal mean temperature (°C) after 10 years of the run. In contour, the meridional overturning stream function (MOC) in the A_ISF experiment. b) Mean temperature difference (°C) with respect to A_ISF experiment (A_PAR-A_ISF). In contour the overturning stream function in the A_PAR experiment. c) as b) but for A_BG03.'*

Fig. 6: Nice figure!

Figs. 7-8: Maybe remind the reader that the model data is averaged over simulation year 10. Add citations for WOA. (I think they might be different for PT and for S.) *DONE, we add this precision also in Figure 14.*

Fig. 13: White was not the best color choice for zero melting because it is hard to tell the difference between absence of ice shelves and presence but with zero melting. This figure is the only one zoomed in enough to give us a sense of how well resolved the smaller ice shelves are. I would suggest using light gray either from zero melting or for the background of each panel so the two can be distinguished (with slight preference for the latter). *DONE, ocean and grounded ice are set in light gray.*

**Typographic and grammatical corrections:**

Line numbering: For future manuscripts, it would be more helpful if line numbering continues through the whole manuscript (as in Latex) rather than being for each page. This makes the review process easier.

P. 1:

L. 13 comma needed after "at the surface)" *DONE*

L. 16 "...under ice shelf seas overturning circulation..." This is an awkward phrase. Might I recommend, "...overturning circulation under ice shelves..."? *DONE*

L. 17 comma missing after "at the surface" *DONE*

L. 17-18 "It yields similar improvements... than the explicit..." In this sentence, "than" should be replace with something like "to those from" (i.e. "similar to", rather than "similar than"). *DONE*

L. 19 "widely used" does not need a hyphen; "3 equations" should be "3 equation" or possibly "3-equation" *DONE. "3 equations" replace by "3 equation" as in the rest of the text.*

P. 2:

L. 14 "...inflowing water mass that could..." should be "...inflowing water mass, which could..." *DONE*

L. 18 no comma needed after "high" *DONE*

L. 18 This is kind of picky, I know, but I would change "...meling can be high..." to "...melt rates can be high...", since melting is kind of a state of being that, to me at least, isn't really high or low. *DONE*

L. 30-33 "Furthermore" seems to imply that the second of these two sentences follows from the first, but they are not really related. I would suggest changing the second sentence to something like, "Global conservation is also an important issue, as the ocean/sea-ice model is used as a component within Earth System Models." *DONE*

L. 32 comma needed after "this issue" *DONE*

L. 33-34 I would change "the z* vertical coordinate" to "a z* vertical coordinate". *DONE*

P. 3:

L. 20 "nonlinear" does not need a hyphen *DONE*

P. 4:

L. 14 "the z axis in" should be "the z axis of" *DONE*

L. 31 "sigma coordinates models" should be "sigma coordinate models" *DONE*

P. 5:

L. 20 I would much prefer TBL to tbl (and FWF to fwf later on).  It is much easier to read and to spot the definition if you encounter the acronym later on and need to be reminded what it stood for.

*We decided to remove the acronym fwf as it was used only once. We also remove the acronym tbl. We add the acronym $H_{TBL}$ for top boundary layer thickness in the section 2.2 for the discussion on the Losh parametrisation.*

L. 23 Here and many other places, you use a period to add spacing to your units.  In Latex, the correct way to do this is with a half-space (\,).  I suspect this manuscript was written in Word, so I don't know how a half-space is achieved and would recommend a full space instead.  In any case, a period is not correct. *DONE*

P. 6:

L. 8 "(Q h) " the second parenthesis should not be subscript. *DONE*

L. 12 "tbl" again better as "TBL" *Acronym removed*

L. 13-14 "(Jenkins et al., 2010)" should be "Jenkins et al. (2010)" *DONE*

L. 17 move "(Jenkins et al., 2010)" (no comma) to after "their values" and change "are based on" to "is based on". Should now read: "Furthermore, uncertainties in the Stanton numbers are also large, as the study used to determine their values (Jenkins et al., 2010) is based on data from a single borehole." *DONE*

L. 18 "Eq. 7-12" should probably be "Eqs. 7-12" and "Eq. (10) to (12)" should be "Eqs. 10-12". *DONE*

L. 25 "smallest" should be "thinnest" (or "thickest" should be "largest") for consistency. *DONE*

P. 7:

L. 10 comma missing after "...parameterisation is that" *DONE*

L. 12 "fresh water" should be "freshwater" *DONE*

L. 16 I would prefer "FWF" to "fwf".  Please define the acronym FWF here. *Acronym removed*

L. 17 Use "BG03" instead of the full citation, since you took the trouble to define a shorthand. *DONE*

P. 8:

L. 10-11 In my experience, URLs are most cleanly done as footnotes. They could also be done as citations, in which case you need an author and the last date they were accessed. Also, John Hunter's website is now down. I had asked Ben Galton-Fenzi to post it on a more permanent place. That place ended up being Ben's staff website, which also now seems to have gone down. I would suggest contacting Ben to get this website posted somewhere permanent (once again!).

*Ben Galton-Fenzi said me (personal communication) that they experiment difficulties in their server. The server is now back to life. URL has been updated.*

L. 11 "(Asay-Davis, 2013)" this citation isn't in the bibliography. Is this my EGU presentation?

*We changed the reference to the Asay-Davis, 2012 presentation at NCAR workshop where the ISOMIP results are presented. We added in the reference the web link with the references*

L. 19 Here and elsewhere, "m/y" should probably be "m y-1" or "m a-1" . *m/y replaced by "m y-1" in the all the text.*

L. 21 "Figure 2" should be "Fig. 2". Also, you are showing melt rates but you have never explicitly said how melt rates are computed from q. Presumably they are in m a-1 Of freshwater and are positive for melting (as stated in the figure caption). In this case, the field plotted in Fig. 2 needs to be multiplied by -1 (i.e. you're plotting positive freezing).

*Yes, the melt rate is in m y-1 of freshwater. We keep the sign convention as it is to fit the convention used in the ISOMIP figure from L08.We corrected the caption.*

L. 21 "similar to the one" would be slightly better as "similar to that" *DONE*

L. 25 "Losch" should probably be "L08" *DONE*

L. 29 "top boundary layer" could be "TBL" *Acronym removed*

L. 30 "9 simulations" should, I think, be "nine simulations". The rule I learned was to write out numbers ten or smaller. *DONE*

P. 12:

L. 20 and 22 "Fig. 7-8" and "Fig. 7, 8 and 10" should be "Figs. 7-8" and "Figs. 7, 8 and 10" *DONE*

P. 13

L. 1 "Figure 7-8" should be "Figs. 7-8". *DONE*

L. 8 "10y" should be written out as "ten years" *DONE*

L. 17-19 Consider reorganizing for clarity: "The position of the ice edge, being too far south in the Amundsen Sea and too far north in the Weddell Sea and around East Antarctica in both simulation, is not changed significantly by the presence of ice shelf cavities (Fig. 11)." When I first read this, I thought the presence or absence of ice shelves was related to the location of the ice edge, whereas you want to point out that these biases exist regardless. *Reorganized as suggested*

L. 25-26 "Sea ice is thus thinner in R_ISF than in R_noISF..." This was already stated above. *The sentence was removed.*

L. 30 "as the impact..." should be "as is the impact..." *DONE*

P. 14

L. 12 "similar in both" should be "similar between" ("similar" implies a relationship between two things, not a property of both things.) *DONE*

L. 18 More grammatical would be "In R_PAR, this is due to the lack of a HSSW circulation..." *DONE*

L. 32: "1o x cos (latitude)" This is some strange formatting with a mix of math and text as well as notation that is not very standard. Maybe "a nominal resolution of 1o cos($\theta$), where $\theta$ is the latitude, which is sufficient to..." *DONE*

P. 15

L. 26 No spaces in "(51-260 Gt y-1)". *DONE*

P. 17

L. 3 "our model setup as the large..." should probably be "our model setup as well as the large..." *DONE*

P. 18

L. 16 "observed" I would opt of another word like "seen" because "observed" seems to imply "observations" to me, which is not your intent here. *We changed the sentence to "… High Salinity Shelf Water (HSSW) simulated in R_noISF is slightly less dense than observations …"*

P. 19

L. 13 "physically sensible" does not need a hyphen. *DONE*

Table 1: many symbols have not been properly subscripted (Cp, Lf, Cpi, Rhoi, Cd). Rhoi should be the Greek symbol rho, right? Remove dots in the units. *DONE*

Table 2: Many of the ISOMIP experiments are not explicitly mentioned in the text. The names need to be explained either here in the caption or in the text, particularly 31L, 46L and 75L.

*We found the explanation gave in table 2 to describe every experiments used in Figure 3 clear enough. Some experiments are only used in Figure 3 and not in the text in order to support the conclusion made, but we do not think there is a need to detail the results of every single experiment as I_10M for example. Some minor changes have been made in the text:*

*"To evaluate the impact of this choice on the ocean circulation beneath the ice shelf, nine simulations with vertical resolution ranging from 5 m (I_5M) to 150 m (I_150M) have been carried out (Table 2)."*

*"With variable vertical resolution (I_31L, I_46L and I_75L), such as is typically used in global configurations of NEMO (Timmermann et al, 2005, Drakkar group, 2007 and Megann et al., 2014), the coarsest resolution in the cavity seems to determine the total melt."*

Table 3: In the caption, it would be good if you could define ++/+/0/-/-- more quantitatively. Otherwise, this seems rather subjective. Regarding the table itself, I don't think GMD is likely to let you format the table the way you have it here (see instructions for authors). Specifically, they are unlikely to support color like this. You do have some control over horizontal lines, and this may be the best way to differentiate the different regions. I don't think you explicitly discuss the last two columns of the table in the text, which it seems like you should.

*Colors have been removed and replace by thick/double lines. Further explanations on have been given in the caption: ++/+/0/-/-- is a summary of the ocean temperature condition at the closest non-extrapolated cell in the WOA2013 observational dataset (Fig. 14). ++ for ocean temperature differences wrt WOA2013 of more than 1˚C, + differences in the range 0.5 and 1˚C, 0 differences in the range 0.5 and -0.5 ˚C, - differences in the range -0.5 and -1 ˚C and -- for ocean temperature differences greater than -1 ˚C.".*

*The last two columns are not explicitly described. It is a reminder for the reader of the comments made along the text on the issue we have on the water properties in front of these ice shelves and about the representation of the cavity geometry itself which can affect the ice shelf melt. No change in the text are made.*

---

## Author Response (AR2)

[revised manuscript text omitted]

**Topical editor: Sophie Valcke**

**Specific comments:**

Thank you very much for your updated manuscript and detailed answers. I consider that the manuscript can be published after the following technical corrections have been taken into account (the pages and line numbering refer to version2 of your manuscript) :

- p8, L244: given the discussion with referee 2, maybe it would be better to write "at which time the system is close to steady state." *DONE*

-p9, L274-275: please add "(not shown)" at the end of the sentence as these numbers cannot be inferred either from Fig.3 nor Fig.4 *DONE*

-p.10, L307: I don't understand if the vertical eddy viscosity and diffusivity are increased from 1 m2s-1 to 10 m2s-1 for the whole simulations or only dynamically "under unstable conditions" (i.e. when unstable conditions are detected, and if this is the case what is the criteria analysed)? From the way the sentence is written, it looks like it is adjusted dynamically but I would think it is fixed to "10 m2s-1".

*The sentence has been reformulate to 'The vertical eddy viscosity and diffusivity, in unstable conditions, is set to 10 $m^2.s^{-1}$ (instead of 0.1 $m^2.s^{-1}$ in ISOMIP configuration) to reduce the noise generated along the ice shelf front.'*

-p12, L350 and p.18, L549: is it "flotation" or "floatation"?

*It is flotation, corrected and check through the entire text.*

-p14, L472, please remove the comma after "because" *DONE*

-p19, L583, please remove "the" in "vertical resolution and the top ..." *DONE*

-p19, L585, please add "and" before "converge" *DONE*

-p19, L586, please put "simulate" instead of "simulated" to use present tense like in the rest of the paragraph *DONE*

-p19, L588, please remove one period at the end of the line *DONE*

-p21, L648, please remove one period after "for example" *DONE*

-p34, Fig. 5 captions: in (a) and (b), please change ". In contour," for "; in contour," *DONE*

1025    -Figs. 7, 8 and 14 captions: contrary to what you write in your reply, the precision that the data is averaged over simulation year 10 is not added. *DONE*